# GMAI-MMBench: A Comprehensive Multimodal Evaluation Benchmark Towards General Medical AI

**Pengcheng Chen**[1,2*] **Jin Ye**[1,3*†] **Guoan Wang**[1,4*] **Yanjun Li**[1,4]
**Zhongying Deng**[5] **Wei Li**[1,6] **Tianbin Li**[1] **Haodong Duan**[1]
**Ziyan Huang**[1,6] **Yanzhou Su**[1] **Benyou Wang**[7,8] **Shaoting Zhang**[1]
**Bin Fu**[9] **Jianfei Cai**[3] **Bohan Zhuang**[3] **Eric J Seibel**[2] **Yu Qiao**[1,9] **Junjun He**[1‡]

[1]Shanghai AI Laboratory    [2]University of Washington    [3]Monash University
[4]East China Normal University    [5]University of Cambridge    [6]Shanghai Jiao Tong University
[7]The Chinese University of Hong Kong, Shenzhen    [8]Shenzhen Research Institute of Big Data
[9]Shenzhen Institute of Advanced Technology (SIAT), Chinese Academy of Sciences

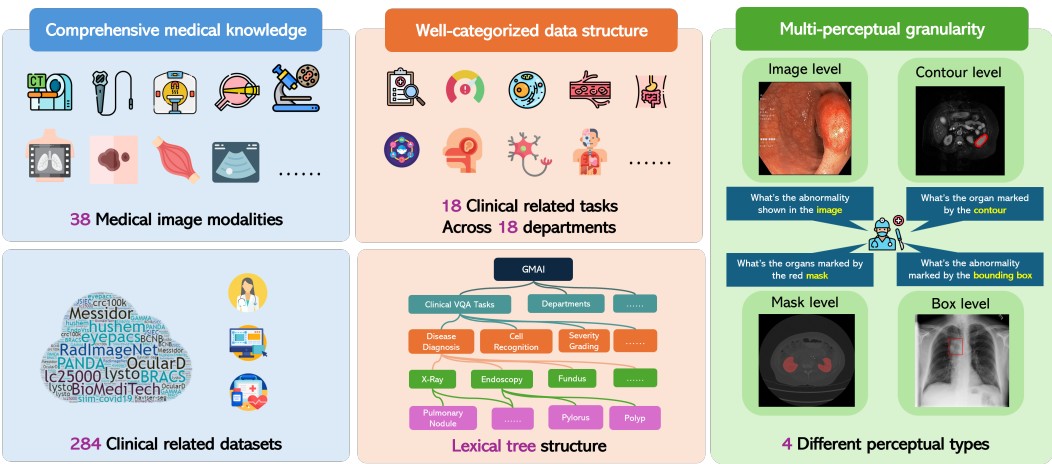

Figure 1: Overview of the GMAI-MMBench. The benchmark is meticulously designed for testing LVLMs' abilities in real-world clinical scenarios with three key features: (1) Comprehensive medical knowledge: It consists of 284 diverse clinical-related datasets from worldwide sources, covering 38 modalities. (2) Well-categorized data structure: It features 18 clinical VQA tasks and 18 clinical departments, meticulously organized into a lexical tree. (3) Multi-perceptual granularity: Interactive methods span from image to region level, offering varying degrees of perceptual details.

## Abstract

Large Vision-Language Models (LVLMs) are capable of handling diverse data types such as imaging, text, and physiological signals, and can be applied in various fields. In the medical field, LVLMs have a high potential to offer substantial assistance for diagnosis and treatment. Before that, it is crucial to develop benchmarks to evaluate LVLMs' effectiveness in various medical applications. Current benchmarks are often built upon specific academic literature, mainly focusing on a single domain, and lacking varying perceptual granularities. Thus, they face specific challenges, including limited clinical relevance, incomplete evaluations,

---

[*]These authors contributed equally to this work.

[†]Project lead.

[‡]Corresponding author: hejunjun@pjlab.org.cn

38th Conference on Neural Information Processing Systems (NeurIPS 2024) Track on Datasets and Benchmarks.

and insufficient guidance for interactive LVLMs. To address these limitations, we developed the GMAI-MMBench, the most comprehensive general medical AI benchmark with well-categorized data structure and multi-perceptual granularity to date. It is constructed from 284 datasets across 38 medical image modalities, 18 clinical-related tasks, 18 departments, and 4 perceptual granularities in a Visual Question Answering (VQA) format. Additionally, we implemented a lexical tree structure that allows users to customize evaluation tasks, accommodating various assessment needs and substantially supporting medical AI research and applications. We evaluated 50 LVLMs, and the results show that even the advanced GPT-4o only achieves an accuracy of 53.96%, indicating significant room for improvement. Moreover, we identified five key insufficiencies in current cutting-edge LVLMs that need to be addressed to advance the development of better medical applications. We believe that GMAI-MMBench will stimulate the community to build the next generation of LVLMs toward GMAI.

🌐 Website: https://uni-medical.github.io/GMAI-MMBench.github.io/
🤗 Huggingface: https://huggingface.co/datasets/OpenGVLab/GMAI-MMBench
📁 OpenDataLab: https://opendatalab.com/GMAI/MMBench
🧩 Evaluation: https://github.com/open-compass/VLMEvalKit [64]

## Introduction

In clinical practice, diverse demands may be proposed by different medical institutions for disease diagnosis and treatment. These demands can be potentially fulfilled by general medical AI which provides general-purpose medical models to tackle a wide range of medical tasks. Such models are typically Large Vision-Language Models (LVLMs) trained on diverse data types, including imaging and clinical texts, to tackle diverse tasks, e.g., disease diagnosis and severity grading. Noticeably, the state-of-the-art LVLMs, including general-purpose ones (e.g., DeepSeek-VL [155], GPT-4V [5] and Claude3-Opus [13]) and medical purposes (like MedDr [95], LLaVA-Med [138], and Med-Flamingo [181]), have both demonstrated promising performance in some medical visual-textual tasks. However, it remains unclear to what extent these LVLMs can accommodate the diverse demands in real clinical scenarios. To validate their effectiveness and promote their application in clinical practice, it is crucial to establish a comprehensive benchmark to address diverse real-world demands. Therefore, an ideal benchmark should achieve three specific aims:

**Aim 1. Comprehensive medical knowledge.** Medical knowledge is embedded in medical data, so comprehensive medical knowledge requires diverse medical data of different modalities from various data sources. In clinical scenarios, various types of imaging modalities, including X-rays, Computed Tomography (CT), Magnetic Resonance Image (MRI), Ultrasound Imaging, Positron Emission Tomography (PET), etc, are employed for diagnostic and therapeutic purposes, reflecting different aspects of medical knowledge [267]. Besides, to encompass the diverse medical knowledge from different clinical facilities, the data used in a comprehensive benchmark should cover a range of different clinical institutions and hospitals which are preferably distributed across the world [205]. These demands favor benchmarks collected from diverse sources. **Aim 2. Comprehensive evaluation across all clinical aspects.** A comprehensive benchmark should be easily customized to evaluate any specific abilities of LVLMs for each clinical professional. This property is necessary because there are an excessive amount of clinical institutions, departments, and practitioners, each having their own specific demand. Their potential demands can be concluded in two sides: 1) *Evaluation across diverse tasks*. Some clinical practitioners may require MRI data for disease diagnosis while others may need to deal with surgical workflow recognition for computer-assisted or robot-assisted surgery systems. Therefore, a comprehensive benchmark should cover all clinical demands by encompassing a sufficient number of diseases and tasks. 2) *Evaluation for diverse clinical departments*. Some departments may be interested in LVLMs' performance on oncology-related tasks only while others may only focus on urology-related ones. As such, a comprehensive benchmark should be easily used for customized evaluation to accommodate the diverse demands of different clinical departments. These demands further require the benchmark to be well-categorized to facilitate ease of use. **Aim 3. Interactive ability in multi-perceptual granularity.** Given a specific medical image, doctors need to look through the whole image (image level) for an overview while also requiring comprehensive

Table 1: Comparison between GMAI-MMBench and other existing benchmarks in the biomedical field. GMAI-MMBench is sourced from extensive data sources worldwide, offering comprehensive medical knowledge detailed in modalities, clinical tasks, departments, and perceptual granularities. Dept and PG indicate department and perceptual granularity, respectively. In the perceptual granularity types, I, B, M, and C denote image, box, mask, and contour, respectively. * indicates the test set.

| Benchmark | Modality | Size | Task | Dept | PG | Source |
|---|---|---|---|---|---|---|
| Medical-Diff-VQA* [105] | 1 | 70K | 7 | ✗ | I | MIMIC-CXR [120] |
| PathVQA* [96] | 1 | 6K | 7 | ✗ | I | Textbook, PEIR [1] |
| Cholec80-VQA* [222] | 1 | 9K | 2 | ✗ | I | Cholec80 [243] |
| VQA-RAD [136] | 3 | 3K | 11 | ✗ | I | Teaching cases from Medpix [2] |
| RadBench [254] | 6 | 137K | 5 | ✗ | I | 13 image-text paired datasets |
| MMMU (H & M) [262] | 6 | 2K | 5 | ✗ | I, B | Exam, Quiz, Textbook |
| SLAKE* [145] | 3 | 2K | 10 | ✗ | I | MSD [227], Chestx-ray8 [250], CHAOS [127] |
| OmniMedVQA [106] | 12 | 128K | 5 | ✗ | I | 73 classification datasets |
| GMAI-MMBench | 38 | 26K | 18 | ✔ | I, B, M, C | 284 datasets from both public and hospital |

explanations in a specific position (mask level) or region (box level). This demand requires LVLMs to perceive the granularity range from a specific position to the entire image. Thus, a comprehensive benchmark should also evaluate LVLMs' perceptual granularity.

As shown in Table 1, there are some medical benchmarks, such as Medical-Diff-VQA [105], PathVQA [96], Cholec80-VQA [222], and Cholec80 [243], dedicated to evaluating specific abilities of LVLMs. These benchmarks effectively assess the performance of LVLMs within a particular modality or task, thereby facilitating the optimization of models for specific applications. Nonetheless, their limited modalities and tasks cannot meet the requirement of modal and task diversity. Other benchmarks including VQA-RAD [136], RadBench [254], and MMMU (Health & Medicine) [262] address this issue by providing multiple modalities and tasks for evaluation, with data consisting of natural image-text pairs sourced from academic papers, textbooks, and specific databases. Though these benchmarks significantly enhance the breadth and depth of medical assessment, they may not accurately reflect actual clinical requirements, as their sources are distant from clinic practice and prone to data leakage [44, 72]. More importantly, *none of these benchmarks can be customized to evaluate various abilities of LVLMs to accommodate highly diverse clinical demands* because their data are not well categorized. For instance, it is hard to obtain the dimension, modality, and task information of a specific data point in these datasets, which prevents a clinical professional from evaluating LVLMs using the CT (modality) of 2D (dimension) images for blood vessel recognition (task). Due to this, they can hardly be used for customized evaluation. In summary, though existing medical multimodal benchmarks provide valuable evaluation frameworks, they present challenges in fully addressing clinical needs. Future developments necessitate more refined and customized benchmarks that are closely aligned with real-world clinical applications.

To address these challenges, we introduce the General Medical AI MultiModal Benchmark (GMAI-MMBench), a comprehensive multimodal benchmark that is well-categorized for medical image understanding and reasoning in real-world clinical scenarios. As shown in Figure 1, its comprehensiveness can be concluded in three aspects: 1) **comprehensive medical knowledge from diverse modalities, tasks, and data sources**, 2) **well-categorized in lexical tree structures**, and 3) **multiple perceptual granularity**.

First, GMAI-MMBench has diverse modalities and data sources because it is built upon 284 high-quality datasets collected across the world. These 284 datasets cover various medical image tasks, including 2D detection, 2D classification, and 2D/3D segmentation, to ensure the diversity of tasks. Using these foundational visual-based tasks has two advantages over using off-the-shelf image-text pair data. 1) It minimizes the risk of data leakage since the data in our benchmark are mostly image-label pairs rather than image-text pairs. The image-label pairs are not directly convertible to LVLMs training samples (usually image-text pairs), thus less likely to be used to train LVLMs; 2) It ensures high clinical relevance, as the images are sourced from hospitals and annotated by professional doctors. We then carefully selected approximately 26K cases with 38 different modalities to construct the GMAI-MMBench, thus meeting the modal diversity goal.

Second, GMAI-MMBench is a well-categorized medical benchmark that can comprehensively evaluate the pros and cons of various aspects of LVLMs, benefiting both model developers and users with specific needs. Specifically, we develop a categorization system, called lexical tree structure, which categorizes all cases into 18 clinical VQA tasks, 18 departments, 38 modalities, etc. The 'clinical VQA tasks' / 'departments' / 'modalities' are the lexicons that can be used to retrieve desired

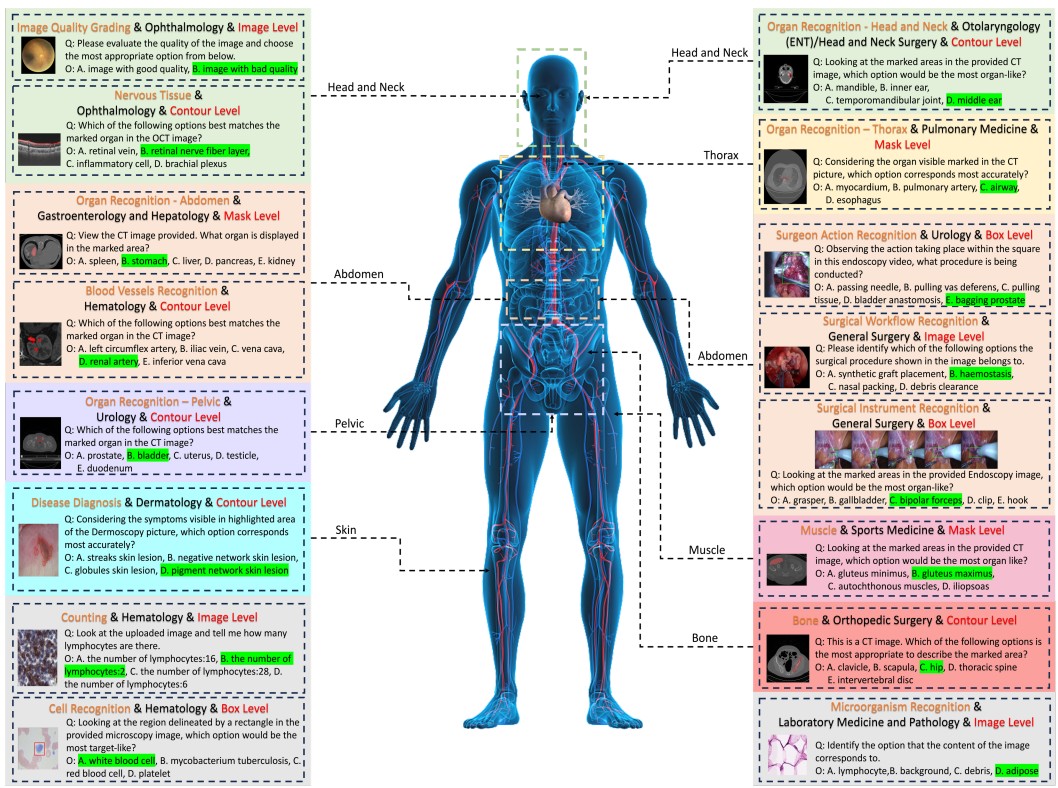

Figure 2: Examples of GMAI-MMBench. The benchmark covers a variety of clinical tasks, departments, and perceptual granularities from worldwide data sources.

cases for evaluation. For instance, the oncology department can select cases related to oncology to evaluate LVLMs' performance for oncology tasks, thus greatly enhancing flexibility and usability for specific demands.

Third, GMAI-MMBench can evaluate LVLMs' abilities to perceive different granularity, such as understanding the local image content in a mask or bounding box as well as recognizing the entire image content. This ability is important for detection, segmentation, and classification tasks as these tasks need different perceptual granularity for better performance. Furthermore, the perception of bounding boxes or masks is vital for interactive LVLMs [132], so the perceptual granularity evaluation in our benchmark can possibly be used to improve interactive LVLMs.

We assess 44 publicly available LVLMs (38 general purpose and 6 medical-specific models) as well as advanced proprietary LVLMs such as GPT-4o, GPT-4V, Claude3-Opus, Gemini 1.0, Gemini 1.5, and Qwen-VL-Max on our GMAI-MMBench. We summarize the key findings as follows:

(1) GMAI-MMBench presents significant challenges in clinical practice. Even the best proprietary GPT-4o only achieves an accuracy of 53.96%, which demonstrates the deficiencies of cutting-edge LVLMs in tackling medical professional issues, thus they can hardly fulfill diverse clinical demands.

(2) Open-source LVLMs, such as MedDr and DeepSeek-VL-7B, achieve approximately 44% accuracy, making them very competitive compared to proprietary models. For instance, they surpass Claude3-Opus and Qwen-VL-Max and achieve comparable performance to Gemini 1.5 and GPT-4V. However, they still exhibit a clear performance disparity compared to the top-performing GPT-4o.

(3) Most medical-specific models have difficulty reaching a general performance level (approximately 30% accuracy) achieved by general LVLMs, except MedDr with 43.69% accuracy.

(4) Most LVLMs exhibit unbalanced performance across different clinical VQA tasks, departments, and perceptual granularity. Notably, in the experiments on different perceptual granularity, box-level annotation consistently results in the worst accuracy, even worse than image-level annotation.

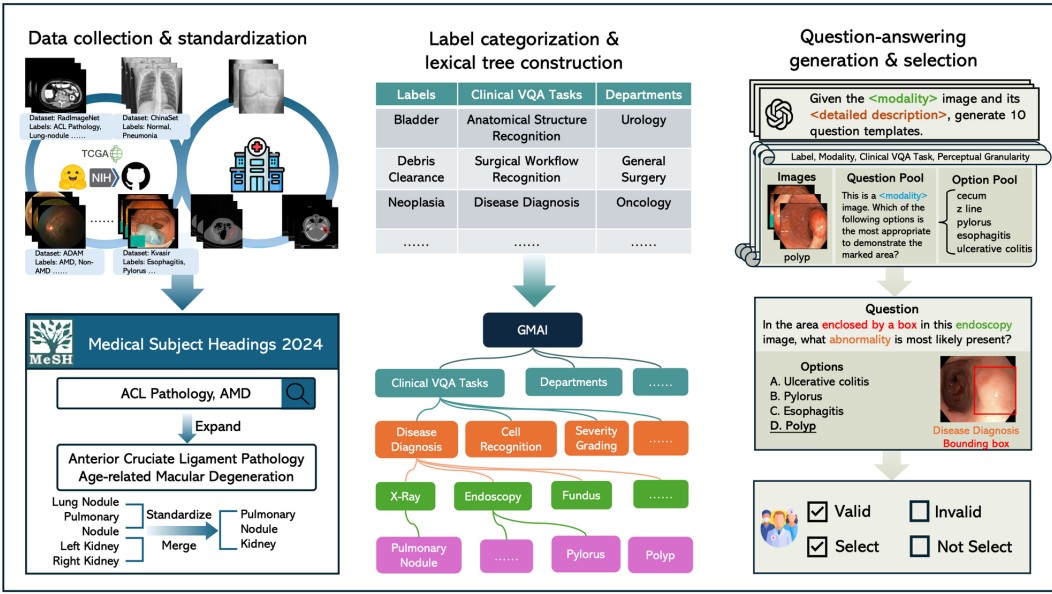

Figure 3: Overall illustration of GMAI-MMBench. The data collection can be divided into three main steps: 1) We search hundreds of datasets from both the public and hospitals, then keep 284 datasets with highly qualified labels after dataset filtering, uniforming image format, and standardizing label expression. 2) We categorize all labels into 18 clinical VQA tasks and 18 clinical departments, then export a lexical tree for easily customized evaluation. 3) We generate QA pairs for each label from its corresponding question and option pool. Each question must include information about image modality, task cue, and corresponding annotation granularity. The final benchmark is obtained through additional validation and manual selection.

(5) The major factors leading to performance bottlenecks include perceptual errors (e.g., misrecognition of image content), lack of medical domain knowledge, irrelevant responses, and rejection of answering questions due to safety protocols.

In summary, our contributions are three-fold. (a) We introduce a comprehensive benchmark, GMAI-MMBench, to evaluate existing LVLMs in clinical practice. GMAI-MMBench covers 38 modalities, 18 clinical VQA tasks, 18 departments, and 4 different perceptual granularity from 284 medical-related datasets, thereby offering a diverse range of modalities, tasks, and data sources. (b) GMAI-MMBench organizes each data point in lexical tree structures, with lexicons used to select desired data points to evaluate various aspects of LVLMs' abilities. Thus, GMAI-MMBench facilitates customized evaluation to meet highly diverse demands in clinical practice. **See Supplementary C.2**. (c) We evaluate 44 representative general-purpose LVLMs, including both open-source and proprietary models, as well as 6 medical-specific LVLMs on GMAI-MMBench. The comprehensive evaluation reveals the pros and cons of different LVLMs from diverse perspectives, providing insights to improve these models to accommodate real-world clinical applications.

## GMAI-MMBench

### Overview

We propose GMAI-MMBench, an innovative benchmark meticulously designed for the medical field, capable of providing comprehensive evaluations of LVLMs across various aspects of healthcare. (shown in the Figure 2) We collect 284 datasets from public sources and hospitals, covering medical imaging tasks of detection, classification, and segmentation, to form the data fuel for establishing such a benchmark. The detailed datasets are listed in the supplementary. Based on the data foundation, we design a reliable pipeline to generate question-answering pairs and organize them from different perspectives with manual validation. Finally, we carefully select approximately 26K questions with

varying levels of perceptual granularity from the manually validated cases to construct the final GMAI-MMBench.

**Benchmark Construction**

The detailed steps of constructing our GMAI-MMBench can be divided into three main steps as shown in Figure 3.

**Dataset collection and standardization.** As our aim is to build a large-scale benchmark for the comprehensive evaluation of LVLMs, the first and most important step is data collection. In contrast to benchmarks that directly use multimodal paired datasets, we source the datasets in two ways to minimize the data leakage problem and ensure the diversity and clinical property: First, we conduct thorough Internet searches to collect as many 2D/3D medical-related datasets as possible, retaining those that involve classification, detection, and segmentation tasks. Second, we collaborate with several hospitals that have agreed to share their ethically approved data. This process has enabled us to curate 284 datasets with highly qualified labels. Following data collection, we standardize both images and labels. For images, we adhere to the SA-Med2D-20M [258] protocol, transforming all 2D/3D medical images into 2D RGB images for further evaluation. For labels, we refer to the Medical Subject Headings (MeSH)[4] to ensure every label is unique, clear, and free from conflict or ambiguity within each task. Specifically, we focus on three main situations: (1) expanding all abbreviations, such as changing "AMD" to "Age-related macular degeneration"; (2) unifying different expressions for the same target, such as standardizing both "lung nodule" and "pulmonary nodule" to "pulmonary nodule"; (3) merging labels with left and right distinctions, such as combining "left kidney" and "right kidney" into "kidney", since our goal is to evaluate the abilities of understanding and reasoning rather than directional judgment.

**Label categorization and lexical tree construction.** We construct a well-categorized lexical tree to ensure GMAI-MMBench can be easily customized to evaluate the specific abilities of LVLMs for each clinical professional. The overview of the tree is shown in Figure 3, and the complete version is in supplementary. First, we integrate data properties and real applications to propose three subjects tailored for the biomedical fields: clinical VQA tasks, departments, and perceptual granularities. Specialized options are generated for each subject individually: For clinical VQA tasks, we extract keywords according to the original dataset descriptions and then lead to 18 categories. For departments, we refer to the Mayo Clinic[5] and assign all labels to 18 departments. For perceptual granularity, we construct 4 types based on annotation methods (see the rightmost panel in Figure 1). We then recruit several biomedical engineering university students (including coauthors) to tag labels from the constructed options in these subjects. Specifically, each label is randomly assigned to 3 people, and their tagging results are merged by voting. After label categorization, the lexical tree can be directly exported for customized evaluation. An example of customized evaluation is presented in Supplementary C.2.

**QA generation and selection.** Following the label categorization, all labels are assigned to specific modalities, clinical VQA tasks, departments, and perceptual granularities. Based on the well-organized structure, we generate the VQA pairs for every label with three steps. First, questions and options generation. For question generation, a question must include three key pieces of information in GMAI-MMBench: modality, clinical task hint, and perceptual granularity information. For each combination of the three elements, we randomly pick 10 labels and generate 10 candidate questions with GPT-4o for each selected label. These questions are then manually reviewed to meet the following criteria: (1) they must include necessary information on modality, clinical task, and perceptual granularity; (2) they do not include any hints that would allow the question to be answered without viewing the image. After manual review, the modality is replaced with a placeholder for standardization. For example, a valid question template for Disease Diagnosis in segmentation task is: "*This is a <modality> image. Which of the following options is the most appropriate to demonstrate symptoms in the marked area?*" Once the question pool is generated, each category has its question pool based on its tags of modality, clinical VQA task, and perceptual granularity. For options generation, the global view (image level) and local view (mask level, bounding box level, and contour level) of perceptual granularity are handled separately. For the global view, the option pool for each answer is sourced from the remaining categories within the answer's dataset to avoid introducing

---

[4]https://www.ncbi.nlm.nih.gov/mesh/1000048
[5]https://www.mayoclinic.org/departments-centers

multiple correct answers. For instance, a fundus image dataset may focus solely on pathological myopia, but the images might also contain other diseases like diabetic retinopathy. Including other categories could render the question invalid. For the local view, we construct a shared option pool for the answers with the combination of modality, clinical VQA task, and perceptual granularity. Second, as each answer with corresponding images has its own question and option pool, we generate all QA pairs for all images. For each image, we randomly select a question from its question pool and replace the placeholder with its modality. Along with the correct answer, we randomly select $n$ options (where $n = \mathbf{randint}(\mathbf{max}(1, \mathbf{len(option\ pool)}), \mathbf{min}(4, \mathbf{len(option\ pool)})$ from the corresponding option pool to create the set of options. Third, to ensure data quality and balanced distribution, we perform additional manual validation and selection. In the validation stage, we assess the QA pairs based on the following criteria: (1) We drop cases whose questions do not contain the three key components and can be answered without the image. (2) We filter out cases with incorrect answers. (3) We drop cases where images have unclear targets or poor image quality. In the selection stage, we choose 30 cases per answer to ensure balance across all tasks (all cases are included if the number is less than 30). The selection rule is based on the consideration of diversity: Selecting images with large differences in appearance, data source, age, gender, etc. As a result, we finalize 25831 QA pairs for the GMAI-MMBench (4550 in the validation set and 21281 in the test set).

## Experiments

### Experiment setup

In this study, we evaluated various LVLMs, including medical-specific, open-source, and proprietary API general models. We selected versions with approximately 7 billion parameters for testing, and the model weights were sourced from their respective official Hugging Face repositories. Our evaluation was conducted using the VLMEvalKit[6] framework and Multi-Modality-Arena[7].

The assessment was performed in a "zero-shot" setting. Specifically, our evaluation prompts did not include any example cues, and the models were required to perform inference on tasks without prior training or examples related to those tasks. This approach better tests the models' generalization capabilities and comprehension, examining their performance when confronted with novel problems. All tests were executed using NVIDIA A100 GPUs with 80GB of memory.

### Models

For completeness, we conducted evaluations using several state-of-the-art LVLMs to benchmark their performance on GMAI-MMBench, including both general models that have extended capabilities in the biomedical domain and medical-specific models that are meticulously trained for clinical medicine. By default, we use the latest, largest, and best-performing available checkpoint for each model family to ensure optimal performance. We picked 29 out of 50 models for demonstration in the main text, additional results are provided in the supplementary material. For medical-specific models, we include 5 latest powerful LVLMs: MedDr [95], LLaVA-Med [138], Med-Flamingo [181], RadFM [254], and Qilin-Med-VL-Chat [149]. For general models, we test 18 representative LVLMs: TransCore-M [3], VisualGLM-6B [61], mPLUG-Owl2 [259], OmniLMM-12B [261], Mini-Gemini-7B [141], Emu2-Chat [237], MMAlaya [154], CogVLM-Chat [249], InstructBLIP-7B [56], DeepSeek-VL-7B [155], Idefics-9B-Instruct [137], XComposer2 [62], Yi-VL-6B [7], InternVL-Chat-V1.5 [46], LLAVA-V1.5-7B [148], LLAVA-InternLM2-7b [54], MiniCPM-V2 [257], and Qwen-VL-Chat [18]. In addition, we also evaluate 6 proprietary LVLMs via API: Qwen-VL-Max [18], Claude3-Opus [13], GPT-4V [5], GPT-4o [5], Gemini 1.0 [240], and Gemini 1.5 [211].

### Metrics

To evaluate the model's performance, we use macro-averaged accuracy (ACC) as the evaluation metric for single-choice questions. For multiple-choice questions, we first count the number of correct predictions for each case, then calculate accuracy ($\mathrm{ACC_{mcq}}$) and recall ($\mathrm{Recall_{mcq}}$) based on

---

[6] https://github.com/open-compass/VLMEvalKit
[7] https://github.com/OpenGVLab/Multi-Modality-Arena/tree/main/MedicalEval/
Question-answering_Score

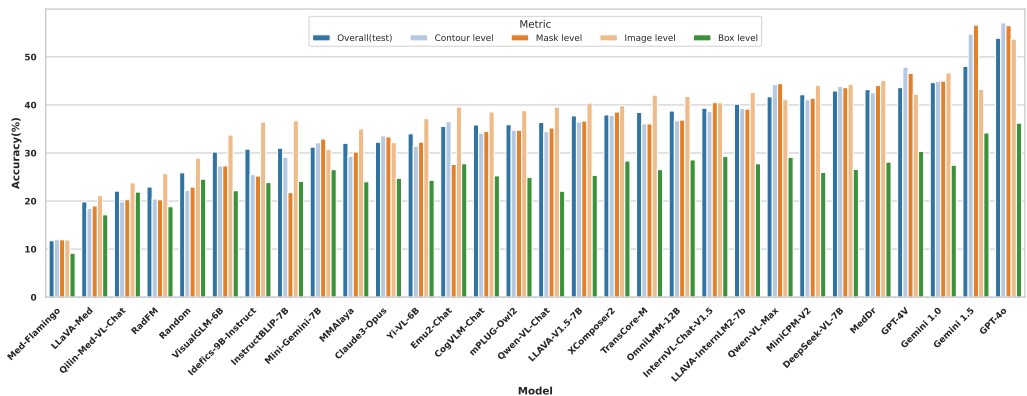

Figure 4: Results for single-choice questions of different models on different perceptual granularities, including Contour level, Mask level, Image level, and Box level.

Table 2: Results for single-choice questions of different LVLMs on clinical VQA tasks. The best-performing model in each category is **in-bold**, and the second best is underlined. Abbreviations: the full terms of all clinical VQA tasks are listed in Table 5 of supplementary material.

| Model name | Overall (val) | Overall (test) | AR | BVR | B | CR | C | DD | IQG | MR | M | NT | OR-A | OR-HN | OR-P | OR-T | SG | SAR | SIR | SWR |
|---|---|---|---|---|---|---|---|---|---|---|---|---|---|---|---|---|---|---|---|---|
| Random | 25.70 | 25.94 | 38.20 | 22.73 | 22.92 | 22.72 | 24.06 | 26.66 | 27.13 | 27.00 | 20.00 | 24.75 | 21.37 | 22.93 | 22.33 | 21.18 | 32.43 | 24.23 | 21.39 | 23.71 |
| Medical Special Model | | | | | | | | | | | | | | | | | | | | |
| Med-Flamingo [181] | 12.74 | 11.64 | 6.67 | 10.14 | 9.23 | 11.27 | 6.62 | 13.43 | 12.15 | 6.38 | 8.00 | 18.18 | 9.26 | 18.27 | 11.00 | 11.53 | 12.16 | 5.19 | 8.47 | 11.43 |
| LLaVA-Med [138] | 20.54 | 19.60 | 24.51 | 17.83 | 17.08 | 19.86 | 15.04 | 19.81 | 20.24 | 21.51 | 13.20 | 15.15 | 20.42 | 23.73 | 17.67 | 19.65 | 21.70 | 19.81 | 14.11 | 20.86 |
| Qilin-Med-VL-Chat [149] | 22.34 | 22.06 | 29.57 | 19.41 | 16.46 | 23.79 | 15.79 | 24.19 | 21.86 | 16.62 | 7.20 | 13.64 | 24.00 | 14.67 | 12.67 | 15.53 | 26.13 | 24.42 | 17.37 | 25.71 |
| RadFM [254] | 22.95 | 22.93 | 27.16 | 20.63 | 13.23 | 19.14 | 20.45 | 24.51 | 23.48 | 22.85 | 15.60 | 16.16 | 14.32 | 24.93 | 17.33 | 21.53 | 29.73 | 17.12 | 19.59 | 31.14 |
| MedDr [95] | 41.95 | 43.69 | 41.20 | 50.70 | 37.85 | 29.87 | 28.27 | 52.53 | 36.03 | 31.45 | 29.60 | 47.47 | 33.37 | 51.33 | 32.67 | 44.47 | 35.14 | 25.19 | 25.58 | 32.29 |
| Open-Source LVLMs | | | | | | | | | | | | | | | | | | | | |
| VisualGLM-6B [61] | 29.58 | 30.45 | 40.16 | 33.92 | 24.92 | 25.22 | 24.21 | 32.99 | 29.96 | 29.53 | 21.20 | 37.88 | 30.32 | 24.80 | 13.33 | 29.88 | 33.11 | 19.62 | 19.16 | 37.43 |
| Idefics-9B-Instruct [137] | 29.74 | 31.13 | 40.39 | 30.59 | 26.46 | 33.63 | 22.56 | 34.38 | 25.51 | 26.71 | 21.60 | 27.78 | 27.47 | 32.80 | 24.67 | 23.41 | 32.66 | 23.08 | 21.39 | 30.57 |
| InstructBLIP-7B [56] | 31.80 | 30.95 | 42.12 | 26.92 | 24.92 | 28.09 | 21.65 | 34.58 | 31.58 | 29.23 | 22.40 | 30.30 | 28.95 | 27.47 | 23.00 | 24.82 | 32.88 | 19.81 | 21.64 | 26.57 |
| Mini-Gemini-7B [141] | 32.17 | 31.09 | 29.69 | 39.16 | 31.85 | 28.26 | 10.38 | 35.58 | 29.96 | 28.78 | 20.80 | 34.34 | 29.58 | 36.53 | 24.00 | 31.76 | 22.45 | 25.96 | 18.56 | 29.43 |
| MMAlaya [154] | 32.19 | 32.30 | 41.20 | 35.14 | 32.15 | 34.17 | 27.82 | 35.09 | 28.34 | 30.27 | 18.00 | 46.97 | 20.21 | 31.20 | 16.00 | 34.59 | 32.28 | 23.65 | 22.93 | 30.29 |
| Yi-VL-6B [7] | 34.82 | 34.31 | 41.66 | 39.16 | 26.62 | 30.23 | 31.88 | 38.01 | 26.72 | 24.93 | 25.20 | 37.37 | 29.58 | 31.20 | 32.33 | 30.59 | 36.71 | 24.81 | 23.18 | 31.43 |
| Qwen-VL-Chat [18] | 35.07 | 36.96 | 38.09 | 40.56 | 38.00 | 32.20 | 25.71 | 44.07 | 24.70 | 30.56 | 24.00 | 40.91 | 29.37 | 36.53 | 26.00 | 27.29 | 35.14 | 16.54 | 20.10 | 34.00 |
| CogVLM-Chat [249] | 35.23 | 36.08 | 40.97 | 30.77 | 27.69 | 32.74 | 19.40 | 41.10 | 36.84 | 34.72 | 24.00 | 40.91 | 36.74 | 37.33 | 26.00 | 33.65 | 36.56 | 20.19 | 23.95 | 26.57 |
| mPLUG-Owl2 [259] | 35.62 | 36.21 | 37.51 | 41.08 | 30.92 | 38.10 | 27.82 | 41.59 | 28.34 | 32.79 | 22.40 | 40.91 | 24.74 | 38.27 | 23.33 | 36.59 | 33.48 | 20.58 | 23.01 | 32.86 |
| Emu2-Chat [237] | 36.50 | 37.59 | 43.27 | 47.73 | 26.31 | 40.07 | 28.12 | 44.00 | 36.44 | 28.49 | 20.40 | 31.82 | 26.74 | 37.60 | 26.67 | 29.76 | 33.63 | 23.27 | 26.43 | 29.43 |
| OmniLMM-12B [261] | 37.89 | 39.30 | 39.82 | 40.56 | 32.62 | 37.57 | 24.81 | 46.68 | 35.63 | 35.01 | 27.60 | 57.58 | 28.42 | 34.00 | 25.00 | 29.18 | 34.46 | 24.42 | 27.54 | 40.29 |
| LLAVA-V1.5-7B [148] | 38.23 | 37.96 | 45.45 | 34.27 | 30.92 | 41.32 | 21.65 | 44.68 | 34.01 | 27.74 | 23.60 | 43.43 | 28.00 | 42.13 | 29.00 | 35.06 | 33.41 | 22.12 | 23.61 | 29.14 |
| XComposer2 [62] | 38.68 | 39.20 | 41.89 | 37.59 | 33.69 | 40.79 | 22.26 | 45.87 | 36.44 | 32.94 | 27.20 | 58.59 | 26.11 | 36.40 | 43.67 | 37.29 | 32.06 | 23.46 | 27.80 | 32.86 |
| TransCore-M [3] | 38.86 | 38.70 | 40.74 | 41.78 | 20.77 | 35.06 | 34.74 | 45.69 | 32.39 | 32.94 | 24.40 | 44.95 | 31.05 | 38.93 | 27.00 | 33.76 | 33.86 | 23.46 | 25.49 | 31.14 |
| InternVL-Chat-V1.5 [46] | 38.86 | 39.73 | 43.84 | 44.58 | 34.00 | 33.99 | 31.28 | 45.59 | 33.20 | 38.28 | 32.40 | 42.42 | 31.89 | 42.80 | 27.00 | 36.82 | 34.76 | 23.27 | 24.72 | 32.57 |
| LLAVA-InternLM2-7b [54] | 40.07 | 40.45 | 39.82 | 37.94 | 30.62 | 35.24 | 29.77 | 48.97 | 34.01 | 25.96 | 20.80 | 53.03 | 30.95 | 42.67 | 32.00 | 39.88 | 32.43 | 21.73 | 24.38 | 38.00 |
| DeepSeek-VL-7B [155] | 41.73 | 43.43 | 38.43 | 47.03 | 42.31 | 37.03 | 26.47 | 51.11 | 33.20 | 31.16 | 26.00 | 44.95 | 36.00 | 58.13 | 36.33 | 47.29 | 34.91 | 18.08 | 25.49 | _39.43_ |
| MiniCPM-V2 [257] | 41.79 | 42.54 | 40.74 | 43.01 | 36.46 | 37.57 | 27.82 | 51.08 | 28.74 | 29.08 | 26.80 | 47.47 | 37.05 | 46.40 | 25.33 | 46.59 | 35.89 | 22.31 | 23.44 | 31.71 |
| Proprietary LVLMs | | | | | | | | | | | | | | | | | | | | |
| Claude3-Opus [13] | 32.37 | 32.44 | 1.61 | 39.51 | 34.31 | 31.66 | 12.63 | 39.26 | 28.74 | 30.86 | 22.40 | 37.37 | 25.79 | 41.07 | 29.33 | 33.18 | 31.31 | 21.35 | 23.87 | 4.00 |
| Qwen-VL-Max [18] | 41.34 | 42.16 | 32.68 | 44.58 | 31.38 | 40.79 | 10.68 | 50.53 | 32.79 | 44.36 | 29.20 | 51.52 | 41.37 | 58.00 | 30.67 | 41.65 | 26.95 | 25.00 | 24.64 | 39.14 |
| GPT-4V [5] | 42.50 | 44.08 | 29.92 | 48.95 | 44.00 | 37.39 | 12.93 | 52.88 | 32.79 | 44.21 | _32.80_ | 63.64 | 39.89 | 54.13 | 37.00 | 50.59 | 27.55 | 23.08 | 25.75 | 37.43 |
| Gemini 1.0 [240] | 44.38 | 44.93 | _42.12_ | 45.10 | 46.46 | 37.57 | 20.45 | 53.29 | 35.22 | 36.94 | 25.20 | 51.01 | 34.74 | 59.60 | 34.00 | 50.00 | **36.64** | 23.65 | 23.87 | 35.43 |
| Gemini 1.5 [211] | _47.42_ | _48.36_ | **43.50** | _56.12_ | _51.23_ | _47.58_ | 2.26 | _55.33_ | _38.87_ | _48.07_ | 30.00 | **76.26** | _51.05_ | **75.87** | _46.33_ | _62.24_ | 20.57 | **27.69** | **30.54** | 40.57 |
| GPT-4o [5] | **53.53** | **53.96** | 38.32 | **61.01** | **57.08** | **49.02** | **46.62** | **61.45** | **46.56** | **56.38** | **34.00** | _75.25_ | **53.79** | _69.47_ | **48.67** | **65.88** | 33.93 | 22.88 | 29.51 | _39.43_ |

the proportion of correct matches to the prediction length and the length of the ground-truth options, respectively. More details are shown in supplementary materials. If a model's output does not include clearly followed instructions to select an answer or letter options, we use ChatGPT-3.5-turbo-0613 to extract the answer. If an answer cannot be extracted, it is treated as an error.

## Results

### Analysis

After reviewing the evaluation results, we have drawn **2 conclusions** and identified **5 insufficiencies** that require further improvement in future LVLMs in the medical domain:

**Conclusion 1. Medical tasks are still challenging for all LVLMs:** Our GMAI-MMBench provides a comprehensive multitask challenge, revealing that even the most advanced model, GPT-4o, is limited to an accuracy of around 54% (see Table 2 and Table 3). This does not meet the clinical requirement and indicates that all current LVLMs in the medical domain still require significant improvement.

**Conclusion 2. Open-source models are catching up to the commercialized models:** In the comparison between open-source and commercialized models, most open-source models lag behind their commercialized counterparts. Leading open-source models such as MedDr and DeepSeek-

Table 3: Results for single-choice questions of different LVLMs on departments. The best-performing model in each category is **in-bold**, and the second best is underlined. Abbreviations: the full terms of all departments are listed in Table 6 of supplementary material

| Model name | Overall (val) | Overall (test) | CS | D | E | GH | GS | H | ID | LMP | NH | N | OG | OM | O | OS | ENT/HNS | PM | SM | U |
|---|---|---|---|---|---|---|---|---|---|---|---|---|---|---|---|---|---|---|---|---|
| Random | 25.70 | 25.94 | 22.82 | 25.19 | 21.00 | 25.97 | 22.24 | 24.45 | 31.13 | 28.99 | 22.86 | 24.00 | 29.15 | 27.77 | 30.36 | 25.92 | 22.53 | 24.74 | 22.87 | 29.19 |
| Medical Special Model | | | | | | | | | | | | | | | | | | | | |
| Med-Flamingo [181] | 12.74 | 11.64 | 11.76 | 12.49 | 10.00 | 10.88 | 9.33 | 5.42 | 7.28 | 10.05 | 12.00 | 10.91 | 12.88 | 14.89 | 15.37 | 12.40 | 13.43 | 12.89 | 14.92 | 10.47 |
| LLaVA-Med [138] | 20.54 | 19.60 | 26.12 | 20.20 | 29.00 | 20.31 | 16.30 | 18.46 | 15.23 | 21.84 | 20.86 | 16.73 | 21.69 | 19.23 | 20.18 | 18.38 | 20.99 | 16.87 | 20.49 | 21.55 |
| Qilin-Med-VL-Chat [149] | 22.34 | 22.06 | 12.94 | 21.06 | 15.50 | 22.09 | 18.98 | 17.33 | 17.88 | 22.92 | 31.14 | 29.82 | 20.00 | 21.83 | 25.55 | 19.07 | 14.81 | 29.42 | 22.17 | 22.29 |
| RadFM [254] | 22.95 | 22.93 | 24.24 | 23.02 | 20.00 | 20.59 | 20.83 | 19.49 | 28.48 | 24.42 | 18.00 | 32.00 | 16.95 | 26.90 | 26.25 | 18.26 | 26.54 | 25.19 | 23.74 | 20.20 |
| MedDr [95] | 41.95 | 43.69 | 53.18 | 45.28 | 33.00 | 44.78 | 28.03 | 29.91 | 47.68 | 35.22 | 38.29 | 78.55 | 25.08 | 49.53 | 45.31 | 52.09 | 48.61 | 52.36 | 54.21 | 39.90 |
| Open-Source LVLMs | | | | | | | | | | | | | | | | | | | | |
| VisualGLM-6B [61] | 29.58 | 30.45 | 52.71 | 25.95 | 14.00 | 31.69 | 22.06 | 25.17 | 30.46 | 25.50 | 30.29 | 59.27 | 15.93 | 29.97 | 37.79 | 30.09 | 23.61 | 32.85 | 38.19 | 23.03 |
| Idefics-9B-Instruct [137] | 29.74 | 31.13 | 19.76 | 33.98 | 21.00 | 30.08 | 24.46 | 26.66 | 50.33 | 28.74 | 36.00 | 58.55 | 36.27 | 29.64 | 36.76 | 36.07 | 24.38 | 31.36 | 32.04 | 29.19 |
| InstructBLIP-7B [56] | 31.80 | 30.95 | 27.06 | 28.99 | 17.50 | 34.24 | 21.78 | 25.84 | 43.05 | 29.15 | 19.14 | 53.09 | 27.46 | 28.64 | 31.99 | 34.58 | 30.25 | 30.76 | 41.09 | 31.28 |
| Mini-Gemini-7B [141] | 32.17 | 31.09 | 34.59 | 39.63 | 23.50 | 35.74 | 23.46 | 19.80 | 41.06 | 25.91 | 40.86 | 56.00 | 19.32 | 21.63 | 35.73 | 35.83 | 33.95 | 40.57 | 29.14 | 29.56 |
| MMAlaya [154] | 32.19 | 32.30 | 71.06 | 37.68 | 38.00 | 28.30 | 27.40 | 27.64 | 51.66 | 32.39 | 28.86 | 83.64 | 29.49 | 27.37 | 35.92 | 36.70 | 20.99 | 27.53 | 29.43 | 28.08 |
| Yi-VL-6B [7] | 34.82 | 34.31 | 39.76 | 43.76 | 56.00 | 27.30 | 25.91 | 27.23 | 45.70 | 32.56 | 44.29 | 65.45 | 47.46 | 36.38 | 39.00 | 35.39 | 25.46 | 29.77 | 39.06 | 35.22 |
| Qwen-VL-Chat [18] | 35.07 | 36.96 | 36.47 | 39.63 | 36.50 | 27.08 | 20.79 | 27.64 | 60.93 | 30.23 | 52.57 | 70.55 | 37.29 | 47.13 | 39.37 | 46.67 | 34.57 | 37.63 | 47.88 | 39.90 |
| CogVLM-Chat [249] | 35.23 | 36.08 | 30.59 | 38.98 | 42.50 | 31.41 | 26.22 | 23.62 | 47.02 | 34.22 | 51.43 | 56.00 | 32.54 | 44.13 | 38.67 | 37.94 | 30.86 | 41.11 | 45.91 | 29.19 |
| mPLUG-Owl2 [259] | 35.62 | 36.21 | 47.76 | 40.50 | 41.00 | 33.46 | 27.22 | 28.16 | 51.66 | 33.14 | 38.86 | 68.73 | 16.27 | 38.58 | 43.34 | 35.70 | 27.78 | 41.61 | 39.76 | 30.91 |
| Emu2-Chat [237] | 36.50 | 37.59 | 27.53 | 35.83 | 27.50 | 34.41 | 28.49 | 29.35 | 60.26 | 36.63 | 34.00 | 64.73 | 28.81 | 44.79 | 43.20 | 37.69 | 37.50 | 41.86 | 43.18 | 35.34 |
| OmniLMM-12B [261] | 37.89 | 39.30 | 39.53 | 37.46 | 41.50 | 36.18 | 27.36 | 28.00 | 60.93 | 37.46 | 55.43 | 80.00 | 31.19 | 35.71 | 44.89 | 42.49 | 28.24 | 43.80 | 51.19 | 42.86 |
| LLAVA-V1.5-7B [148] | 38.23 | 37.96 | 42.35 | 37.57 | 44.50 | 36.13 | 27.99 | 24.91 | 49.01 | 31.31 | 34.00 | 68.36 | 27.12 | 45.39 | 42.46 | 42.80 | 33.80 | 44.20 | 41.21 | 38.92 |
| XComposer2 [62] | 38.68 | 39.20 | 32.71 | 42.13 | 70.50 | 33.13 | 29.62 | 27.02 | 54.30 | 34.05 | 23.14 | 83.64 | 39.66 | 46.53 | 44.23 | 45.73 | 28.86 | 45.55 | 41.32 | 41.87 |
| TransCore-M [3] | 38.86 | 38.70 | 39.06 | 43.87 | 24.50 | 40.18 | 29.08 | 30.79 | 52.98 | 32.48 | 38.86 | 66.91 | 42.37 | 42.79 | 44.75 | 40.44 | 36.73 | 34.00 | 47.19 | 35.71 |
| InternVL-Chat-V1.5 [46] | 38.86 | 39.73 | 36.47 | 44.84 | 53.50 | 37.07 | 26.63 | 31.61 | 60.26 | 34.14 | 36.00 | 67.27 | 37.63 | 55.21 | 47.13 | 38.69 | 41.98 | 39.17 | 37.55 | 41.26 |
| LLAVA-InternLM2-7b [54] | 40.07 | 40.45 | 43.53 | 40.72 | 60.50 | 34.74 | 30.12 | 27.44 | 51.66 | 33.39 | 50.86 | 74.55 | 26.44 | 49.13 | 42.74 | 43.12 | 31.94 | 50.87 | 47.01 | 39.04 |
| DeepSeek-VL-7B [155] | 41.73 | 43.43 | 60.00 | 43.97 | 47.50 | 45.12 | 28.22 | 31.20 | 46.36 | 32.97 | 52.29 | 67.64 | 42.36 | 49.27 | 44.23 | 49.97 | 52.78 | 45.00 | 53.63 | 38.79 |
| MiniCPM-V2 [257] | 41.79 | 42.54 | 37.88 | 43.65 | 35.50 | 42.67 | 26.49 | 29.24 | 37.75 | 33.31 | 59.71 | 67.27 | 38.64 | 50.87 | 42.64 | 50.59 | 40.90 | 51.07 | 57.81 | 35.10 |
| Proprietary LVLMs | | | | | | | | | | | | | | | | | | | | |
| Claude3-Opus [13] | 32.37 | 32.44 | 38.59 | 34.42 | 43.50 | 27.97 | 22.96 | 23.62 | 52.32 | 25.42 | 25.14 | 66.91 | 15.93 | 35.25 | 41.06 | 36.07 | 37.50 | 40.67 | 35.40 | 34.24 |
| Qwen-VL-Max [18] | 41.34 | 42.16 | 50.59 | 47.23 | 74.00 | 40.68 | 29.03 | 26.71 | 58.94 | 34.05 | 62.29 | 85.45 | 27.80 | 44.39 | 43.90 | 42.99 | 48.61 | 49.38 | 51.13 | 40.52 |
| GPT-4V [5] | 42.50 | 44.08 | 64.00 | 44.95 | 58.50 | 42.45 | 30.03 | 29.40 | 58.28 | 32.31 | 54.57 | 83.27 | 37.63 | 48.26 | 49.04 | 48.41 | 44.60 | 51.87 | 53.98 | 40.89 |
| Gemini 1.0 [240] | 44.38 | 44.93 | 57.41 | 46.25 | 57.50 | 36.40 | 28.67 | 27.80 | 45.03 | 38.21 | 58.57 | 86.55 | 40.68 | 51.74 | 47.45 | 55.64 | 50.46 | 47.83 | 61.58 | 41.87 |
| Gemini 1.5 [211] | 47.42 | 48.36 | 55.29 | 50.81 | 54.00 | 51.05 | 36.59 | 29.86 | 56.95 | 36.88 | 58.00 | 88.00 | 47.46 | 48.13 | 51.19 | 56.88 | 64.51 | 56.50 | 59.78 | 31.65 |
| GPT-4o [5] | 53.53 | 53.96 | 66.82 | 48.53 | 64.50 | 55.94 | 35.10 | 48.53 | 74.17 | 43.52 | 64.57 | 91.64 | 37.63 | 57.88 | 55.21 | 62.80 | 66.98 | 58.39 | 64.60 | 46.18 |

VL-7B, although not as accurate as GPT-4o, have surpassed Claude3 Opus and Qwen-VL-Max, approaching the performance of GPT-4V. This suggests that open-source models in the medical field are gradually catching up to the top-performing commercialized models.

**Insufficiency 1. Performance on different clinical VQA tasks needs improvement:** Table 2 shows that the best-performing clinical VQA tasks are Disease Diagnosis (DD) and Nervous Tissue (NT), with models exceeding the random baseline by an average of over 10%. However, in clinical VQA tasks such as Severity Grading (SG) and Attribute Recognition (AR), most LVLMs face challenges, and most of them perform worse than the random baseline. Overall, despite the advanced models like GPT-4o and Gemini 1.5 significantly outperforming the random baseline, there remains a substantial gap between their performance and the requirements of real-world applications, indicating that all the models still need more specialized medical knowledge for training.

**Insufficiency 2. The performance across different departments needs further balancing:** In examining performance across different medical departments, as shown in Table 3, we found that the Infectious Diseases (ID) and Neurosurgery (N) departments performed the best. In contrast, departments such as General Surgery (GS) and Obstetrics and Gynecology (OG) showed a need for improvement, as the performance of all models in these areas did not significantly exceed the random baseline compared to other departments. This indicates that current large models exhibit specialization biases, suggesting that future development of LVLMs aiming to achieve general medical AI should focus on balancing capabilities across all departments.

**Insufficiency 3. The LVLMs are not robust among different perceptual types:** As shown in Figure 4, models perform slightly better with contour-level perception compared to mask-level perception, and both outperform image-level perception (without annotation) significantly. However, bounding box-level perception shows the worst performance among all perceptual types, indicating that models are sensitive to this perceptual type. This evaluation underscores the need for LVLMs to address robustness issues across different perceptual types, which is crucial for their effectiveness in interactive applications.

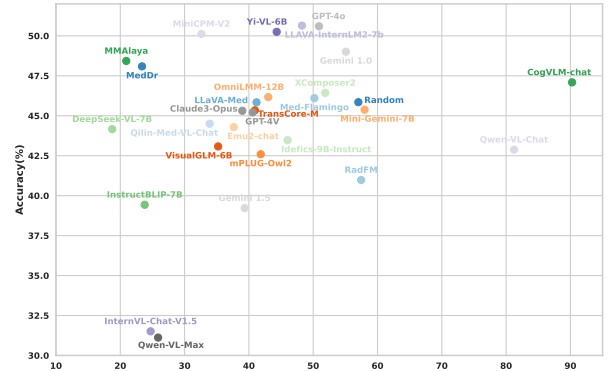

Figure 5: Overall results for multiple-choice questions of different models.

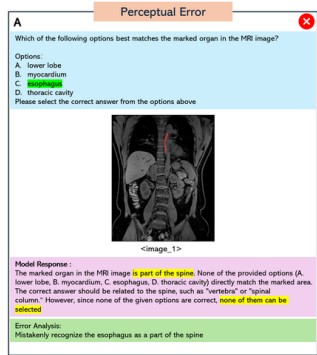 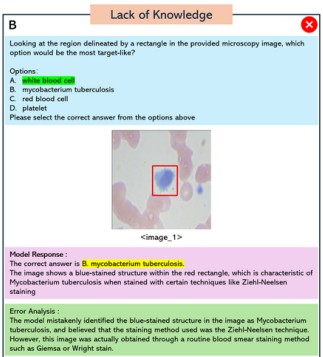 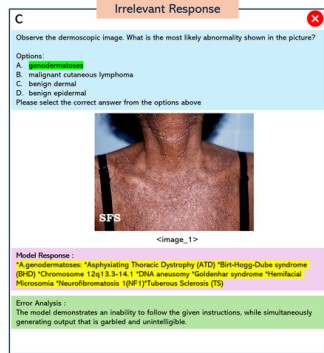

Figure 6: Three examples of error cases. **A:** Question misunderstanding. **B:** Perceptual Errors. **C:** Lack of Knowledge. More studies can be found in the appendix.

**Insufficiency 4. Medical-specific models need to enhance their instruction tuning:** Interestingly, medical-specific models significantly underperform compared to general models, despite being trained and fine-tuned directly on relevant medical data. Specifically, LLaVA-Med is fine-tuned from the LLaVA model series in the medical field, but its performance is even worse than LLAVA-V1.5-7B. The primary reason for the poor performance of these medical-specific models is their inability to follow instructions correctly and their failure to understand or answer medical-related questions accurately. Detailed analysis can be found in the case study and supplementary materials sections on medical model analysis. Among these, the best-performing medical-specific model is MedDr, which is fine-tuned from the InternVL series and successfully surpasses the InternVL-Chat-V1.5. Unlike other medical-specific models that derive instruction-tuning data from papers, online sources, and books, MedDr builds its dataset based on high-quality medical image classification datasets. This result suggests that the quality of currently available medical instruction tuning datasets on the internet needs improvement and highlights the effectiveness of MedDr's dataset construction strategy, serving as a valuable reference for future medical-specific models.

**Insufficiency 5. The performance of most LVLMs on multiple-choice questions needs improvement:** Based on our tests, none of the models can totally match the correct answers (they always miss or over-select), so we adopt a relatively loose evaluation method for multiple-choice questions: using multi-choice hit rate ($\mathrm{ACC}_{\mathrm{mcq}}$) and recall rate ($\mathrm{Recall}_{\mathrm{mcq}}$). The experimental results are shown in Figure 5. Using this method, we found that most models have an accuracy rate of around 40%-50% and a recall rate of around 40%-60%. Surprisingly, InternVL-Chat-V1.5 and Qwen-VL-Max performed well in single-choice questions but showed very poor recall and accuracy rates in multiple-choice questions. In contrast, Qwen-VL-Chat and CogVLM-Chat, which performed relatively poorly in single-choice questions, achieved very high recall rates and moderate accuracy rates in multiple-choice questions, especially CogVLM-Chat with over 90% recall rate. Nonetheless, even with this less strict evaluation method, all models had accuracy rates below 55%, indicating that there is still significant room for improvement in answering multiple-choice questions.

**Case Study**

We further analyze the results by requiring the models to output content beyond the provided options and explain their reasoning process. This approach helps us better understand the causes of errors. Through detailed testing and analysis, we identify 5 typical errors present in the LVLMs:

**Question misunderstanding:** This occurs when the model incorrectly understands the purpose of the question, leading to an inability to provide a correct response. As shown in Figure 6A, the model is asked to answer a multiple-choice question, but it describes the problem or repeats the options rather than choosing an option.

**Perceptual Error:** These errors occur when there is a mislocation or misrecognition of image content. This means that the model's understanding or interpretation of the visual content is incorrect, leading to an inaccurate response. As shown in Figure 6B, the model mistakenly identifies the esophagus as the spine, suggesting that while the model can locate the target on the image (The annotated esophagus is very close to the spine), it makes an error in perceiving the masked content.

**Lack of knowledge:** While the model can recognize text and images, it makes errors in specific areas that require specific knowledge, indicating a deficiency in relevant training or fine-tuning in those areas. For example, in Figure 6C, the model incorrectly identifies the staining method as Ziehl-Neelsen and misrecognizes the blue-stained structure as Mycobacterium tuberculosis, where it is actually a white blood cell stained with Giemsa or Wright stain. This error indicates the model's lack of knowledge in experimental medicine.

**Irrelevant Responses:** This error indicates the model fails to generate a readable answer, which is easily found in medical-specific models like RadFM. Examples are listed in the appendix.

**Reject to Answer:** Some models, especially proprietary LVLMs like GPT-4V, GPT-4o, Gemini 1.0, and Gemini 1.5, commonly refuse to provide an answer due to policy reasons, because safety is crucial according to the commercial rules and regulations. Many potentially risky responses are declined to ensure compliance with guidelines. Those models' strict adherence to safety protocols and ethical standards limits response capabilities in certain domains.

## Conclusion

The development of GMAI-MMBench as a benchmark for evaluating LVLMs' capabilities represents a significant advancement in the pursuit of general medical AI. GMAI-MMBench epitomizes the expertise of skilled medical professionals, serving as a pivotal guide for advancing large models toward GMAI by testing the limits of current LVLMs. Owing to the extensive and diverse source of GMAI-MMBench, which comprises medical datasets annotated by professional healthcare providers worldwide, this benchmark can comprehensively evaluate the model's capability across various specific aspects. In this way, GMAI-MMBench can guide the model development at a more fine-grained level, accelerating the development of robust and reliable GMAI systems. Moreover, this benchmark supports the advancement of interactive multimodal medical models by providing more perceptual modes and annotations that are commonly used by physicians in clinical practice, thereby creating a framework for their evaluation and improvement.

However, GMAI-MMBench, like all benchmarks, has its limitations. The manual curation process, despite being thorough, might introduce biases, and focusing solely on medical subjects may not fully meet the criteria for general medical AI as defined. Nevertheless, we assert that high performance on GMAI-MMBench is essential for demonstrating the extensive subject knowledge and expert-level reasoning skills required for general medical AI. Looking ahead, we intend to integrate human evaluations into GMAI-MMBench. This addition will offer a more grounded comparison between model capabilities and expert performance, providing insights into how close current AI systems are to achieving general medical AI in the medical field.

## Acknowledgement

This work was supported by the National Key Research and Development Program of China (Grant No. 2022ZD0160101). Additionally, the authors acknowledge the funding provided by the National Natural Science Foundation of China (Grant No. 62272450), which has been instrumental in facilitating the research and development efforts presented in this study.

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

# GMAI-MMBench: A Comprehensive Multimodal Evaluation Benchmark Towards General Medical AI

## Supplementary Materials

## Contents

## A   Related work

### A.1   Large Vision-Language Model(LVLMs)

In contrast to traditional deep learning models, Large Vision-Language Models (LVLMs) offer a broader spectrum of possibilities for AI-assisted healthcare. Their user-friendly and intuitive interaction mechanisms make them one of the most promising paradigms for future AI applications. Among the multitude of LVLMs, prominent proprietary models such as GPT-4o [5], Claude3-opus [13], and Qwen-max [18] exemplify the pinnacle of contemporary general-purpose large models. Additionally, numerous open-source general-purpose models have emerged, including the InternVL series [47, 46], LLAVA series [147, 148, 43], DeepSeek series [155], CogVLM series [249], InstructBLIP series [56], Idefics series [137], XComposer series [43, 266, 62, 63], Yi-VL series [7], Xtuner series [54], and MiniCPM series [103, 257]. These open-source models are rapidly evolving due to their accessibility and collaborative development.

To address specialized medical tasks, researchers have trained and fine-tuned these large models using domain-specific medical data, resulting in specialized large models. Noteworthy examples include LLaVA-Med [138] derived from the LLAVA series, and MedDr [95] based on the InternLM framework. The advent of these specialized medical models has laid a solid foundation for the application of LVLMs in the healthcare sector, highlighting their transformative potential and accelerating their development within the medical domain.

## A.2 Benchmarks

In the swiftly emerging and burgeoning domain of LVLMs, the significance of rigorous evaluation cannot be overstated. Benchmarking serves as a crucial metric for guiding model enhancement, identifying deficiencies, and steering the trajectory of model development. Within the medical domain, benchmarks are typically categorized into specialized and general-purpose benchmarks.

Specialized benchmarks are often concentrated on a particular modality or medical discipline. For instance, VQA-RAD [136], SLAKE [145], and RadBench [253] focus on radiology, while PathVQA [96] and PathMMU [238] are dedicated to pathology. These benchmarks provide a wealth of evaluation data for specific modalities or disciplines, enabling comprehensive assessment of capabilities within targeted fields. However, their limited generalizability constrains their broader applicability.

In addition to these specialized benchmarks, there exist general-purpose medical benchmarks that span multiple medical domains. Prominent examples include MMMU [263], OminimedVQA [106], and MMT-Bench [260]. These comprehensive benchmarks facilitate a more holistic evaluation of a model's overall competence in the medical field. Nonetheless, these general-purpose benchmarks often exhibit shortcomings in various aspects such as the volume of tasks, number of modalities, data distribution, and granularity of data. Addressing these limitations presents a significant challenge that necessitates prompt resolution.

The development and refinement of benchmarks are indispensable for the progress of LVLMs in healthcare. By elucidating the capabilities and limitations of specialized and general-purpose benchmarks, it becomes evident that while specialized benchmarks excel in evaluating domain-specific performance, their lack of generalizability is a notable drawback. Conversely, general-purpose benchmarks offer a broader assessment across multiple medical fields but often fall short in task diversity, modality coverage, and data granularity. Therefore, there is an urgent need for more comprehensive and robust benchmarks to bridge these gaps and better support the advancement of LVLMs in healthcare.

## B  Dataset Details

In this section, we provide the detailed datasets used in GMAI-MMBench, including the name of the dataset or challenge, the number of sub-datasets in it, the modality, the dimension of data, the task type, and the number of cases. As shown in Table 4, GMAI-MMBench is constructed from 284 datasets across 38 medical image modalities. These datasets are derived from the public (268) and several hospitals (16) that have agreed to share their ethically approved data.

Table 4: Detailed datasets information in GMAI-MMBench. As one challenge/dataset may contain several sub-tasks or sub-challenges in the medical area, we count them in the "N" (second column). In the dimension (Dim) column, 2D and 3D denote the dimensions of the original data, respectively. In the task type (Task) column, Cls, MCls, Seg, and Det indicate classification, multi-label classification, segmentation, and detection, respectively. The count represents the number of cases used in GMAI-MMBench.

| Challenge / Dataset | N | Modality | Dim | Task | Count |
|---|---|---|---|---|---|
| 5K+ CT Images on Fractured Limbs [215] | 1 | CT | 2D | Cls | 60 |
| AAPM RT-MAC 2019 [40] | 1 | T2 weighted MRI | 3D | Seg | 68 |
| Abdomenatlas 1.0 [205] | 1 | CT | 3D | Seg | 52 |
| AbdomenCT-1K [164] | 1 | CT | 3D | Seg | 28 |
| ACDC 2017 [30] | 1 | MRI | 3D | Seg | 10 |
| ACRIMA [60] | 1 | Fundus Photography | 2D | Cls | 1 |
| ADAM 2020 [68] | 1 | Fundus Photography | 2D | Cls | 1 |
| Adrenal-ACC-Ki67-Seg [177] | 1 | CT | 3D | Seg | 60 |
| AGE 2019 [74] | 1 | OCT | 2D | MCls | 20 |
| AIDA-E 2016 | 3 | Endoscopy | 2D | Cls | 187 |
| AIIB23 [183] | 1 | CT | 3D | Seg | 34 |
| AIROGS [58] | 1 | Fundus Photography | 2D | Cls | 57 |
| AMOS 2022 [116] | 1 | MRI, CT | 3D | Seg | 148 |
| APTOS 2019 [125] | 1 | Fundus Photography | 2D | Cls | 14 |

| | | | | | |
|---|---|---|---|---|---|
| ATLAS 2023 [206] | 1 | T1 weighted MRI | 3D | Seg | 16 |
| ATM 2022 [265] | 1 | CT | 3D | Seg | 26 |
| AtriaSeg 2018 [265] | 1 | LGE MRI | 3D | Seg | 2 |
| Augemnted ocular diseases | 1 | Fundus Photography | 2D | Cls | 97 |
| AV Nicking Quantification [186] | 1 | Fundus Photography | 2D | Cls | 71 |
| Bacteria Detection with Darkfield Microscopy [201] | 1 | Microscopy | 2D | Seg | 120 |
| BCNB [256] | 9 | Histopathology | 2D | Cls | 360 |
| BCSS [12] | 1 | Histopathology | 2D | Seg | 102 |
| BioMediTech [184] | 1 | Microscopy | 2D | Cls | 120 |
| Blood Cell Images [180] | 1 | Microscopy | 2D | Cls | 55 |
| BloodCell from Heywhale | 1 | Microscopy | 2D | Det | 90 |
| Bone-Marrow-Cytomorphology [172] | 1 | Histopathology | 2D | Cls | 484 |
| Brain-Tumor-Progression [221] | 1 | T2 weighted MRI, T1 weighted MRI, FLAIR MRI, ADC MRI | 3D | Seg | 60 |
| BraTS 2020 [33, 22, 23] | 1 | FLAIR MRI | 3D | Seg | 4 |
| BraTS 2021 [22, 23, 20] | 1 | FLAIR MRI | 3D | Seg | 2 |
| BraTS-TCGA-GBM [216] | 1 | T1 MRI | 3D | Seg | 4 |
| BraTS-TCGA-LGG [21] | 1 | T2 MRI, FLAIR MRI, T1Gd MRI | 3D | Seg | 16 |
| BreakHis [232] | 4 | Histopathology | 2D | Cls | 60 |
| Breast Cancer Cell Seg [79] | 1 | Histopathology | 2D | Seg | 18 |
| BRIGHT [111, 193] | 1 | Histopathology | 2D | Cls | 117 |
| BTCV-Abdomen [135] | 1 | CT | 3D | Seg | 60 |
| BTCV-Cervix [135] | 1 | CT | 3D | Seg | 96 |
| BUSI [8] | 1 | UltraSound | 2D | Seg | 60 |
| C-NMC 2019 [182] | 1 | Histopathology | 2D | Cls | 28 |
| CAD-PE [83] | 1 | CT | 3D | Seg | 46 |
| cataract dataset [121] | 1 | Fundus Photography | 2D | Cls | 34 |
| Cervix93 Cytology Dataset [198] | 1 | Microscopy | 2D | Cls | 60 |
| CETUS 2014 | 1 | UltraSound | 3D | Seg | 2 |
| CHAOS [127, 128] | 1 | T2 weighted MRI, T1 weighted MRI | 3D | Seg | 14 |
| Chest CT-Scan images Dataset [90] | 1 | CT | 2D | Cls | 81 |
| Chest X-Ray Images with Pneumothorax Masks [264] | 1 | X-ray | 2D | Seg | 30 |
| ChestX-Det [143] | 1 | X-ray | 2D | Seg | 674 |
| ChestX-Det [143] | 1 | X-ray | 2D | Det | 339 |
| Chiu_BOE_2013_dataset [49] | 1 | Adaptive Optics Ophthalmoscopy | 2D | Seg | 52 |
| CMRxMotion 2022 [248] | 1 | CMR | 3D | Seg | 12 |
| Colorectal-Liver-Metastases [228] | 1 | CT | 3D | Seg | 10 |
| Continuous Registration | 1 | CT | 3D | Seg | 6 |
| Corneal Nerve [218] | 1 | Microscopy | 2D | Cls | 35 |
| Corneal Nerve Tortuosity Grading [219] | 1 | Microscopy | 2D | Cls | 30 |
| CoronaHack [52] | 1 | X-ray | 2D | Cls | 8 |
| COVID-19 CT scans [192, 81, 122] | 1 | CT | 3D | Seg | 74 |
| Covid-19 Image Dataset [209] | 1 | X-ray | 2D | Cls | 5 |
| COVID-19 Radiography Database [51] | 1 | X-ray | 2D | Cls | 40 |
| COVID-19-20 [214] | 1 | CT | 3D | Seg | 30 |
| COVID-19-CT-Seg [192] | 1 | CT | 3D | Seg | 30 |
| COVID19 with Pneumonia and Normal Chest Xray(PA) Dataset [16] | 1 | X-ray | 2D | Cls | 21 |
| COVIDGR [239] | 1 | X-ray | 2D | Cls | 1 |
| COVIDx CXR-4 [247] | 2 | X-ray | 2D | Cls | 59 |
| CRAG [84] | 1 | Histopathology | 2D | Seg | 16 |
| CRASS12 [101] | 1 | X-ray | 2D | Seg | 60 |
| CRC100K [126] | 1 | Histopathology | 2D | Cls | 210 |
| CT-ICH [102] | 1 | CT | 2D | Seg | 60 |
| CT-ORG [212] | 1 | CT | 3D | Seg | 40 |
| CTPelvic1K [150] | 1 | CT | 3D | Seg | 168 |

| Dataset | | Modality | Dim | Task | Count |
|---|---|---|---|---|---|
| CTSpine1K [59] | 1 | CT | 3D | Seg | 40 |
| Curious 2022 [255] | 1 | UltraSound | 3D | Seg | 60 |
| CVC-ClinicDB [28] | 1 | Endoscopy | 2D | Seg | 10 |
| DDTI [195] | 1 | UltraSound | 2D | Seg | 60 |
| DeepDRiD [152] | 3 | Fundus Photography | 2D | Cls | 73 |
| derm7pt [129] | 1 | Dermoscopy | 2D | Cls | 5 |
| Diabetic Retinopathy Arranged [185] | 1 | Fundus Photography | 2D | Cls | 60 |
| Diabetic Retinopathy Detection [65] | 1 | Fundus Photography | 2D | Cls | 52 |
| Diagnosis of Diabetic Retinopathy [57] | 1 | Fundus Photography | 2D | Cls | 42 |
| DigestPath 2019 [55] | 1 | Histopathology | 2D | Seg | 60 |
| DigestPath 2020 [55] | 1 | Histopathology | 2D | Cls | 60 |
| DRAC 2022 [204] | 1 | Fundus Photography | 2D | Seg | 58 |
| DRIMDB [225] | 1 | Fundus Photography | 2D | Cls | 37 |
| DRIVE [233] | 1 | Fundus Photography | 2D | Seg | 14 |
| EAD 2020 [9] | 1 | Endoscopy | 2D | Det | 210 |
| EDD 2020 [9] | 2 | Endoscopy | 2D | Seg | 198 |
| EDD 2020 [9] | 1 | Endoscopy | 2D | Det | 120 |
| EMIDEC 2020 [134] | 1 | MRI | 3D | Seg | 62 |
| EndoVis 2015 [29] | 1 | Endoscopy | 2D | Seg | 10 |
| EndoVis 2017 KBD [11] | 1 | Endoscopy | 2D | Seg | 16 |
| EndoVis 2018 RSS [10] | 1 | Endoscopy | 2D | Seg | 370 |
| EndoVisSub-Instrument | 1 | Endoscopy | 2D | Seg | 86 |
| Eye OCT Datasets [167] | 1 | OCT | 2D | Cls | 14 |
| Finding and Measuring Lungs in CT Data [166] | 1 | CT | 2D | Seg | 60 |
| Finding and Measuring Lungs in CT Data [166] | 1 | CT | 3D | Seg | 8 |
| Fitzpatrick17k [85] | 1 | Dermoscopy | 2D | Cls | 270 |
| FLARE 2021 [162] | 1 | CT | 3D | Seg | 22 |
| FLARE 2022 [163] | 1 | CT | 3D | Seg | 76 |
| Fundus Images for the Study of Diabetic Retinopathy [26] | 1 | Fundus Photography | 2D | Cls | 134 |
| FUSC 2021 [246] | 1 | Dermoscopy | 2D | Seg | 60 |
| GAMMA [73] | 1 | Fundus Photography | 2D | Cls | 70 |
| GlaS [229] | 1 | Histopathology | 2D | Seg | 44 |
| GOALS 2022 [69] | 1 | OCT | 2D | Seg | 180 |
| HaN-Seg [199] | 1 | CT | 3D | Seg | 96 |
| Harvard-GDP1000 [161] | 1 | Fundus Photography | 2D | Cls | 53 |
| HCC-TACE-Seg [178] | 1 | CT | 3D | Seg | 24 |
| HeartSegMRI [241] | 1 | MRI | 3D | Seg | 2 |
| HErlev [110] | 1 | Histopathology | 2D | Cls | 166 |
| HRF [35] | 1 | Fundus Photography | 2D | Cls | 3 |
| Human Protein Atlas - Single Cell Classification [252] | 1 | Microscopy | 2D | MCls | 2927 |
| HVSMR 2016 [190] | 1 | MRI | 3D | Seg | 16 |
| ICIAR 2018 [15] | 1 | Microscopy | 2D | Cls | 28 |
| ICIAR 2018 [15] | 1 | Microscopy | 2D | Seg | 238 |
| IDRiD [202] | 1 | Fundus Photography | 2D | Seg | 232 |
| Intel & MobileODT Cervical Cancer Screening [27] | 1 | Colposcopy | 2D | Cls | 90 |
| ISIC 2016 [88] | 1 | Dermoscopy | 2D | Cls | 60 |
| ISIC 2016 [88] | 1 | Dermoscopy | 2D | Seg | 48 |
| ISIC 2018 [242] | 1 | Dermoscopy | 2D | Seg | 252 |
| ISIC 2018 [242] | 1 | Dermoscopy | 2D | Cls | 32 |
| ISIC 2019 | 1 | Dermoscopy | 2D | Cls | 171 |
| ISIC 2020 [213] | 1 | Dermoscopy | 2D | Cls | 30 |
| ISPY1-Tumor-SEG-Radiomics [48] | 1 | DCE MRI | 3D | Seg | 60 |
| IVDM3Seg [86] | 1 | Fat MRI, Water MRI, In-phase MRI, Opposed-phase MRI | 3D | Seg | 60 |
| IvyGAP-Radiomics [194] | 1 | FLAIR MRI | 3D | Seg | 2 |
| JSIEC [41] | 1 | Fundus Photography | 2D | Cls | 509 |
| JSRT [226] | 1 | X-ray | 2D | Seg | 60 |
| JSRT [226] | 1 | X-ray | 2D | Cls | 120 |

| Dataset | | Modality | Dim | Task | Count |
|---|---|---|---|---|---|
| Kidney Boundary Detection [94] | 1 | Endoscopy | 2D | Seg | 44 |
| KiPA 2022 [97] | 1 | CT | 3D | Seg | 158 |
| KiTS 2019 [99] | 1 | CT | 3D | Seg | 16 |
| KiTS 2021 [269] | 1 | CT | 3D | Seg | 82 |
| Knee Osteoarthritis Dataset with Severity Grading [45] | 1 | X-ray | 2D | Cls | 150 |
| Kvasir [200] | 1 | Endoscopy | 2D | Cls | 237 |
| Kvasir-SEG [114] | 1 | Endoscopy | 2D | Seg | 10 |
| KvasirCapsule-SEG [115] | 1 | Endoscopy | 2D | Seg | 6 |
| LAScarQS 2022 [140] | 1 | LGE MRI | 3D | Seg | 2 |
| LC25000 [34] | 1 | Histopathology | 2D | Cls | 150 |
| Learn2Reg2022 | 1 | CT | 3D | Seg | 56 |
| Leukemia Classification [87] | 1 | Microscopy | 2D | Cls | 32 |
| LiTS [32] | 1 | CT | 3D | Seg | 24 |
| LNDb [196] | 1 | CT | 3D | Seg | 20 |
| Longitudinal Multiple Sclerosis Lesion Segmentation Challenge [39] | 1 | MP-RAGE MRI, T2 MRI, PD MRI, FLAIR MRI | 3D | Seg | 22 |
| LUAD-CT-Survival [82] | 1 | CT | 3D | Seg | 60 |
| LUNA 2016 [224] | 1 | CT | 3D | Seg | 8 |
| LYSTO [245] | 1 | Histopathology | 2D | Cls | 853 |
| M&Ms-2 [170] | 1 | MRI | 3D | Seg | 12 |
| m2cai16-tool-locations [117] | 1 | Endoscopy | 2D | Det | 210 |
| m2caiSeg [169] | 1 | Endoscopy | 2D | Seg | 690 |
| Malaria from Heywhale | 1 | Histopathology | 2D | Cls | 30 |
| Malignant Lymphoma Classification [189] | 1 | Histopathology | 2D | Cls | 90 |
| MED-NODE [80] | 1 | Dermoscopy | 2D | Cls | 11 |
| MESSIDOR [4] | 1 | Fundus Photography | 2D | Cls | 60 |
| MHSMA [112] | 4 | Microscopy | 2D | Cls | 234 |
| MIAS Mammography [235] | 1 | X-ray | 2D | Cls | 145 |
| MM-WHS 2017 [160] | 1 | MRI, CT | 3D | Seg | 140 |
| Mpox Skin Lesion Dataset [108] | 1 | Dermoscopy | 2D | Cls | 150 |
| MRL Eye Dataset [76] | 6 | Infrared Reflectance (IR) imaging | 2D | Cls | 329 |
| MSD - Colon [227] | 1 | CT | 3D | Seg | 60 |
| MSD - Heart [227] | 1 | MRI | 3D | Seg | 2 |
| MSD - HepaticVessel [14] | 1 | CT | 3D | Seg | 60 |
| MSD - Liver [14] | 1 | CT | 3D | Seg | 16 |
| MSD - Lung [14] | 1 | CT | 3D | Seg | 18 |
| MSD - Pancreas [14] | 1 | CT | 3D | Seg | 68 |
| MSD - Spleen [14] | 1 | CT | 3D | Seg | 6 |
| MSSEG 2008 [258] | 1 | T2 MRI, T1 MRI | 3D | Seg | 6 |
| MSSEG 2016 [53] | 1 | T2 MRI, MRI, Gadolinium MRI, T1 MRI, FLAIR MRI | 3D | Seg | 32 |
| MyoPS 2020 [160] | 1 | DE MRI, T2 MRI, MRI | 3D | Seg | 100 |
| NIH Chest X-rays [236] | 1 | X-ray | 2D | Cls | 16 |
| NIH Chest X-rays [187, 250] | 1 | X-ray | 2D | MCls | 2293 |
| NODE21 [231] | 1 | X-ray | 2D | Det | 4 |
| OCCISCOverlapping Cervical Cytology Image Segmentation [156, 157] | 1 | Microscopy | 2D | Seg | 90 |
| ODIR 2019 | 1 | Fundus Photography | 2D | MCls | 116 |
| OLIVES [203] | 1 | Fundus Photography | 2D | Cls | 60 |
| Osteosarcoma-Tumor-Assessment [230] | 1 | Histopathology | 2D | Cls | 60 |
| PAD-UFES-20 [191] | 1 | Dermoscopy | 2D | Cls | 68 |
| PALM 2019 [107] | 1 | Fundus Photography | 2D | Cls | 25 |
| PANDA [36] | 1 | Histopathology | 2D | Cls | 139 |
| PanNuke [77, 78] | 1 | Histopathology | 2D | Seg | 300 |
| Parse 2022 [158] | 1 | CT | 3D | Seg | 14 |
| PDDCA [210] | 2 | CT | 3D | Seg | 78 |
| PH2 Database [175] | 1 | Dermoscopy | 2D | Cls | 97 |
| PI-CAI [217] | 1 | T2 weighted MRI, MRI | 3D | Seg | 32 |

| | | | | | |
|---|---|---|---|---|---|
| PI-CAI [217] | 1 | T2 weighted MRI, MRI | 3D | Seg | 28 |
| PitVis | 1 | Endoscopy | 2D | Cls | 360 |
| PleThora [133] | 1 | CT | 3D | Seg | 120 |
| PROMISE 2009 [31] | 1 | T2 weighted MRI | 3D | Seg | 8 |
| PROMISE 2012 [144] | 1 | MRI | 3D | Seg | 8 |
| Prostate-Anatomical-Edge-Cases [123] | 1 | CT | 3D | Seg | 18 |
| PROSTATEx-Seg-HiRes [220] | 1 | T2 weighted MRI | 3D | Seg | 6 |
| Pulmonary Chest X-Ray Abnormalities [109] | 1 | X-ray | 2D | Cls | 12 |
| Pulmonary Chest X-Ray Abnormalities [244] | 1 | X-ray | 2D | Cls | 13 |
| Pulmonary Embolism in CT images [171] | 1 | CT | 3D | Seg | 14 |
| QIBA-VolCT-1B [173] | 1 | CT | 3D | Seg | 60 |
| QIN-LungCT-Seg [113] | 1 | CT | 3D | Seg | 6 |
| QIN-PROSTATE-Repeatability [70] | 1 | T2 weighted MRI, DCE MRI, ADC MRI | 3D | Seg | 80 |
| RadImageNet [174] | 1 | UltraSound, MRI, CT | 2D | Cls | 4608 |
| RAVIR [93] | 1 | Infrared Reflectance (IR) imaging | 2D | Seg | 92 |
| REFUGE2 [139, 188] | 1 | Fundus Photography | 2D | Seg | 20 |
| Retina Fundus Image Registration [100] | 1 | OCT | 2D | Cls | 135 |
| Retinal OCT Images [131] | 1 | OCT | 2D | Cls | 14 |
| RHUH-GBM [42] | 1 | T1ce MRI, T2 MRI, ADC MRI | 3D | Seg | 10 |
| RibFrac2020 [118] | 1 | CT | 3D | Seg | 60 |
| RIDER-LungCT-Seg [6] | 1 | CT | 3D | Seg | 26 |
| RIM-ONE [75] | 1 | Fundus Photography | 2D | Seg | 60 |
| RITE [104] | 1 | Fundus Photography | 2D | Seg | 16 |
| Robotic Instrument Segmentation [11] | 1 | Endoscopy | 2D | Seg | 74 |
| ROSE [165] | 1 | Fundus Photography | 2D | Seg | 30 |
| RSNA Intracranial Hemorrhage Detection [71] | 1 | CT | 2D | MCls | 289 |
| RSNA Pediatric Bone Age Challenge [89] | 1 | X-ray | 2D | Cls | 1 |
| RUS-CHN | 1 | X-ray | 2D | Cls | 265 |
| RUS-CHN SAML [151] | 1 | T2 weighted MRI | 3D | Seg | 6 |
| SARAS-MESAD [25, 24] | 1 | Endoscopy | 2D | Det | 635 |
| SEG.A. 2023 [119, 197, 208, 168] | 1 | CT | 3D | Seg | 2 |
| SegPC-2021 [15, 32] | 1 | Histopathology | 2D | Seg | 30 |
| SegTHOR [98] | 1 | CT | 3D | Seg | 48 |
| SIIM-ACR Pneumothorax Segmentation [264] | 1 | X-ray | 2D | Seg | 16 |
| SIIM-ACR Pneumothorax Segmentation [264] | 1 | X-ray | 2D | Cls | 58 |
| SIIM-FISABIO-RSNA COVID-19 Detection [130] | 1 | X-ray | 2D | Cls | 90 |
| SinaFarsiu-008-Chiu BOE 2012 [50] | 1 | OCT | 2D | Seg | 46 |
| SinaFarsiu-009-Chiu BOE 2013 [49] | 1 | OCT | 2D | Seg | 8 |
| SinaFarsiu-010-Rabbani IOVS 2014 [207] | 1 | OCT | 2D | Seg | 48 |
| SinaFarsiu-013-Estrada PAMI 2015 [67] | 1 | OCT | 2D | Cls | 30 |
| SLIVER 2007 [98] | 1 | CT | 3D | Seg | 6 |
| SLN-Breast [38] | 1 | Histopathology | 2D | Cls | 2 |
| SPPIN2023 | 1 | T1Gd MRI | 3D | Seg | 60 |
| STACOM SLAWT 2016 [124] | 1 | MRI, CT | 3D | Seg | 4 |
| StructSeg 2019 [98] | 4 | CT | 3D | Seg | 242 |
| SUN-SEG [176] | 1 | Endoscopy | 2D | Seg | 6 |
| Surgical Instrument Multi-Domain Segmentation Challenge | 1 | Endoscopy | 2D | Seg | 210 |

| | | | | | |
|---|---|---|---|---|---|
| Surgical Instrument Multi-Domain Segmentation Challenge | 1 | Endoscopy | 2D | Seg | 2 |
| Syn-ISS | 1 | Endoscopy | 2D | Seg | 58 |
| TCB Challenge [92] | 1 | Texture Characterization of Bone Radiograph | 2D | Cls | 60 |
| TotalSegmentator [251] | 1 | CT | 3D | Seg | 1218 |
| UCSF-PDGM [37] | 1 | ASL MRI, DWI MRI, T1 weighted MRI, SWI MRI, DTI MRI, MRI, FLAIR MRI | 3D | Seg | 22 |
| Ultrasound Nerve Segmentation [179] | 1 | UltraSound | 2D | Seg | 60 |
| UW-Madison GI Tract Image Segmentation [91] | 1 | MRI | 2D | Seg | 150 |
| VerSe 2019 [223, 153] | 1 | CT | 3D | Seg | 94 |
| VerSe 2020 [223, 153] | 1 | CT | 3D | Seg | 14 |
| VinBigData Chest X-ray Abnormalities Detection [66] | 1 | X-ray | 2D | Det | 107 |
| WORD [159] | 1 | CT | 3D | Seg | 72 |
| Yangxi Dataset [146] | 1 | Fundus Photography | 2D | Cls | 60 |
| In-House Dataset | 1 | Fundus Photography | 2D | Cls | 23 |
| In-House Dataset | 1 | CT | 3D | Seg | 40 |
| In-House Dataset | 1 | CT | 3D | Seg | 12 |
| In-House Dataset | 1 | CT | 3D | Seg | 80 |
| In-House Dataset | 1 | CTA | 3D | Seg | 10 |
| In-House Dataset | 1 | CT | 3D | Seg | 18 |
| In-House Dataset | 1 | CT | 3D | Seg | 34 |
| In-House Dataset | 1 | CT | 3D | Seg | 60 |
| In-House Dataset | 1 | CT | 3D | Seg | 76 |
| In-House Dataset | 1 | CT | 3D | Seg | 60 |
| In-House Dataset | 1 | CT | 3D | Seg | 18 |
| In-House Dataset | 1 | CT | 3D | Seg | 96 |
| In-House Dataset | 1 | CT | 3D | Seg | 150 |
| In-House Dataset | 1 | CT | 3D | Seg | 40 |
| In-House Dataset | 1 | CT | 3D | Seg | 14 |
| In-House Dataset | 1 | CT | 3D | Seg | 82 |

# C Details of Well-categorized Data Structure

## C.1 Data Statistics

In this section, we present the comprehensive statistical information of GMAI-MMBench. Figure 6 offers a global view of the label distribution proportions for different clinical VQA tasks, departments, and perceptual granularities. The left pie chart (A) shows the distribution of clinical VQA tasks, with Disease Diagnosis (DD) being the most prevalent at 51.6%, followed by Severity Grading (SG) at 9.1%, Counting (C) at 5.4%, and Organ Recognition – Abdomen (OR-A) at 4.0%. The middle pie chart (B) depicts the distribution of cases across various departments, where Ophthalmolog (O) has the highest proportion at 11.3%, followed by Hematology (H) at 10.7%, General Surgery (GS) at 10.2%, and Urolog (U) at 9.7%. The right pie chart (C) represents the distribution of perceptual granularities, with Image Level accounting for the largest share at 49.2%, followed by Mask Level at 22.0%, and Contour Level at 22.0%. Specifically, Table 5 provides the statistical details for different clinical VQA tasks, including their full terms, abbreviations, and the number of questions associated with each task. Table 6 presents the statistical information for different departments, including each department's full term, abbreviation, and the number of questions contained within each department. Table 7 shows the statistical information for different granularity. In the detailed tables, the statistical information for multiple-choice questions is also included, **specially, for multiple-choice questions, we count the frequency of choice appearances rather than the actual number of cases.**

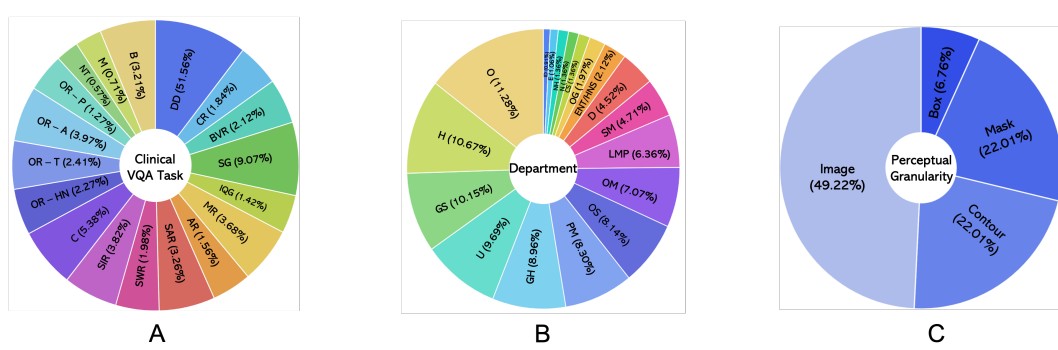

Figure 6: Label distribution for clinical VQA tasks, departments, and perceptual granularities.

Table 5: Statistics of the clinical VQA tasks and their sub-task abbreviations mentioned in the paper with their corresponding full terms.

| Full Name | Abbreviation | Single Choice | | | Multiple Choice | | |
|---|---|---|---|---|---|---|---|
| | | Modalities | Labels | Cases | Modalities | Labels | Cases |
| Attribute Recognition | AR | 5 | 26 | 780 | 1 | 4 | 40 |
| Blood Vessels Recognition | BVR | 7 | 15 | 436 | - | - | - |
| Bone | B | 6 | 22 | 655 | - | - | - |
| Cell Recognition | CR | 4 | 13 | 383 | 1 | 18 | 7614 |
| Counting | C | 1 | 38 | 853 | - | - | - |
| Disease Diagnosis | DD | 29 | 364 | 10167 | 3 | 26 | 8037 |
| Image Quality Grading | IQG | 2 | 10 | 300 | - | - | - |
| Microorganism Recognition | MR | 3 | 26 | 779 | - | - | - |
| Muscle | M | 1 | 5 | 150 | - | - | - |
| Nervous Tissue | NT | 2 | 4 | 120 | - | - | - |
| Organ Recognition - Abdomen | OR-A | 7 | 28 | 838 | - | - | - |
| Organ Recognition - Head and Neck | OR-HN | 5 | 16 | 480 | - | - | - |
| Organ Recognition - Pelvic | OR-P | 6 | 9 | 270 | - | - | - |
| Organ Recognition - Thorax | OR-T | 9 | 17 | 510 | - | - | - |
| Severity Grading | SG | 5 | 64 | 1678 | - | - | - |
| Surgeon Action Recognition | SAR | 1 | 23 | 635 | - | - | - |
| Surgical Instrument Recognition | SIR | 1 | 27 | 790 | - | - | - |
| Surgical Workflow Recognition | SWR | 1 | 14 | 420 | - | - | - |

Table 6: Statistics of the departments and their sub-task abbreviations mentioned in the paper with their corresponding full terms.

| Full Name | Abbreviation | Single Choice | | | Multiple Choice | | |
|---|---|---|---|---|---|---|---|
| | | Modalities | Labels | Cases | Modalities | Labels | Cases |
| Cardiovascular Surgery | CS | 9 | 9 | 270 | 1 | 1 | 424 |
| Dermatology | D | 1 | 30 | 894 | - | - | - |
| Endocrinology | E | 3 | 7 | 210 | - | - | - |
| Gastroenterology and Hepatology | GH | 7 | 60 | 1774 | - | - | - |
| General Surgery | GS | 6 | 68 | 2009 | - | - | - |
| Hematology | H | 6 | 80 | 2112 | - | - | - |
| Infectious Diseases | ID | 2 | 7 | 180 | - | - | - |
| Laboratory Medicine and Pathology | LMP | 2 | 45 | 1259 | 1 | 18 | 7614 |
| Nephrology and Hypertension | NH | 4 | 9 | 270 | - | - | - |
| Neurosurgery | N | 8 | 9 | 270 | - | - | - |
| None (Attributes that do not belong to any department) | N/A | 2 | 15 | 450 | - | - | - |
| Obstetrics and Gynecology | OG | 5 | 14 | 389 | - | - | - |
| Oncology (Medical) | OM | 20 | 51 | 1399 | - | - | - |
| Ophthalmology | O | 6 | 97 | 2232 | 2 | 11 | 218 |
| Orthopedic Surgery | OS | 8 | 54 | 1611 | - | - | - |
| Otolaryngology (ENT)/Head and Neck Surgery | ENT/HNS | 5 | 14 | 420 | 1 | 6 | 1015 |
| Pulmonary Medicine | PM | 2 | 55 | 1643 | 1 | 12 | 6420 |
| Sports Medicine | SM | 3 | 64 | 1919 | - | - | - |
| Urology | U | 8 | 33 | 933 | - | - | - |

Table 7: Statistics of the perceptual granularities. $*$ and $\#$ denote the case for single choice and multiple choice, respectively.

| Full Name | Modalities | Labels | Cases |
|---|---|---|---|
| Mask Level | 36 | 188 | 5587 |
| Contour Level | 36 | 188 | 5587 |
| Box Level | 3 | 59 | 1715 |
| Image Level$^*$ | 13 | 474 | 12942 |
| Image Level$^\#$ | 5 | 48 | 15691 |

## C.2 Lexical Tree

To make the GMAI-MMBench more intuitive and user-friendly, we have systematized our labels and structured the entire dataset into a lexical tree, which is presented in HTML format as shown in Figure 7. Users can freely select the test contents based on this lexical tree. We believe that this customizable benchmark will effectively guide the improvement of models in specific areas. For instance, as mentioned in the main text, most models perform poorly at the bounding box level perception. Users can then update their models and test the accuracy at the bounding box level using this lexical tree, thereby achieving targeted improvements in model performance.

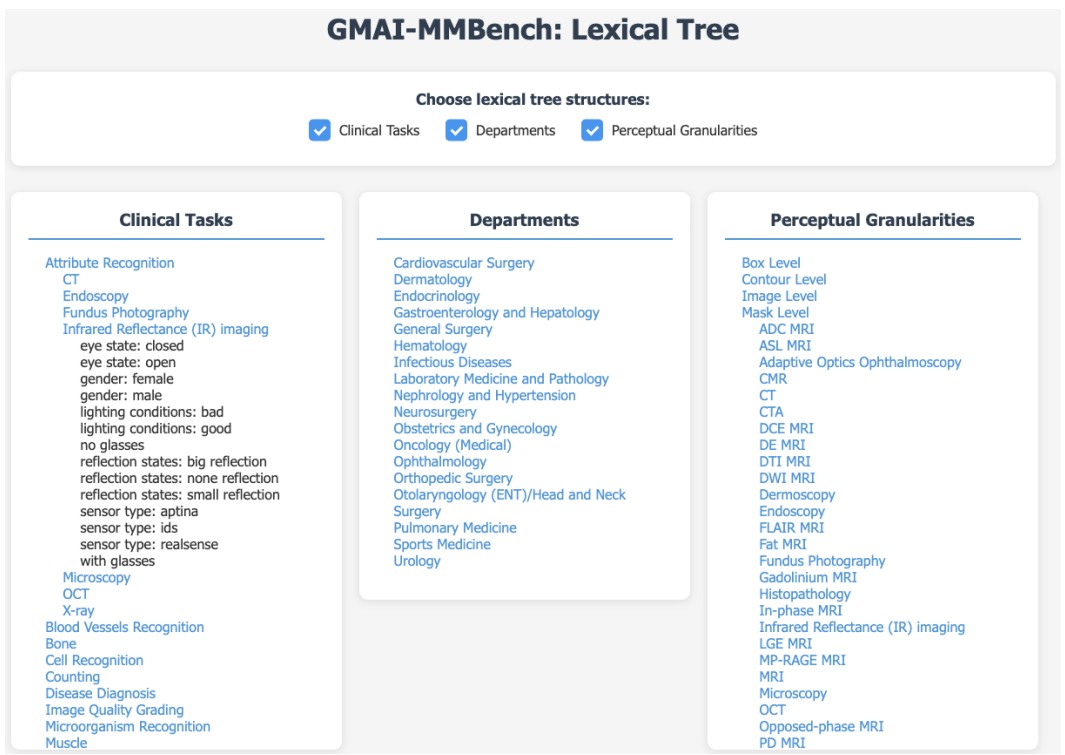

Figure 7: Overview of the lexical tree. The whole tree is provided in the attached HTML file named "Lexical tree.html".

Here, we specifically demonstrate how to customize the use of the lexical tree. First, select the data we need to test based on the users' requirements. In this example, we will focus on **ophthalmology** department and only **fundus photography** modality.

**Step-by-Step Process:**

1. **Select the Department:** First, navigate to the Lexical Tree interface and select the department relevant to our testing. In our case, we choose the "Ophthalmology" department from the available clinical tasks, as shown in Figure 8.

2. **Choose the Modality:** Within the ophthalmology department, several modalities related to eye conditions are listed. We specifically select the "Fundus Photography" modality. This selection allows us to access all the keywords associated with fundus images, which are crucial for the next step.

3. **Keyword Filtering:** After selecting the fundus photography modality, a comprehensive list of keywords appears. These keywords are critical as they will be used to filter the relevant questions for the evaluation. Examples of keywords include "advanced glaucoma", "age-related macular degeneration", and "diabetic retinopathy" among others.

4. **Retrieve Question List:** The system filters and retrieves questions from the pre-prepared question list using the selected keywords. Each question includes multiple options, and the correct answer corresponds to the keyword used for filtering. However, the correct answers are hidden from the users during the evaluation process. For instance, a question may ask about identifying a condition shown in an image, with options like "A. advanced glaucoma", "B. early glaucoma", "C. non glaucoma", etc. The correct answer, such as "advanced glaucoma" is derived from the keyword used for filtering.

5. **Model Evaluation:** The filtered question list is then used to evaluate various models. In this example, models such as GPT-4, Claude3-Opus, Qwen-Max, and others are assessed for their accuracy in answering the questions. The results are compiled and displayed in a tabular format, showcasing each model's performance.

In addition to the provided example, this method allows for the independent testing of **any other departments, modalities, clinical tasks, and their combinations.** For instance, if the objective is to evaluate only ophthalmology, fundus photographs, and disease diagnosis tasks, further refinement of the keywords can be achieved following the initial selection. By accessing the disease diagnosis task and selecting the fundus photography modality, we can intersect the keywords from the department-fundus photography section with those from the clinical tasks-disease diagnosis section. The resulting keywords will represent those relevant exclusively to disease diagnosis tasks within the context of fundus photographs in ophthalmology.

In summary, the lexical tree provides a versatile framework for customizing evaluation processes across various medical domains, ensuring a comprehensive and focused assessment of model performance.

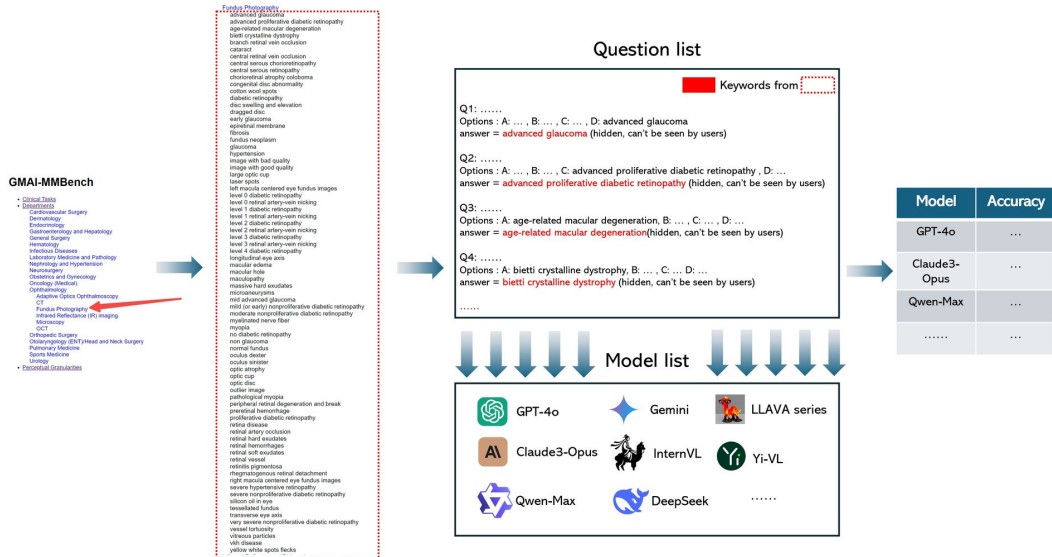

Figure 8: Example of how to use the Lexical Tree for customizing evaluations for the **ophthalmology** department and **fundus photography** modality. The process involves selecting the department (ophthalmology), choosing the modality (fundus photography), filtering questions using relevant keywords, and evaluating different models based on their accuracy in answering the filtered questions.

# D  Evaluation

In this section, we will describe the evaluation process in detail. We evaluated various LVLMs, including medical-specific models, open-source general models, and closed-source API general models. We selected versions with approximately 7 billion parameters for testing, and the model weights were sourced from their respective official Hugging Face repositories. Our evaluation was conducted using the VLMEvalKit[8] framework. For medical-specific models, we utilized the Multi-Modality-Arena[9] repository for testing. Specifically, we input the prompt shown in Table 8 into the tested model to for evaluation, the option-only answers are expected. However, it's hard for some models to follow the instructions, if a model neither outputs a clear answer tagged by the letter options nor provides instructions to select an answer, we use ChatGPT-3.5-turbo-0613 to extract the answer from the model's outputs. If the answer cannot be extracted, we treat the outputs as errors. Otherwise, the extracted answers will be considered as the model's predicted answer for that question.

Table 8: Examples of single-choice and multiple-choice question prompts.

| **Prompt example for single-choice questions** |
| --- |
| *Question*: Observe the image. What is the most likely abnormality shown in the picture? |
| *Options*: |
| A.osteoporotic bone |
| B.healthy bone |
| Please select the correct answer from the options above. |
| <image> |
| **Prompt example for multiple-choice questions** |
| *Question*: Determine which part(s) is illustrated in the image. |
| *Options*: |
| A. cytosol |
| B. actin filaments |
| C. vesicles and punctate cytosolic patterns |
| D. microtubules |
| E. plasma membrane |
| F. endoplasmic reticulum |
| Please select all correct answers from the options above. Note that there is more than one correct answer. Please output the answer options directly, separated by commas. For example: A,B |
| <image> |

## D.1  Evaluation Metric for Single-choice Questions

For all single-choice questions, we denote $n_{\text{correct}}$ as the number of questions for which the model offered the correct answer, and $n_{\text{questions}}$ as the total number of questions. The ACC can be calculated as follows:

$$\text{ACC} = \frac{n_{\text{correct}}}{n_{\text{questions}}}. \tag{1}$$

## D.2  Evaluation Metric for Multiple-choice Questions

For all multiple-choice questions, we first count the number of correct predictions by the model within the groundtruth for each case, denoted as $n_{\text{match}}$. The length of the prediction is denoted as $l_{\text{prediction}}$, and the length of the groundtruth options are denoted as $l_{\text{truth}}$. The evaluation metrics for multiple-choice questions is calculated as follows:

$$\text{ACC}_{mcls} = \frac{n_{\text{match}}}{l_{\text{prediction}}}, \tag{2}$$

$$\text{Recall}_{mcls} = \frac{n_{\text{match}}}{l_{\text{truth}}}. \tag{3}$$

---

[8]https://github.com/open-compass/VLMEvalKit
[9]https://github.com/OpenGVLab/Multi-Modality-Arena/tree/main/MedicalEval/Question-answering_Score

Table 9: The model architecture of 50 LVLMs evaluated on GMAIMMBench.

| Series | Models | #Params | Vision Encoder | LLM |
|---|---|---|---|---|
| Med model series | MedVInT [268] | - | - | - |
| | Med-Flamingo [181] | 8.3B | CLIP ViT/L-14 | LLaMA-7B |
| | LLaVA-Med [138] | - | CLIP ViT/L-14 | Mistral-7B |
| | RadFM [254] | 14B | 3D ViT | MedLLaMA-13B |
| | Qilin-Med-VL-Chat [149] | - | Clip ViT/L-14 | Chinese-LLaMA2-Chat-13B |
| | MedDr [95] | 40B | InternViT-6B | Nous-Hermes-2-Yi-34B |
| Ungrouped series | TransCore-M [3] | 13.4B | CLIP ViT/L-14 | PCITransGPT-13B |
| | VisualGLM-6B [61] | 7.8B | EVA-CLIP | ChatGLM-6B |
| | mPLUG-Owl2 [259] | 8.2B | CLIP ViT-L/14 | LLaMA2-7B |
| | OmniLMM-12B [261] | 12B | EVA02-5B | Zephyr-7B-$\beta$ |
| | PandaGPT 13B [234] | 13B | ImageBind ViT-H/14 | Vicuna-v0-13B |
| | Mini-Gemini-7B [141] | 7B | CLIP-L | Vicuna-v1.5-7B |
| | Emu2-Chat [237] | 37B | EVA-02-CLIP-E-plus | LLaMA-33B |
| | Flamingo v2 [17] | 9B | CLIP ViT-L/14 | MPT-7B |
| | MMAlaya [154] | 7.8B | EVA-G | Alaya-7B-Chat |
| CogVLM series | CogVLM-Chat [249] | 17B | EVA-CLIP-E | Vicuna-v1.5-7B |
| | CogVLM-grounding-generalist [249] | 17B | EVA-CLIP-E | Vicuna-v1.5-7B |
| InstructBLIP series | InstructBLIP-7B [56] | 8B | EVA-G | Vicuna-7B |
| DeepSeek series | DeepSeek-VL-1.3B [155] | 1.3B | SAM-B & SigLIP-L | DeekSeek-1B |
| | DeepSeek-VL-7B [155] | 7.3B | SAM-B & SigLIP-L | DeekSeek-7B |
| Idefics series | Idefics-9B-Instruct [137] | 9B | CLIP ViT-H/14 | LLaMA 7B |
| XComposer series | ShareCaptioner [43] | 8B | EVA-G | InternLM-7B |
| | XComposer [266] | 8B | EVA-CLIP-G | InternLM-7B |
| | XComposer2 [62] | 7B | CLIP ViT-L/14 | InternLM2-7B |
| | XComposer2-4KHD [63] | 7B | CLIP ViT-L/14 | InernLM2-7B |
| Yi-VL series | Yi-VL-6B [7] | 6.6B | CLIP ViT-H/14 | Yi-6B |
| InternVL series | InternVL-Chat-V1.1 [47] | 19B | InternViT-6B | LLaMA2-13B |
| | InternVL-Chat-V1.2 [47] | 40B | InternViT-6B | Nous-Hermes-2-Yi-34B |
| | InternVL-Chat-V1.2-Plus [47] | 40B | InternViT-6B | Nous-Hermes-2-Yi-34B |
| | InternVL-Chat-V1.5 [46] | 25.5B | InternViT-6B | InternLM2-Chat-20B |
| LLaVA series | LLaVA-NeXT-mistral-7B [147] | 7.6B | CLIP ViT-L/14 | Mistral-7B |
| | LLaVA-NeXT-vicuna-7B [147] | 7.1B | CLIP ViT-L/14 | Vicuna-v1.5-7B |
| | LLAVA-V1.5-7B [148] | 7.2B | CLIP ViT-L/14 | Vicuna-v1.5-7B |
| | ShareGPT4V-7B [43] | 7.2B | CLIP ViT-L/14 | Vicuna-v1.5-7B |
| Xtuner series | LLAVA-InternLM-7b [54] | 7.6B | CLIP ViT-L/14 | InternLM-7B |
| | LLAVA-InternLM2-7b [54] | 8.1B | CLIP ViT-L/14 | InternLM2-7B |
| | LLAVA-V1.5-7B-xtuner [54] | 7.2B | CLIP ViT-L/14 | Vicuna-v1.5-7B |
| | LLAVA-V1.5-13b-xtuner [54] | 13.4B | CLIP ViT-L/14 | Vicuna-v1.5-13B |
| MiniCPM series | MiniCPM-V [103] | 2.8B | SigLip-400M | MiniCPM-2.4B |
| | MiniCPM-V2 [257] | 2.8B | SigLip-400M | MiniCPM-2.4B |
| Qwen series | Monkey [142] | 9.8B | CLIP-ViT-BigHuge | Qwen-7B |
| | Monkey-Chat [142] | 9.8B | ViT-BigHuge | Qwen-7B |
| | Qwen-VL [19] | 9.6B | CLIP ViT-G/16 | QWen-7B |
| | Qwen-VL-Chat [19] | 9.6B | CLIP ViT-G/16 | Qwen-7B |
| API series | Qwen-VL-Max [18] | - | - | QwenLM |
| | Claude3-Opus [13] | - | - | - |
| | GPT-4o [5] | - | - | - |
| | GPT-4V [5] | - | - | - |
| | Gemini 1.0 [240] | - | - | - |
| | Gemini 1.5 [211] | - | - | - |

## D.3 Evaluated Models

In this paper, we evaluate 50 models on our GMAI-MMBench, and we list them in Table 9.

## E Results

In this section, we first provide the complete quantitative results in our experiments, and then perform the case study by analyzing 53 representative examples of models' outputs.

## E.1 Quantitative Results

The complete test results are shown in the table below. Table 10 shows the results in different clinical VQA tasks; Table 11 shows the results across different departments; Table 12 shows the results in different perceptual granularities.

Table 10: Results for single-choice questions of 50 different LVLMs on clinical VQA tasks. The best-performing model in each category is **in-bold**, and the second best is underlined.

| Model name | Overall (val) | Overall (test) | AR | BVR | B | CR | C | DD | IQG | MR | M | NT | OR-A | OR-HN | OR-P | OR-T | SG | SAR | SIR | SWR |
|---|---|---|---|---|---|---|---|---|---|---|---|---|---|---|---|---|---|---|---|---|
| Random | 25.70 | 25.94 | 38.20 | 22.73 | 22.92 | 22.72 | 24.06 | 26.66 | 27.13 | 27.00 | 20.00 | 24.75 | 21.37 | 22.93 | 22.33 | 21.18 | 32.43 | 24.23 | 21.39 | 23.71 |
| *Medical Special Model* | | | | | | | | | | | | | | | | | | | | |
| MedVInT [268] | 2.29 | 1.96 | 5.75 | 0.00 | 0.00 | 0.00 | 2.56 | 2.11 | 4.05 | 0.00 | 0.00 | 0.00 | 0.11 | 0.00 | 0.00 | 0.12 | 7.36 | 0.00 | 1.88 | 0.00 |
| Med-Flamingo [181] | 12.74 | 11.64 | 6.67 | 10.14 | 9.23 | 11.27 | 6.62 | 13.43 | 12.15 | 6.38 | 8.00 | 18.18 | 9.26 | 18.27 | 11.00 | 11.53 | 12.16 | 5.19 | 8.47 | 11.43 |
| LLaVA-Med [138] | 20.54 | 19.60 | 24.51 | 17.83 | 17.08 | 19.86 | 15.04 | 19.81 | 20.24 | 21.51 | 13.20 | 15.15 | 20.42 | 23.73 | 17.67 | 19.65 | 21.70 | 19.81 | 14.11 | 20.86 |
| Qilin-Med-VL-Chat [149] | 22.34 | 22.06 | 29.57 | 19.41 | 16.46 | 23.79 | 15.79 | 24.19 | 21.86 | 16.62 | 7.20 | 13.64 | 24.00 | 14.67 | 12.67 | 15.53 | 26.13 | 24.42 | 17.37 | 25.71 |
| RadFM [254] | 22.95 | 22.93 | 27.16 | 20.63 | 13.23 | 19.14 | 20.45 | 24.51 | 23.48 | 22.85 | 15.60 | 16.16 | 14.32 | 24.93 | 17.33 | 21.53 | 29.73 | 17.12 | 19.59 | 31.14 |
| MedDr [95] | 41.95 | 43.69 | 41.20 | 50.70 | 37.85 | 29.87 | 28.27 | 52.53 | 36.03 | 31.45 | 29.60 | 47.47 | 33.37 | 51.33 | 32.67 | 44.47 | 35.14 | 25.19 | 25.58 | 32.29 |
| *Open-Source LVLMs* | | | | | | | | | | | | | | | | | | | | |
| CogVLM-grounding-generalist [249] | 5.20 | 5.66 | 3.11 | 4.02 | 2.92 | 3.22 | 10.83 | 7.98 | 9.72 | 0.15 | 0.00 | 11.11 | 8.32 | 1.87 | 1.67 | 2.00 | 1.65 | 0.00 | 4.02 | 0.57 |
| XComposer [266] | 8.92 | 7.67 | 1.38 | 7.69 | 8.31 | 12.34 | 22.86 | 7.31 | 6.07 | 5.49 | 2.80 | 16.16 | 5.05 | 8.67 | 2.00 | 9.76 | 11.94 | 7.31 | 3.17 | 4.00 |
| PandaGPT 13B [234] | 16.69 | 16.27 | 24.51 | 23.60 | 22.15 | 23.61 | 14.29 | 14.95 | 13.36 | 12.17 | 18.40 | 28.79 | 18.63 | 27.33 | 18.67 | 16.71 | 11.04 | 9.23 | 13.43 | 9.71 |
| Flamingo v2 [17] | 25.58 | 26.34 | 37.74 | 21.50 | 20.62 | 22.00 | 22.41 | 27.29 | 25.91 | 27.45 | 18.00 | 28.79 | 25.16 | 22.13 | 22.00 | 22.00 | 34.61 | 22.88 | 20.44 | 27.43 |
| VisualGLM-6B [61] | 29.58 | 30.45 | 40.16 | 33.92 | 24.92 | 25.22 | 24.21 | 32.99 | 29.96 | 29.53 | 21.20 | 37.88 | 30.32 | 24.80 | 13.33 | 29.88 | 33.11 | 19.62 | 19.16 | 37.43 |
| Idefics-9B-Instruct [137] | 29.74 | 31.13 | 40.39 | 30.59 | 26.46 | 33.63 | 22.56 | 34.38 | 25.51 | 26.71 | 21.60 | 27.78 | 27.47 | 32.80 | 24.67 | 23.41 | 32.66 | 23.08 | 21.39 | 30.57 |
| InstructBLIP-7B [56] | 31.80 | 30.95 | 42.12 | 26.92 | 24.92 | 28.09 | 21.65 | 34.58 | 31.58 | 29.23 | 22.40 | 30.30 | 28.95 | 27.47 | 23.00 | 24.82 | 32.88 | 19.77 | 21.64 | 26.57 |
| Mini-Gemini-7B [141] | 32.17 | 31.09 | 29.69 | 39.16 | 31.85 | 28.26 | 10.38 | 35.58 | 29.96 | 28.78 | 20.80 | 34.34 | 29.58 | 36.53 | 24.00 | 31.76 | 22.45 | 25.96 | 18.56 | 29.43 |
| MMAlaya [154] | 32.19 | 32.30 | 41.20 | 35.14 | 32.15 | 34.17 | 27.82 | 35.09 | 28.34 | 30.27 | 18.00 | 46.97 | 20.21 | 31.20 | 16.00 | 34.59 | 32.28 | 23.65 | 22.93 | 30.29 |
| Qwen-VL [19] | 34.80 | 36.05 | 37.05 | 37.24 | 35.85 | 28.98 | 24.81 | 43.60 | 24.70 | 30.12 | 19.20 | 44.44 | 29.68 | 31.87 | 25.00 | 31.18 | 30.26 | 21.54 | 20.10 | 26.86 |
| Yi-VL-6B [7] | 34.82 | 34.31 | 41.66 | 39.16 | 26.62 | 30.23 | 31.88 | 38.01 | 26.72 | 24.93 | 25.20 | 37.37 | 29.58 | 31.20 | 32.33 | 30.59 | 36.71 | 24.81 | 23.18 | 31.43 |
| LLaVA-NeXT-vicuna-7B [147] | 34.86 | 35.42 | 40.62 | 38.64 | 21.08 | 35.42 | 23.91 | 41.22 | 32.39 | 28.04 | 20.53 | 44.95 | 27.92 | 34.98 | 20.22 | 32.82 | 33.63 | 23.08 | 25.06 | 34.86 |
| Qwen-VL-Chat [19] | 35.07 | 36.96 | 38.00 | 40.56 | 38.00 | 32.20 | 25.71 | 44.07 | 24.70 | 30.56 | 24.00 | 40.91 | 29.37 | 36.53 | 26.00 | 27.29 | 35.14 | 16.54 | 20.10 | 34.00 |
| CogVLM-Chat [249] | 35.23 | 36.08 | 40.97 | 30.77 | 27.69 | 32.74 | 19.40 | 41.10 | 36.84 | 34.72 | 24.00 | 40.91 | 36.74 | 37.33 | 26.00 | 33.65 | 36.56 | 20.19 | 23.95 | 26.57 |
| Monkey [142] | 35.48 | 36.39 | 38.32 | 35.31 | 35.54 | 34.53 | 23.16 | 43.40 | 31.98 | 30.12 | 19.20 | 33.33 | 30.00 | 32.53 | 25.33 | 31.65 | 34.46 | 20.00 | 20.27 | 30.29 |
| mPLUG-Owl2 [259] | 35.62 | 36.21 | 37.51 | 41.08 | 30.92 | 38.10 | 27.82 | 41.59 | 28.34 | 32.79 | 22.40 | 40.91 | 24.74 | 38.27 | 23.33 | 36.59 | 33.48 | 20.58 | 23.01 | 32.86 |
| ShareCaptioner [43] | 36.37 | 36.19 | 42.35 | 32.69 | 31.08 | 27.19 | 30.83 | 41.19 | 30.36 | 33.23 | 28.40 | 42.93 | 27.79 | 33.73 | 28.33 | 40.71 | 29.58 | 20.96 | 28.83 | 30.00 |
| Emu2-Chat [237] | 36.50 | 37.59 | 43.27 | 47.73 | 26.31 | 40.07 | 28.12 | 44.00 | 36.44 | 28.49 | 20.40 | 31.82 | 26.74 | 37.60 | 26.67 | 29.76 | 33.63 | 23.27 | 26.43 | 29.43 |
| XComposer2-4KHD [63] | 36.66 | 38.54 | 41.89 | 39.86 | 28.77 | 40.43 | 20.60 | 44.25 | 35.22 | 33.53 | 22.80 | 42.42 | 34.84 | 29.60 | 44.00 | 39.53 | 35.21 | 21.54 | 27.20 | 38.00 |
| ShareGPT4V-7B [43] | 36.71 | 36.70 | 43.96 | 37.59 | 21.54 | 37.57 | 18.80 | 43.26 | 32.39 | 27.30 | 22.80 | 43.43 | 29.47 | 37.33 | 22.00 | 31.76 | 34.98 | 24.42 | 25.06 | 30.00 |
| LLaVA-NeXT-mistral-7B [147] | 37.20 | 37.16 | 38.43 | 27.98 | 20.31 | 29.16 | 20.60 | 47.19 | 30.36 | 32.64 | 22.40 | 55.56 | 32.75 | 25.58 | 17.56 | 34.04 | 28.38 | 23.27 | 24.12 | 37.43 |
| LLAVA-V1.5-13b-xtuner [54] | 37.82 | 38.74 | 44.65 | 29.02 | 27.08 | 38.28 | 28.87 | 45.32 | 32.79 | 30.12 | 20.40 | 45.96 | 33.47 | 42.53 | 44.33 | 37.53 | 33.48 | 19.62 | 22.58 | 35.43 |
| OmniLMM-12B [261] | 37.89 | 39.30 | 39.82 | 40.56 | 32.62 | 37.57 | 24.81 | 46.68 | 35.63 | 35.01 | 27.60 | 57.58 | 28.42 | 34.00 | 25.00 | 29.18 | 34.46 | 24.42 | 27.54 | 40.29 |
| InternVL-Chat-V1.1 [47] | 38.16 | 39.41 | 42.46 | 43.88 | 35.23 | 45.08 | 23.31 | 45.96 | 38.87 | 29.23 | 29.60 | 40.40 | 31.68 | 41.87 | 26.67 | 38.82 | 32.13 | 19.42 | 25.58 | 30.29 |
| LLAVA-V1.5-7B [148] | 38.23 | 37.96 | 45.45 | 34.27 | 30.92 | 41.32 | 21.65 | 44.68 | 34.01 | 27.74 | 23.60 | 43.43 | 28.00 | 42.13 | 29.00 | 35.06 | 33.41 | 22.12 | 23.61 | 29.14 |
| Monkey-Chat [142] | 38.39 | 39.50 | 40.62 | 41.43 | 37.08 | 35.24 | 23.76 | 47.73 | 29.96 | 32.94 | 26.00 | 37.88 | 34.84 | 32.67 | 24.67 | 33.18 | 34.91 | 21.73 | 22.24 | 34.00 |
| LLAVA-V1.5-7B-xtuner [54] | 38.68 | 38.22 | 38.90 | 40.03 | 28.00 | 40.25 | 30.08 | 44.08 | 33.60 | 32.49 | 21.20 | 40.91 | 29.47 | 40.40 | 30.33 | 38.59 | 31.46 | 23.85 | 26.95 | 36.86 |
| XComposer2 [62] | 38.68 | 39.20 | 41.89 | 37.59 | 33.69 | 40.79 | 22.26 | 45.87 | 36.44 | 32.94 | 27.20 | 58.59 | 26.11 | 36.40 | 43.67 | 37.29 | 32.06 | 23.46 | 27.80 | 32.86 |
| LLAVA-InternLM-7b [54] | 38.71 | 39.11 | 36.36 | 36.54 | 32.62 | 38.10 | 30.68 | 46.53 | 34.82 | 28.19 | 25.20 | 48.99 | 28.11 | 40.53 | 33.33 | 36.00 | 34.08 | 26.73 | 24.12 | 29.71 |
| TransCore-M [3] | 38.86 | 38.70 | 40.74 | 41.78 | 20.77 | 35.06 | 34.74 | 45.69 | 32.39 | 32.94 | 24.40 | 44.95 | 31.05 | 38.93 | 27.00 | 33.76 | 33.86 | 23.46 | 25.49 | 31.14 |
| InternVL-Chat-V1.5 [46] | 38.86 | 39.73 | 43.84 | 44.58 | 34.00 | 33.99 | 31.28 | 45.59 | 33.20 | 38.28 | 32.40 | 42.42 | 31.89 | 42.80 | 27.00 | 36.82 | 34.76 | 23.27 | 24.72 | 32.57 |
| InternVL-Chat-V1.2-Plus [47] | 39.41 | 40.79 | 42.58 | 42.31 | 32.46 | 37.03 | 31.43 | 47.49 | 42.51 | 35.01 | 21.20 | 50.51 | 34.95 | 42.93 | 22.67 | 42.47 | 35.74 | 22.31 | 24.98 | 28.29 |
| InternVL-Chat-V1.2 [47] | 39.52 | 40.01 | 41.66 | 44.06 | 27.38 | 38.46 | 34.29 | 46.99 | 33.60 | 34.42 | 21.20 | 47.98 | 30.63 | 42.80 | 27.67 | 35.88 | 35.59 | 23.85 | 24.98 | 28.00 |
| LLAVA-InternLM2-7b [54] | 40.07 | 40.45 | 39.82 | 37.94 | 30.62 | 35.24 | 29.77 | 48.97 | 34.01 | 25.96 | 20.80 | 53.03 | 30.95 | 42.67 | 32.00 | 39.88 | 32.43 | 21.73 | 24.38 | 38.00 |
| DeepSeek-VL-1.3B [155] | 40.25 | 40.77 | 38.55 | 35.14 | 38.92 | 40.07 | 27.97 | 48.12 | 35.63 | 31.75 | 22.80 | 46.97 | 40.74 | 44.93 | 31.00 | 40.47 | 33.33 | 22.31 | 21.39 | 31.71 |
| MiniCPM-V [103] | 40.95 | 41.05 | 39.70 | 46.50 | 36.31 | 39.36 | 22.26 | 48.09 | 34.82 | 35.76 | 24.00 | 45.45 | 34.11 | 44.80 | 23.00 | 44.47 | 36.19 | 21.15 | 23.95 | 35.14 |
| DeepSeek-VL-7B [155] | 41.73 | 43.43 | 38.43 | 47.03 | 42.31 | 37.03 | 26.47 | 51.11 | 33.20 | 31.16 | 26.00 | 44.95 | 36.00 | 58.13 | 36.33 | 47.29 | 34.91 | 18.08 | 25.49 | **39.43** |
| MiniCPM-V2 [257] | 41.79 | 42.54 | 40.74 | 43.01 | 36.46 | 37.57 | 27.82 | 51.08 | 28.74 | 29.08 | 26.80 | 47.47 | 37.05 | 46.40 | 25.33 | 46.59 | 35.89 | 22.31 | 23.44 | 31.71 |
| *Proprietary LVLMs* | | | | | | | | | | | | | | | | | | | | |
| Claude3-Opus [13] | 32.37 | 32.44 | 1.61 | 39.51 | 34.31 | 31.66 | 12.63 | 39.26 | 28.74 | 30.86 | 22.40 | 37.37 | 25.79 | 41.07 | 29.33 | 33.18 | 31.31 | 21.35 | 23.87 | 4.00 |
| Qwen-VL-Max [18] | 41.34 | 42.16 | 32.68 | 44.58 | 31.38 | 40.79 | 10.68 | 50.53 | 32.79 | 44.36 | 29.20 | 51.52 | 41.37 | 58.00 | 30.67 | 41.65 | 26.95 | 25.00 | 24.64 | 39.14 |
| GPT-4V [5] | 42.50 | 44.08 | 29.92 | 48.95 | 44.00 | 37.39 | 12.93 | 52.88 | 32.79 | 44.21 | 32.80 | 63.64 | 39.89 | 54.13 | 37.00 | 50.59 | 27.55 | 23.08 | 25.75 | 37.43 |
| Gemini 1.0 [240] | 44.38 | 44.93 | 42.12 | 45.10 | 46.46 | 37.57 | 20.45 | 53.29 | 35.22 | 36.94 | 25.20 | 51.01 | 34.74 | 59.60 | 34.00 | 50.00 | **36.64** | 23.65 | 23.87 | 35.43 |
| Gemini 1.5 [211] | 47.42 | 48.36 | 43.50 | 56.12 | 51.23 | 47.58 | 2.26 | 55.33 | 38.87 | 48.07 | 30.00 | **76.26** | 51.05 | **75.87** | 46.33 | 62.24 | 20.57 | **27.69** | **30.54** | **40.57** |
| GPT-4o [5] | **53.53** | **53.96** | 38.32 | **61.01** | **57.08** | **49.02** | 46.62 | **61.45** | **46.56** | **56.38** | 34.00 | 75.25 | **53.79** | 69.47 | **48.67** | **65.88** | 33.93 | 22.88 | 29.51 | 39.43 |

Table 11: Results for single-choice questions of 50 LVLMs on different departments. The best-performing model in each category is **in-bold**, and the second best is underlined.

| Model name | Overall (val) | Overall (test) | CS | D | E | GH | GS | H | ID | LMP | NH | N | OG | OM | O | OS | ENT/HNS | PM | SM | U |
|---|---|---|---|---|---|---|---|---|---|---|---|---|---|---|---|---|---|---|---|---|
| Random | 25.70 | 25.94 | 22.82 | 25.19 | 21.00 | 25.97 | 22.24 | 24.45 | 31.13 | 28.99 | 22.86 | 24.00 | 29.15 | 27.77 | 30.36 | 25.92 | 22.53 | 24.74 | 22.87 | 29.19 |
| *Medical Special Model* | | | | | | | | | | | | | | | | | | | | |
| MedVInT [268] | 2.29 | 1.96 | 0.24 | 2.50 | 1.00 | 1.94 | 1.09 | 0.88 | 3.31 | 5.23 | 1.14 | 0.73 | 0.00 | 1.40 | 4.44 | 0.56 | 0.00 | 2.24 | 0.64 | 0.86 |
| Med-Flamingo [181] | 12.74 | 11.64 | 11.76 | 12.49 | 10.00 | 10.88 | 9.33 | 5.42 | 7.28 | 10.05 | 12.00 | 10.91 | 12.88 | 14.89 | 15.37 | 12.40 | 13.43 | 12.89 | 14.92 | 10.47 |
| LLaVA-Med [138] | 20.54 | 19.60 | 26.12 | 20.20 | 29.00 | 20.31 | 16.30 | 18.46 | 15.23 | 21.84 | 20.86 | 16.73 | 21.69 | 19.23 | 20.18 | 18.38 | 20.99 | 16.87 | 20.49 | 21.55 |
| Qilin-Med-VL-Chat [149] | 22.34 | 22.06 | 12.94 | 21.06 | 15.50 | 22.09 | 18.98 | 17.33 | 17.88 | 22.92 | 31.14 | 29.82 | 20.00 | 21.83 | 25.55 | 19.07 | 14.81 | 29.42 | 22.17 | 22.29 |
| RadFM [254] | 22.95 | 22.93 | 24.24 | 23.02 | 20.00 | 20.59 | 20.83 | 19.49 | 28.48 | 24.42 | 18.00 | 32.00 | 16.95 | 26.90 | 26.25 | 18.26 | 26.54 | 25.19 | 23.74 | 20.20 |
| MedDr [95] | 41.95 | 43.69 | 53.18 | 45.28 | 33.00 | 44.78 | 28.03 | 29.91 | 47.68 | 35.22 | 38.29 | 78.55 | 25.08 | 49.53 | 45.31 | 52.09 | 48.61 | 52.36 | 54.21 | 39.90 |
| *Open-Source LVLMs* | | | | | | | | | | | | | | | | | | | | |
| CogVLM-grounding-generalist [249] | 5.20 | 5.66 | 6.59 | 7.27 | 4.50 | 4.94 | 3.58 | 4.44 | 5.96 | 2.66 | 19.14 | 17.82 | 7.80 | 7.94 | 5.00 | 5.36 | 5.40 | 7.86 | 4.59 | 2.34 |
| XComposer [266] | 8.92 | 7.67 | 13.18 | 2.71 | 5.00 | 5.33 | 4.35 | 10.88 | 3.31 | 6.40 | 4.00 | 25.09 | 6.44 | 9.15 | 9.95 | 8.91 | 4.01 | 8.11 | 9.87 | 5.54 |
| PandaGPT 13B [234] | 16.69 | 16.27 | 17.41 | 12.70 | 17.00 | 17.20 | 12.68 | 15.42 | 23.84 | 14.70 | 14.86 | 10.55 | 8.81 | 14.29 | 24.75 | 16.26 | 17.13 | 18.07 | 12.07 | 13.92 |
| Flamingo v2 [17] | 25.58 | 26.34 | 28.47 | 26.06 | 18.50 | 28.58 | 21.11 | 24.24 | 29.14 | 28.07 | 13.43 | 29.45 | 22.37 | 28.17 | 31.85 | 23.12 | 27.78 | 23.54 | 27.57 | 29.19 |
| VisualGLM-6B [61] | 29.58 | 30.45 | 52.71 | 25.95 | 14.00 | 31.69 | 22.06 | 25.17 | 30.46 | 25.50 | 30.29 | 59.27 | 15.93 | 29.97 | 37.79 | 30.09 | 23.61 | 32.85 | 38.19 | 23.03 |
| Idefics-9B-Instruct [137] | 29.74 | 31.13 | 19.76 | 33.98 | 21.00 | 30.08 | 24.46 | 26.66 | 50.33 | 28.74 | 36.00 | 58.55 | 36.27 | 29.64 | 36.76 | 36.07 | 24.38 | 31.36 | 32.04 | 29.19 |
| InstructBLIP-7B [56] | 31.80 | 30.95 | 27.06 | 28.99 | 17.50 | 34.24 | 21.78 | 25.84 | 43.05 | 29.15 | 19.14 | 53.09 | 27.46 | 28.64 | 31.99 | 34.58 | 30.25 | 30.76 | 41.09 | 31.28 |
| Mini-Gemini-7B [141] | 32.17 | 31.09 | 34.59 | 39.63 | 23.50 | 35.74 | 23.46 | 19.80 | 41.06 | 25.91 | 40.86 | 56.00 | 19.32 | 21.63 | 35.73 | 35.83 | 33.95 | 40.57 | 29.14 | 29.56 |
| MMAlaya [154] | 32.19 | 32.30 | 71.06 | 37.68 | 38.00 | 28.30 | 27.40 | 27.64 | 51.66 | 32.39 | 28.86 | 83.64 | 29.49 | 27.37 | 35.92 | 36.70 | 20.99 | 27.53 | 29.43 | 28.08 |
| Qwen-VL [19] | 34.80 | 36.05 | 39.53 | 41.59 | 40.50 | 28.69 | 20.74 | 26.77 | 45.03 | 28.82 | 56.57 | 73.09 | 39.32 | 41.39 | 39.23 | 43.36 | 33.64 | 35.74 | 45.15 | 42.73 |
| Yi-VL-6B [7] | 34.82 | 34.31 | 39.76 | 43.76 | 56.00 | 27.30 | 25.91 | 27.23 | 45.70 | 32.56 | 44.29 | 65.45 | 47.46 | 36.38 | 39.00 | 35.39 | 25.46 | 29.77 | 39.06 | 35.22 |
| LLaVA-NeXT-vicuna-7B [147] | 34.86 | 35.42 | 40.00 | 37.13 | 51.60 | 31.82 | 29.15 | 26.18 | 49.01 | 31.06 | 32.94 | 65.33 | 28.44 | 35.98 | 43.21 | 38.71 | 26.87 | 40.02 | 36.47 | 32.36 |
| Qwen-VL-Chat [19] | 35.07 | 36.96 | 36.47 | 39.63 | 36.50 | 27.08 | 20.79 | 27.64 | 60.93 | 30.23 | 52.57 | 70.55 | 37.29 | 47.13 | 39.37 | 46.67 | 34.57 | 37.63 | 47.88 | 39.90 |
| CogVLM-Chat [249] | 35.23 | 36.08 | 30.59 | 38.98 | 42.50 | 31.41 | 26.22 | 23.62 | 47.02 | 34.22 | 51.43 | 56.00 | 32.54 | 44.13 | 38.67 | 37.94 | 30.86 | 41.11 | 45.91 | 29.19 |
| Monkey [142] | 35.48 | 36.39 | 38.59 | 39.52 | 35.00 | 29.74 | 20.97 | 25.73 | 52.98 | 28.90 | 48.29 | 68.00 | 34.24 | 41.46 | 40.78 | 45.23 | 31.79 | 39.27 | 45.91 | 42.49 |
| mPLUG-Owl2 [259] | 35.62 | 36.21 | 47.76 | 40.50 | 41.00 | 33.46 | 27.22 | 28.16 | 51.66 | 33.14 | 38.86 | 68.73 | 16.27 | 38.58 | 43.34 | 35.70 | 27.78 | 41.61 | 39.76 | 30.91 |
| ShareCaptioner [43] | 36.37 | 36.19 | 37.88 | 35.50 | 45.50 | 35.63 | 25.54 | 28.16 | 56.29 | 31.15 | 27.14 | 64.00 | 35.59 | 38.52 | 39.65 | 38.57 | 30.56 | 44.05 | 36.68 | 40.15 |
| Emu2-Chat [237] | 36.50 | 37.59 | 27.53 | 35.83 | 27.50 | 34.41 | 28.49 | 29.35 | 60.26 | 36.63 | 34.00 | 64.73 | 28.81 | 44.79 | 43.20 | 37.69 | 37.50 | 41.86 | 43.18 | 35.34 |
| XComposer2-4KHD [63] | 36.66 | 38.54 | 48.00 | 40.17 | 75.50 | 36.46 | 28.80 | 28.11 | 49.67 | 35.96 | 50.29 | 69.45 | 38.64 | 40.45 | 43.86 | 39.63 | 29.94 | 43.26 | 34.13 | 42.86 |
| ShareGPT4V-7B [43] | 36.71 | 36.70 | 43.76 | 39.09 | 48.50 | 37.24 | 27.90 | 23.88 | 49.01 | 30.40 | 46.29 | 60.73 | 29.15 | 44.06 | 44.56 | 37.57 | 30.40 | 38.03 | 35.98 | 36.95 |
| LLaVA-NeXT-mistral-7B [147] | 37.20 | 37.16 | 42.96 | 40.17 | 46.40 | 37.84 | 28.53 | 23.76 | 52.32 | 31.81 | 46.59 | 73.00 | 21.25 | 47.08 | 42.61 | 33.37 | 22.75 | 46.94 | 37.45 | 33.48 |
| LLAVA-V1.5-13b-xtuner [54] | 37.82 | 38.74 | 43.06 | 39.20 | 43.50 | 42.01 | 26.36 | 26.41 | 48.34 | 35.55 | 38.29 | 70.55 | 38.64 | 51.60 | 42.08 | 34.70 | 34.41 | 43.90 | 39.35 | 41.26 |
| OmniLMM-12B [261] | 37.89 | 39.30 | 39.53 | 37.46 | 41.50 | 36.18 | 27.36 | 28.00 | 60.93 | 37.46 | 55.43 | 80.00 | 31.19 | 35.71 | 44.89 | 42.49 | 28.24 | 43.80 | 51.19 | 42.86 |
| InternVL-Chat-V1.1 [47] | 38.16 | 39.41 | 45.88 | 40.07 | 56.00 | 34.30 | 26.68 | 26.20 | 52.32 | 37.79 | 45.14 | 64.00 | 35.93 | 52.74 | 44.14 | 40.56 | 39.51 | 41.16 | 45.56 | 35.84 |
| LLAVA-V1.5-7B [148] | 38.23 | 39.46 | 42.35 | 37.57 | 44.50 | 36.13 | 27.99 | 24.91 | 49.01 | 31.31 | 34.00 | 68.36 | 27.12 | 45.39 | 42.46 | 42.80 | 33.80 | 44.20 | 41.21 | 38.92 |
| Monkey-Chat [142] | 38.39 | 39.50 | 43.53 | 40.28 | 40.00 | 33.30 | 23.28 | 29.09 | 54.97 | 29.73 | 55.71 | 72.36 | 35.25 | 50.53 | 42.41 | 45.98 | 33.49 | 42.66 | 50.15 | 44.83 |
| LLAVA-V1.5-7B-xtuner [54] | 38.68 | 38.22 | 51.53 | 35.07 | 31.00 | 38.07 | 31.52 | 29.58 | 58.94 | 36.79 | 28.29 | 69.09 | 29.15 | 50.80 | 39.89 | 40.12 | 27.78 | 40.82 | 39.12 | 36.08 |
| XComposer2 [62] | 38.68 | 39.20 | 32.71 | 42.13 | 70.50 | 33.13 | 29.62 | 27.02 | 54.30 | 34.05 | 23.14 | 83.64 | 39.66 | 46.53 | 44.23 | 45.73 | 28.86 | 45.55 | 41.32 | 41.87 |
| LLAVA-InternLM-7b [54] | 38.71 | 39.11 | 44.94 | 39.85 | 33.50 | 43.06 | 27.54 | 27.08 | 52.98 | 34.22 | 31.14 | 79.64 | 37.97 | 50.67 | 42.41 | 39.69 | 36.73 | 37.63 | 46.72 | 39.78 |
| TransCore-M [3] | 38.86 | 38.70 | 39.06 | 43.87 | 24.50 | 40.18 | 29.08 | 30.79 | 52.98 | 32.48 | 38.86 | 66.91 | 42.37 | 42.79 | 44.75 | 40.44 | 36.73 | 34.00 | 47.19 | 35.71 |
| InternVL-Chat-V1.5 [46] | 38.86 | 39.73 | 36.47 | 44.84 | 53.50 | 37.07 | 26.63 | 31.61 | 60.26 | 34.14 | 36.29 | 67.27 | 37.63 | 55.21 | 47.13 | 38.69 | 41.98 | 39.17 | 37.55 | 41.26 |
| InternVL-Chat-V1.2-Plus [47] | 39.41 | 40.79 | 51.06 | 43.54 | 60.00 | 39.07 | 29.39 | 31.82 | 50.99 | 37.54 | 54.00 | 79.64 | 30.17 | 50.87 | 43.72 | 37.88 | 36.88 | 42.61 | 43.53 | 38.55 |
| InternVL-Chat-V1.2 [47] | 39.52 | 40.01 | 40.71 | 46.25 | 77.50 | 31.52 | 26.36 | 31.10 | 50.33 | 36.96 | 52.00 | 80.00 | 31.19 | 45.46 | 43.20 | 40.06 | 34.10 | 44.40 | 46.66 | 42.36 |
| LLAVA-InternLM2-7b [54] | 40.07 | 40.45 | 43.53 | 40.72 | 60.50 | 34.74 | 30.12 | 27.44 | 51.66 | 33.39 | 50.86 | 74.55 | 26.44 | 49.13 | 42.74 | 43.12 | 31.94 | 50.87 | 47.01 | 39.04 |
| DeepSeek-VL-1.3B [155] | 40.25 | 40.77 | 56.71 | 37.13 | 27.00 | 45.73 | 28.40 | 27.85 | 52.32 | 35.96 | 45.43 | 71.64 | 45.42 | 50.20 | 41.66 | 47.48 | 37.81 | 43.90 | 45.50 | 33.50 |
| MiniCPM-V [103] | 40.95 | 41.05 | 28.47 | 42.02 | 40.00 | 42.79 | 28.80 | 28.62 | 46.36 | 36.30 | 40.00 | 67.27 | 31.53 | 42.46 | 44.04 | 50.28 | 37.50 | 51.92 | 52.29 | 27.22 |
| DeepSeek-VL-7B [155] | 41.73 | 43.43 | 60.00 | 43.97 | 47.50 | 45.12 | 28.22 | 31.20 | 46.36 | 32.97 | 52.29 | 67.64 | 61.36 | 49.27 | 44.23 | 49.97 | 52.78 | 45.00 | 53.63 | 38.79 |
| MiniCPM-V2 [257] | 41.79 | 42.54 | 37.88 | 43.65 | 35.50 | 42.67 | 26.49 | 29.24 | 37.75 | 33.31 | 59.71 | 67.27 | 38.64 | 50.87 | 42.64 | 50.59 | 40.90 | 51.07 | 57.81 | 35.10 |
| *Proprietary LVLMs* | | | | | | | | | | | | | | | | | | | | |
| Claude3-Opus [13] | 32.37 | 32.44 | 38.59 | 34.42 | 43.50 | 27.97 | 22.96 | 23.62 | 52.32 | 25.42 | 25.14 | 66.91 | 15.93 | 35.25 | 41.06 | 36.07 | 37.50 | 40.67 | 35.40 | 34.24 |
| Qwen-VL-Max [18] | 41.34 | 42.16 | 50.59 | 47.23 | 74.00 | 40.68 | 29.03 | 26.71 | 58.94 | 34.05 | 62.29 | 85.45 | 27.80 | 44.39 | 43.90 | 42.99 | 48.61 | 49.38 | 51.13 | 40.52 |
| GPT-4V [5] | 42.50 | 44.08 | 64.00 | 44.95 | 58.50 | 42.45 | 30.03 | 29.40 | 58.28 | 32.31 | 54.57 | 83.27 | 37.63 | 48.26 | 49.04 | 48.41 | 44.60 | 51.87 | 53.98 | 40.89 |
| Gemini 1.0 [240] | 44.38 | 44.93 | 57.41 | 46.25 | 57.50 | 36.40 | 28.67 | 27.80 | 45.03 | 38.21 | 58.57 | 86.55 | 40.68 | 51.74 | 47.45 | 55.64 | 50.46 | 47.83 | 61.58 | 41.87 |
| Gemini 1.5 [211] | 47.42 | 48.36 | 55.29 | 50.81 | 54.00 | 51.05 | 36.59 | 29.86 | 56.95 | 36.88 | 58.00 | 88.00 | 47.46 | 48.13 | 51.19 | 56.88 | 64.51 | 56.50 | 59.78 | 31.65 |
| GPT-4o [5] | 53.53 | 53.96 | 66.82 | 48.53 | 64.50 | 55.94 | 35.10 | 48.53 | 74.17 | 43.52 | 64.57 | 91.64 | 37.63 | 57.88 | 55.21 | 62.80 | 66.98 | 58.39 | 64.60 | 46.18 |

Table 12: Results for single-choice questions of 50 LVLMs on perceptual granularities. The best-performing model in each category is **in-bold**, and the second best is underlined.

| Model name | Size | Overall(val) | Overall(test) | Seg C | Seg M | 2D Cls update | 2D Det | 2D Mcls_acc | 2D Mcls_recall |
|---|---|---|---|---|---|---|---|---|---|
| Random | - | 25.70 | 25.88 | 22.19 | 22.91 | 28.93 | 24.55 | 45.85 | 57.02 |
| Medical Special Model | | | | | | | | | |
| MedVInT [268] | - | 2.29 | 1.98 | 0.82 | 0.25 | 3.48 | 0.12 | 0.05 | 0.02 |
| Med-Flamingo [181] | 8.3B | 12.74 | 11.75 | 11.95 | 11.94 | 11.92 | 9.15 | 46.10 | 50.19 |
| LLaVA-Med [138] | - | 20.54 | 19.83 | 18.45 | 18.97 | 21.15 | 17.14 | 45.84 | 41.19 |
| Qilin-Med-VL-Chat [149] | - | 22.34 | 22.06 | 19.84 | 20.30 | 23.80 | 21.87 | 44.50 | 33.90 |
| RadFM [254] | 14B | 22.95 | 22.93 | 20.43 | 20.27 | 25.71 | 18.83 | 40.98 | 57.45 |
| MedDr [95] | 40B | 41.95 | 43.18 | 42.55 | 44.03 | 45.08 | 28.10 | 48.09 | 23.38 |
| Open-Source LVLMs | | | | | | | | | |
| CogVLM-grounding-generalist [249] | 17B | 5.20 | 5.39 | 6.80 | 5.51 | 5.11 | 2.57 | 46.24 | 49.82 |
| XComposer [266] | 8B | 8.92 | 7.71 | 8.87 | 6.24 | 8.02 | 6.30 | 31.45 | 23.68 |
| PandaGPT 13B [234] | 13B | 16.69 | 15.94 | 19.25 | 18.88 | 13.74 | 12.24 | 41.22 | 49.95 |
| Flamingo v2 [17] | 9B | 25.58 | 26.23 | 22.52 | 22.48 | 30.12 | 21.17 | 41.80 | 19.17 |
| VisualGLM-6B [61] | 7.8B | 29.58 | 30.20 | 27.30 | 27.31 | 33.75 | 22.16 | 43.08 | 35.22 |
| Idefics-9B-Instruct [137] | 9B | 29.74 | 30.81 | 25.50 | 25.21 | 36.45 | 23.85 | 43.47 | 46.02 |
| InstructBLIP-7B [56] | 8B | 31.80 | 31.00 | 29.12 | 21.77 | 36.71 | 24.08 | 39.43 | 23.79 |
| Mini-Gemini-7B [141] | 7B | 32.17 | 31.22 | 32.13 | 32.92 | 30.72 | 26.53 | 45.38 | 57.99 |
| MMAlaya [154] | 7.8B | 32.19 | 32.02 | 29.33 | 30.22 | 35.02 | 24.02 | 48.43 | 20.93 |
| Qwen-VL [19] | 9.6B | 34.80 | 35.55 | 33.20 | 33.43 | 38.95 | 24.49 | 44.95 | 56.97 |
| Yi-VL-6B [7] | 6.6B | 34.82 | 34.00 | 31.42 | 32.26 | 37.15 | 24.31 | 50.25 | 44.32 |
| LLaVA-NeXT-vicuna-7B [147] | 7.1B | 34.86 | 35.59 | 33.06 | 32.95 | 38.96 | 27.06 | 44.75 | 42.45 |
| Qwen-VL-Chat [19] | 9.6B | 35.07 | 36.35 | 34.45 | 35.20 | 39.55 | 22.04 | 42.88 | 81.23 |
| CogVLM-Chat [249] | 17B | 35.23 | 35.83 | 34.13 | 34.49 | 38.55 | 25.25 | 47.09 | **90.26** |
| Monkey [142] | 9.8B | 35.48 | 35.92 | 33.18 | 34.01 | 39.32 | 25.42 | 44.57 | 42.35 |
| mPLUG-Owl2 [259] | 8.2B | 35.62 | 35.89 | 33.68 | 34.74 | 38.80 | 24.90 | 42.59 | 41.84 |
| ShareCaptioner [43] | 8B | 36.37 | 36.07 | 34.74 | 35.93 | 38.25 | 24.37 | 40.00 | 16.95 |
| Emu2-Chat [237] | 37B | 36.50 | 35.54 | 36.54 | 27.62 | 39.57 | 27.76 | 44.29 | 37.65 |
| XComposer2-4KHD [63] | 7B | 36.66 | 37.93 | 36.84 | 38.02 | 39.84 | 26.65 | 48.83 | 44.08 |
| ShareGPT4V-7B [43] | 7.2B | 36.71 | 36.52 | 34.74 | 35.15 | 39.24 | 26.18 | 46.11 | 43.52 |
| LLaVA-NeXT-mistral-7B [147] | 7.6B | 37.20 | 37.02 | 36.29 | 35.20 | 39.34 | 27.87 | 44.05 | 47.70 |
| LLAVA-V1.5-13b-xtuner [54] | 13.4B | 37.82 | 38.27 | 38.29 | 36.95 | 40.48 | 25.83 | 47.54 | 33.19 |
| OmniLMM-12B [261] | 12B | 37.89 | 38.74 | 36.70 | 36.86 | 41.77 | 28.57 | 46.17 | 43.01 |
| InternVL-Chat-V1.1 [47] | 19B | 38.16 | 38.93 | 38.54 | 40.00 | 40.07 | 28.16 | 39.82 | 27.32 |
| LLAVA-V1.5-7B [148] | 7.2B | 38.23 | 37.72 | 36.45 | 36.65 | 40.38 | 25.36 | 14.10 | 57.09 |
| Monkey-Chat [142] | 9.8B | 38.39 | 39.00 | 37.16 | 37.75 | 42.13 | 25.36 | 43.91 | 28.86 |
| LLAVA-V1.5-7B-xtuner [54] | 7.2B | 38.68 | 37.96 | 36.75 | 36.34 | 40.55 | 27.52 | 46.78 | 43.06 |
| XComposer2 [62] | 7B | 38.68 | 38.95 | 37.86 | 38.52 | 41.00 | 28.34 | 46.43 | 51.87 |
| LLAVA-InternLM-7b [54] | 7.6B | 38.71 | 38.84 | 37.57 | 36.65 | 41.84 | 27.46 | 50.02 | 40.21 |
| TransCore-M [3] | 13.4B | 38.86 | 38.43 | 36.09 | 36.06 | 42.04 | 26.53 | 45.34 | 40.93 |
| InternVL-Chat-V1.5 [46] | 25.5B | 38.86 | 39.32 | 38.61 | 40.48 | 40.45 | 29.27 | 31.51 | 24.72 |
| InternVL-Chat-V1.2-Plus [47] | 40B | 39.41 | 40.25 | 40.68 | 41.50 | 40.82 | 30.38 | 36.50 | 37.09 |
| InternVL-Chat-V1.2 [47] | 40B | 39.52 | 39.57 | 39.04 | 39.75 | 41.05 | 29.62 | 41.08 | 46.06 |
| LLAVA-InternLM2-7b [54] | 8.1B | 40.07 | 40.15 | 39.30 | 39.14 | 42.60 | 27.76 | **50.64** | 48.25 |
| DeepSeek-VL-1.3B [155] | 1.3B | 40.25 | 40.54 | 40.61 | 40.71 | 42.13 | 27.64 | 48.71 | 21.38 |
| MiniCPM-V [103] | 2.8B | 40.95 | 40.89 | 39.48 | 39.18 | 44.08 | 27.00 | 42.87 | 32.09 |
| DeepSeek-VL-7B [155] | 7.3B | 41.73 | 42.90 | 43.87 | 43.60 | 44.32 | 26.59 | 44.16 | 18.74 |
| MiniCPM-V2 [257] | 2.8B | 41.79 | 42.13 | 41.11 | 41.41 | 45.03 | 25.95 | 50.12 | 32.62 |
| Proprietary LVLMs | | | | | | | | | |
| Claude3-Opus [13] | - | 32.37 | 32.24 | 33.56 | 33.36 | 32.17 | 24.72 | 45.31 | 38.98 |
| Qwen-VL-Max [18] | - | 41.34 | 41.70 | 44.23 | 44.42 | 41.09 | 29.10 | 31.12 | 25.88 |
| GPT-4V [5] | - | 42.50 | 43.61 | 47.87 | 46.58 | 42.24 | 30.32 | 45.21 | 40.59 |
| Gemini 1.0 [240] | - | 44.38 | 44.65 | 44.92 | 44.96 | 46.67 | 27.46 | 49.01 | 55.09 |
| Gemini 1.5 [211] | - | 47.42 | 48.03 | 54.75 | **56.59** | 43.25 | 34.17 | 39.22 | 39.34 |
| GPT-4o [5] | - | **53.53** | **53.88** | **57.09** | 56.49 | **53.70** | **36.21** | 50.60 | 50.90 |

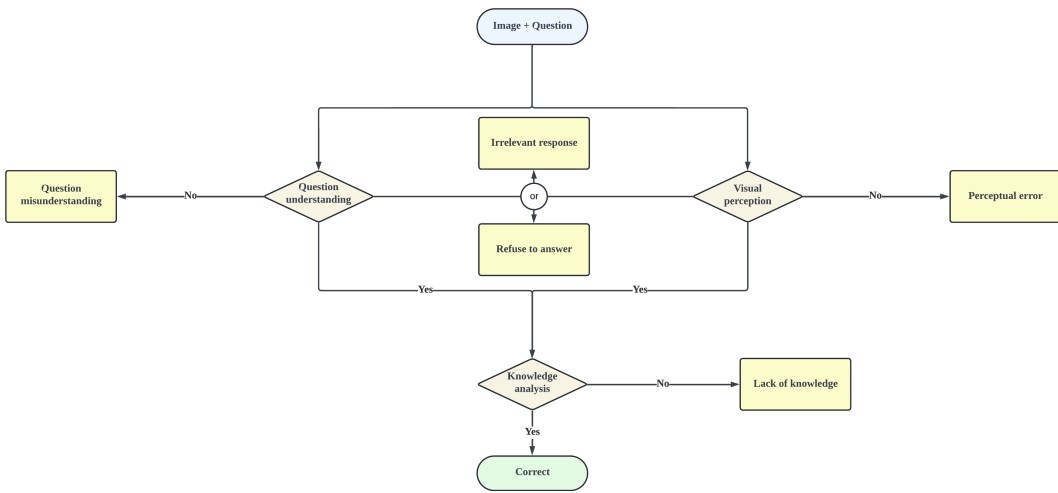

Figure 9: The illustration of the entire logical process from input to output in our case study.

## E.2 Case Study

In this section, we present a case study analysis of several LVLMs on various cases in GMAI-MMBench. The entire logical process of our study is illustrated in Figure 9. Other than **Correct**, we classify the error types from input to output into five major categories:

**Correct:** LVLMs offer the correct answer. This indicates that the model accurately understands both the image and the question, and provides an appropriate and relevant response.

**Question misunderstanding:** LVLMs fail to correctly understand the question and generate erroneous answers. For example: LLAVA-Med may not understand the purpose of identifying the surgical process from the question, instead, it describes the image content in detail as shown in Figure 27.

**Perceptual error:** LVLMs fail to locate, detect, or recognize the content or objects in images, which are necessary for answering the questions. This includes scenarios where the model misses critical details or misinterprets the image's content. For example: GPT-4o may ignore the important tool in the lower left corner that is clearing the debris in Figure 32. Claude3-Opus chooses the wrong answer as it cannot correctly identify the content in the mask in Figure 38.

**Lack of knowledge:** LVLMs can recognize both the image and the question but still make errors in specific cases, suggesting a lack of domain-specific knowledge required to answer specialized questions. For example: Models directly show their insufficient knowledge to answer or fail to respond without additional information as shown in Figure 52, Figure 54, Figure 52, etc. Another case in Figure 51 shows that GPT-4o correctly describes the image and understands the question but still chooses a wrong answer, suggesting it may lack the ability to distinguish between carcinoma in situ and invasive carcinoma.

**Irrelevant response:** LVLMs do not address the question directly and produce unreadable or unrelated responses. This problem is especially noticeable in open-source models. For example: RadFM only generates a reference paper without any additional outputs in Figure 57.

**Refuse to answer:** LVLMs decline to answer certain questions to keep the system safe for all users, such as those involving sensitive or ethical issues, and refuse to provide medical advice when they determine that human professional assistance is required. This issue only occurs in proprietary models like GPTs and Claudes.

In our test, we randomly select 53 VQA pairs from different clinical VQA tasks, departments, and perceptual granularities. All cases are listed in Table 13. Based on our observations of the evaluation results, we find that proprietary models like GPT-4o and Claude3-Opus rarely encounter difficulties in question understanding. The majority of errors for these models stem from perceptual error and lack of knowledge. In contrast, specialized medical models such as RadFM and LLAVA-Med frequently exhibit language understanding errors, making it difficult to effectively evaluate visual perceptual abilities. As a result, the case study indicates that general models need to enhance their performance

on specialized medical images, which may require more medical data for training. Meanwhile, specialized medical models need further training or fine-tuning in language aspects.

Table 13: Table index of our case study figures.

| Figure | Clinical VQA task | Department | Perceptual granularity | Category |
|---|---|---|---|---|
| 10 | MR | H | Image Level | Correct |
| 11 | C | H | Image Level | Correct |
| 12 | SWR | ENT | Image Level | Correct |
| 13 | DD | GH | Image Level | Correct |
| 14 | ASR | NH | Image Level | Correct |
| 15 | SAR | U | Box Level | Correct |
| 16 | DD | PM | Box Level | Correct |
| 17 | OR-NH | E | Mask Level | Correct |
| 18 | OR-P | U | Contour Level | Correct |
| 19 | SIR | GS | Box Level | Correct |
| 20 | BVR | H | Mask Level | Correct |
| 21 | CR | H | Box Level | Correct |
| 22 | DD | CS | Mask Level | Correct |
| 23 | DD | OS | Contour Level | Correct |
| 24 | NT | O | Mask Level | Correct |
| 25 | OR-T | PM | Mask Level | Correct |
| 26 | SIR | GS | Mask Level | Correct |
| 27 | SWR | GS | Image Level | Question misunderstanding |
| 28 | BVR | O | Mask Level | Question misunderstanding |
| 29 | ACR | OS | Mask Level | Question misunderstanding |
| 30 | MR | GH | Image Level | Question misunderstanding |
| 31 | C | H | Image Level | Perceptual error |
| 32 | SWR | GS | Image Level | Perceptual error |
| 33 | OR-T | PM | Mask Level | Perceptual error |
| 34 | AR | LMP | Image Level | Perceptual error |
| 35 | NT | N | Mask Level | Perceptual error |
| 36 | DD | CS | Box Level | Perceptual error |
| 37 | DD | D | Mask Level | Perceptual error |
| 38 | DD | GH | Contour Level | Perceptual error |
| 39 | OR-T | PM | Mask Level | Perceptual error |
| 40 | NT | N | Mask Level | Perceptual error |
| 41 | OR-T | PM | Contour Level | Perceptual error |
| 42 | DD | O | Image Level | Lack of knowledge |
| 43 | IQG | O | Image Level | Lack of knowledge |
| 44 | MR | LMP | Image Level | Lack of knowledge |
| 45 | SAR | GS | Box Level | Lack of knowledge |
| 46 | SAR | U | Box Level | Lack of knowledge |
| 47 | DD | PM | Mask Level | Lack of knowledge |
| 48 | NT | O | Mask Level | Lack of knowledge |
| 49 | SG | LMP | Image Level | Lack of knowledge |
| 50 | DD | O | Image Level | Lack of knowledge |
| 51 | SG | LMP | Image Level | Lack of knowledge |
| 52 | DD | OM | Image Level | Lack of knowledge |
| 53 | AR | GS | Image Level | Lack of knowledge |
| 54 | AR | OG | Image Level | Lack of knowledge |
| 55 | DD | D | Image Level | Lack of knowledge |
| 56 | DD | U | Image Level | Lack of knowledge |
| 57 | DD | OS | Image Level | Irrelevant response |
| 58 | AR | ID | Image Level | Irrelevant response |
| 59 | AR | OS | Image Level | Irrelevant response |
| 60 | ASR | OG | Image Level | Irrelevant response |
| 61 | DD | PM | Image Level | Refuse to answer |
| 62 | BVR | O | Mask Level | Refuse to answer |

Question: Identify the option that the content of the microscopic image corresponds to.

Options:
A.  basophil
B.  hairy cell
C.  proerythroblast
D.  band neutrophil
Please select the correct answer from the options above

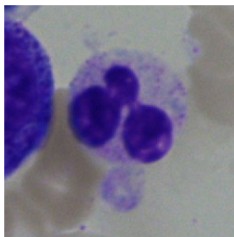

GPT-4o Response :
The image illustrates a basophil. So, the correct option is :
A.  basophil
Granular Appearance: Basophils are characterized by large granules that stain dark purple or blue with standard stains like Wright or Giemsa stain. The image shows a cell with prominent granules.
Lobed Nucleus: Basophils typically have a lobed nucleus, which may sometimes be obscured by the granules. The nucleus appears to be lobed in this image, although it is partially covered by granules.
Size and Shape: The size and shape of the cell in the image match the typical morphology of a basophil, which is generally larger than red blood cells and has a distinct granular cytoplasm.
These characteristics match those of a basophil, confirming that the image is indeed illustrating a basophil.

Figure 10: An example of correct case. Green highlights the right answer. Back to Table Index.

## Correct

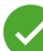

Question: From the microscopic image input, please count and tell me the number of lymphocytes shown.

Options：
A.  the number of lymphocytes:2
B.  the number of lymphocytes:12
C.  the number of lymphocytes:7
D.  the number of lymphocytes:10
Please select the correct answer from the options above

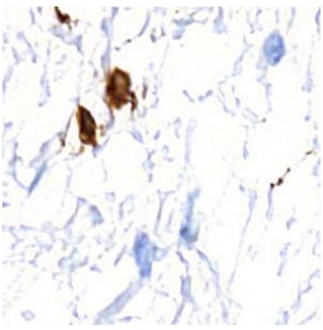

GPT-4o Response :
To determine the number of lymphocytes in the provided image, I will carefully examine and count the cells that fit the typical characteristics of lymphocytes: small size, round shape, and dense, round nucleus.
Upon examining the image closely, the number of lymphocytes visible in the image is :
A. the number of lymphocytes: 2

Figure 11: An example of correct case. Green highlights the right answer. Back to Table Index.

## Correct 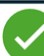

Question: Please identify which of the following options the surgical procedure shown in the endoscopic image belongs to.

Options:
A.   synthetic graft placement
B.   debris clearance
C.   nasal corridor creation
D.   operation not started or operation ended\tor out of patient
Please select the correct answer from the options above

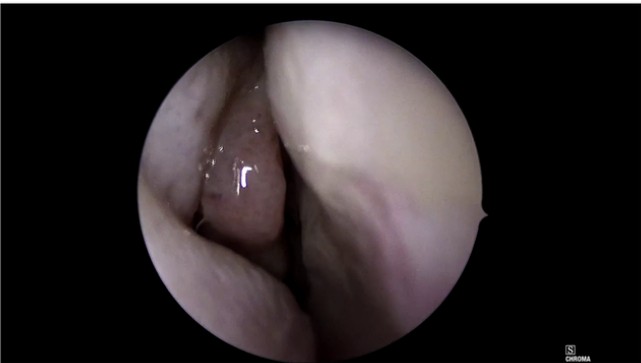

GPT-4o Response :
The image appears to show an endoscopic view of the nasal cavity. Based on the visual characteristics and the provided options, the most suitable answer is :
C. nasal corridor creation
Anatomical Structures: The image shows structures consistent with the nasal cavity, which include the nasal septum and turbinate. This setting is typical for procedures involving the nasal passages.
Procedure Context: Creating a nasal corridor is a common part of endoscopic sinus surgery or procedures that require access to the nasal passages. This involves clearing and creating pathways within the nasal cavity.

Figure 12: An example of correct case. Green highlights the right answer. Back to Table Index.

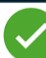

Question: Observe the image. What is the most likely abnormality shown in this endoscopic picture?

Options：
A. ulcerative colitis
B. z line
C. dyed resection margins
D. polyp
Please select the correct answer from the options above

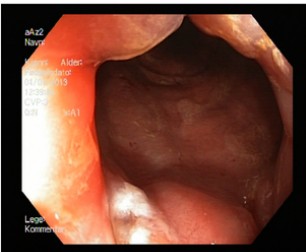

Model Response :
The image appears to be an endoscopic view of the gastrointestinal tract. Based on the visual characteristics and the provided options, the most likely abnormality shown in the picture is :
A. ulcerative colitis

Mucosal Appearance: The image shows areas of mucosa that appear erythematous (red), with some loss of the normal vascular pattern and possible ulceration or erosion. This is consistent with the inflammatory changes seen in ulcerative colitis.

Distribution: Ulcerative colitis typically affects the colon and rectum, causing continuous areas of inflammation. The diffuse and continuous nature of the mucosal changes in the image supports this diagnosis.

Figure 13: An example of correct case. Green highlights the right answer. Back to Table Index.

## Correct
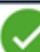

Question: Determine which part is illustrated in this ultrasound image.

Options:
A. aorta
B. kidney
C. thyroid nodule
D. pancreas
Please select the correct answer from the options above

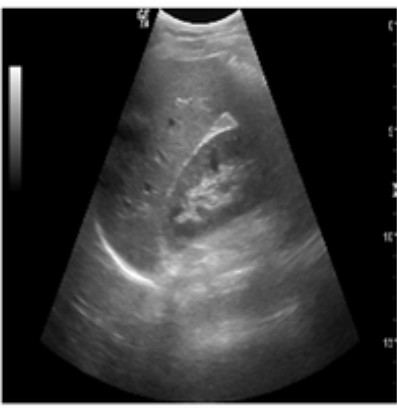

**GPT-4o Response :**
The image is identified as an ultrasound of the kidney for the following reasons:
1. Typical Shape and Structure: The kidney has a characteristic bean shape and appears as a homogenous structure with a distinct outline on an ultrasound. The image shows this familiar shape and internal architecture.
2. Renal Pelvis: In the center of the kidney, there is a more hypoechoic (darker) area which represents the renal pelvis, a common feature in kidney ultrasounds.
3. Surrounding Fat and Musculature: The image shows the kidney surrounded by hyperechoic (brighter) areas which likely represent perinephric fat and adjacent musculature, common anatomical landmarks in renal ultrasound.

So, the correct option is:
B. kidney

Figure 14: An example of correct case. Green highlights the right answer. Back to Table Index.

## Correct 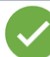

Question: Focusing on the boxed section of this endoscopy video, what surgical action is in progress?

Options:
A. bladder anastomosis
B. cutting vas deferens
C. pulling prostate
D. clipping bladder neck
Please select the correct answer from the options above

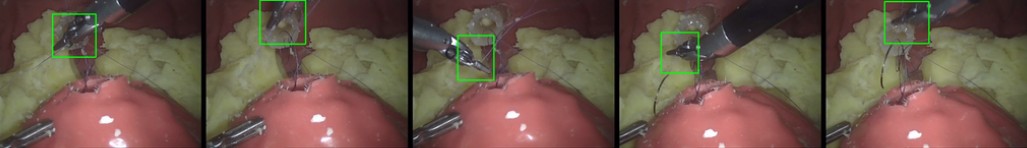

GPT-4o Response :
The sequence of images shows a surgical tool (likely a needle holder) being used to manipulate a needle through tissue. The green boxes highlight the specific action involving the needle and sutures. This action involves stitching or suturing, which is a key indicator for procedures like anastomosis.

Given the context and the nature of the procedure shown (suturing), the most appropriate action described in the options is A. Bladder anastomosis.

Figure 15: An example of correct case. Green highlights the right answer. Back to Table Index.

## Correct 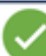

Question: Considering the box-marked region in the X-ray image, what is the most likely diagnosis?

Options:
A.  pulmonary calcification
B.  cardiomegaly
C.  pleural effusion
D.  atelectasis
Please select the correct answer from the options above

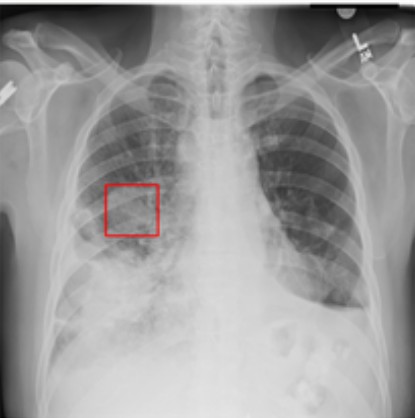

GPT-4o Response :
The correct answer to the question is D. atelectasis.
The box-marked region in the X-ray image shows an area of increased density in the left lower lung field. This appearance is consistent with atelectasis, which is the collapse or closure of a lung resulting in reduced or absent gas exchange. The other options (pulmonary calcification, cardiomegaly, pleural effusion) have different radiographic features. Pulmonary calcification would appear as more distinct, localized, dense spots; cardiomegaly would be characterized by an enlarged cardiac silhouette; and pleural effusion would show a fluid level, often with a meniscus sign, in the costophrenic angles.

Figure 16: An example of correct case. Green highlights the right answer. Back to Table Index.

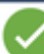

## Correct

Question: This is a CT image. Which of the following options is the most appropriate to describe the marked area?

Options:
A. optic nerve
B. anterior hippocampus
C. thyroid gland
D. temporal lobe
Please select the correct answer from the options above

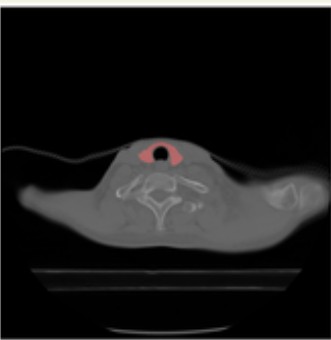

GPT-4o Response :
The marked area in the CT image is located anteriorly in the neck, surrounding the trachea, which corresponds to the anatomical position of the thyroid gland. Other options like the optic nerve, anterior hippocampus, and temporal lobe are not located in the neck region, making C. Thyroid gland the most appropriate answer.

Figure 17: An example of correct case. Green highlights the right answer. 

## Correct

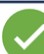

Question: This is a T2 weighted MRI image. Which of the following options is the most appropriate to describe the marked area?

Options:
A. duodenum
B. rectum
C. prostate
D. femoral head
Please select the correct answer from the options above

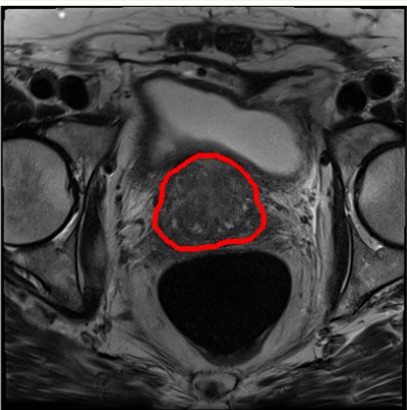

GPT-4o Response :
The correct answer to the question is C. prostate.
The marked area in the T2-weighted MRI image is located in the pelvic region, just below the bladder and in front of the rectum, which is the typical anatomical position of the prostate gland. This is consistent with the high signal intensity (bright appearance) commonly seen in T2-weighted images of the prostate. The other options (duodenum, rectum, femoral head) are located in different regions of the body and do not correspond to the anatomical location of the marked area in the image.

Figure 18: An example of correct case. Green highlights the right answer. Back to Table Index.

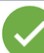

Question: View the endoscopy image provided. What target is displayed in the boxed region?

Options：
A. irrigator
B. clipper
C. bipolar
D. specimen bag
E. hook
Please select the correct answer from the options above

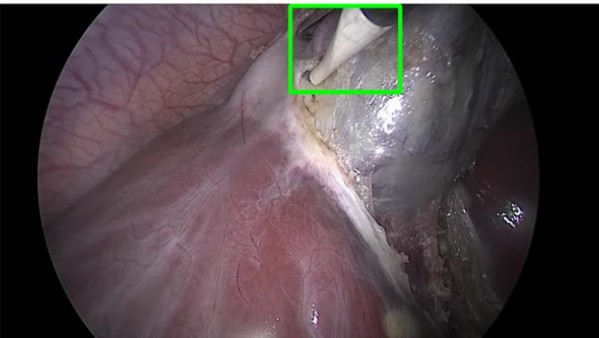

GPT-4o Response :
The boxed region in the image shows a tool that has a distinct curved end, which is characteristic of a hook. Among the options provided, the tool that best matches the shape and appearance of the object in the boxed region is a hook. An irrigator, clipper, bipolar, and specimen bag have distinct and different appearances that do not match the curved end visible in the image. Based on the visual evidence and the shape of the tool within the boxed region, the correct answer is E. Hook. The characteristic curved shape is a definitive feature that identifies it as a hook.

Figure 19: An example of correct case. Green highlights the right answer. Back to Table Index.

## Correct

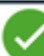

Question: Looking at the marked are in the provided CT image, which option would be the most organ like?

Options:
A. lung vessel
B. renal artery
C. renal vein
D. inferior vena cava"
Please select the correct answer from the options above

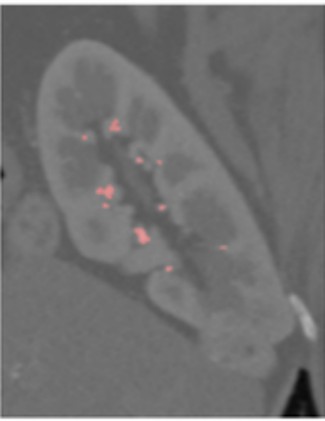

GPT-4o Response :
The CT image shows a cross-section of the kidney, with the marked areas being small, bright structures within the kidney. These structures are typically indicative of arteries, as they appear brighter due to the contrast agent used in CT scans. Given the location and characteristics of the marked areas, they are most likely the renal arteries, which supply blood to the kidneys.
Therefore, the correct answer is B. renal artery.

Figure 20: An example of correct case. Green highlights the right answer. Back to Table Index.

## Correct

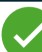

Question: Observe the microscopy image. Can you identify the target within the outlined box?

Options：
A. red blood cell
B. white blood cell
C. platelet
D. mycobacterium tuberculosis
Please select the correct answer from the options above

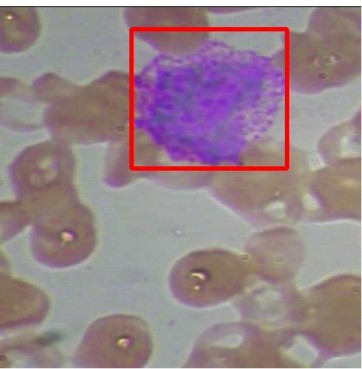

**GPT-4o Response :**
The target within the outlined box is a white blood cell.

White blood cells are typically larger and have a distinct nucleus, which can be seen in the image. They are stained differently compared to red blood cells, which lack a nucleus and appear more uniform and red.

So, the correct answer is:

B. white blood cell

Figure 21: An example of correct case. Green highlights the right answer. .

## Correct

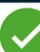

Question: Which of the following options best matches the symptoms of marked region shown in the X-ray image?
Options：
A.  retinal hemorrhages
B.  cardiomegaly
C.  emphysema
D.  barrett's esophagus
Please select the correct answer from the options above

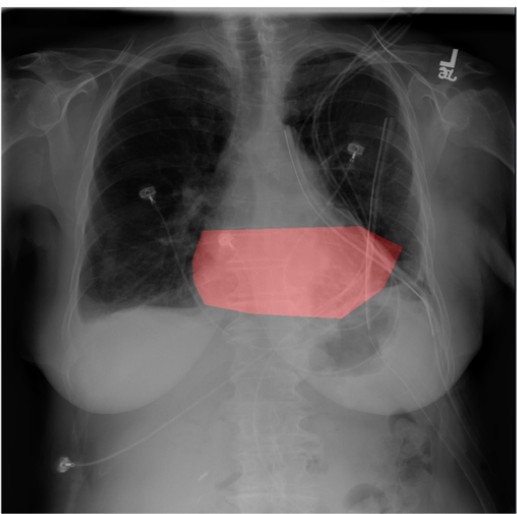

**GPT-4o Response :**
The marked region in the X-ray image is located in the area of the heart. The best match for the symptoms related to this region is:

B. cardiomegaly

Figure 22: An example of correct case. Green highlights the right answer. Back to Table Index.

## Correct 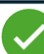

Question: Observe the X-ray image. What is the most likely abnormality shown in the highlight area?
Options：
A.  barrett's esophagus
B.  melanocytic lesions
C.  rib fracture
D.  pigment network skin lesion
E.  emphysema
Please select the correct answer from the options above

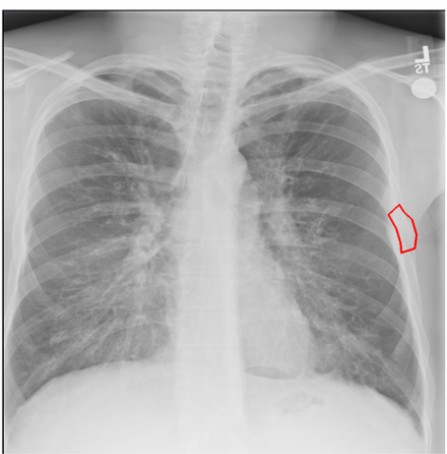

**Claude3-Opus Response :**
The X-ray image shows a chest radiograph. In the highlighted area on the right side of the image, there appears to be a rib fracture. The rib shows an abnormal contour and disruption compared to the normal curving ribs elsewhere in the image. Therefore, based on the options provided, the most likely abnormality shown in the highlighted area is:
C. rib fracture
The other options like Barrett's esophagus, melanocytic lesions, pigment network skin lesions are unrelated to rib anatomy seen on chest X-rays. While emphysema can be evaluated on chest imaging, it affects the lung tissue rather than the ribs.

Figure 23: An example of correct case. Green highlights the right answer. Back to Table Index.

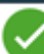

## Correct

Question: Observe the OCT image. Can you identify the organ in the highlight area?

Options:
A.  choroidal layer
B.  ganglion cell-inner plexiform layer
C.  kidney boundary
D.  retinal nerve fiber layer
E.  cell nucleus
Please select the correct answer from the options above

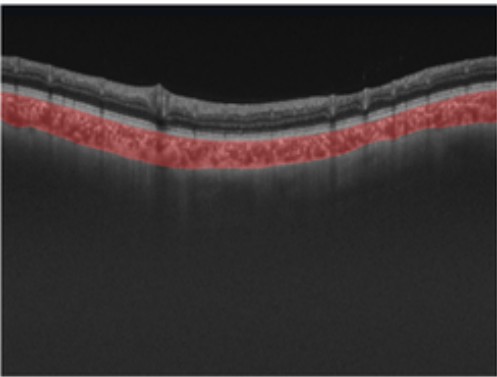

Gemini 1.5 Response:
The correct answer is A. choroidal layer.

The highlighted area in the OCT (Optical Coherence Tomography) image represents the choroid, a layer of tissue that lies between the retina and the sclera (white outer layer of the eye). The choroid is rich in blood vessels that nourish the outer layers of the retina.

The other options are incorrect because they represent different structures within the eye or elsewhere in the body:

B. Ganglion cell-inner plexiform layer: Located within the retina, closer to the center of the eye.
C. Kidney boundary: Not visible in an eye image.
D. Retinal nerve fiber layer: Located within the retina, closer to the center of the eye.
E. Cell nucleus: Too small to be distinguished at this level of magnification.

Figure 24: An example of correct case. Green highlights the right answer. Back to Table Index.

## Correct

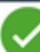

Question: Considering the organ visible marked in the CT picture, which option corresponds most accurately?
Options:
A. pulmonary artery
B. breast
C. lung
D. bronchus
Please select the correct answer from the options above

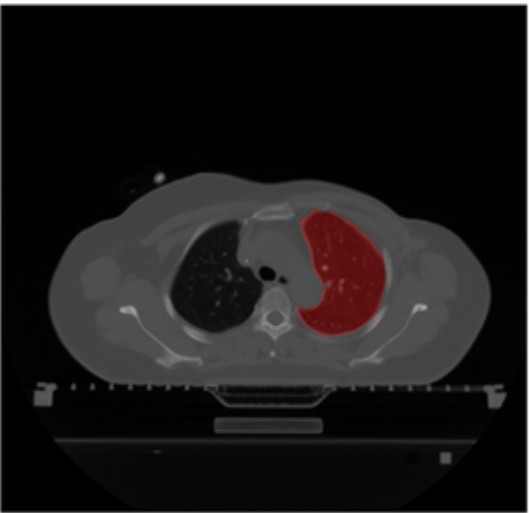

Claude3-Opus Response :
Based on the CT scan image showing cross-sections of the chest cavity, the organ marked in red appears to be the lungs. The distinctive shape and location of the highlighted structures on both sides of the chest strongly indicate they are the right and left lungs.
Therefore, the most accurate answer from the given options is:
C. lung
The other options - pulmonary artery, breast, and bronchus - do not match the anatomy visible in this particular CT slice depicting the lungs.

Figure 25: An example of correct case. Green highlights the right answer. 

## Correct

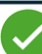

Question: Looking at the marked ares in the provided Endoscopy image, which option would be the most organ like?
Options：
A.  specimen bag
B.  clip
C.  instrument suction
D.  grasper
Please select the correct answer from the options above

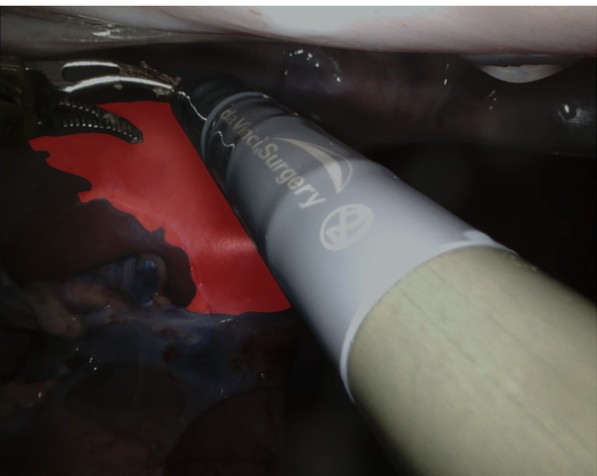

**GPT-4o Response :**

The marked area in the provided endoscopy image appears to be part of an organ. Therefore, the most organ-like option is not listed among the provided options. However, if you are asking which of the options is most likely to be interacting with the organ, the correct answer would be:

C. instrument suction

Figure 26: An example of correct case. Green highlights the right answer. Back to Table Index.

## Question Misunderstanding

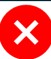

Question: Identify the option that matches the surgical process illustrated in the endoscopic image

Options：
A.  ulcerative colitis
B.  normal z line
C.  polyp
D.  dyed lifted polyps
Please select the correct answer from the options above

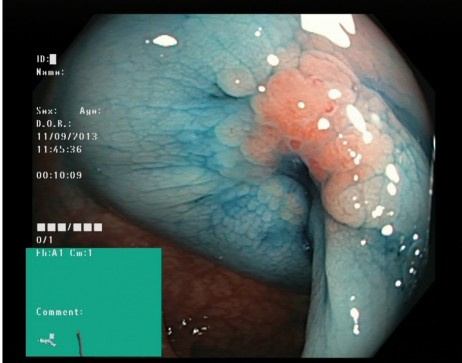

**LLAVA-Med Response :**
The polyp is a growth that forms on the inner lining of an organ, typically as a result of tissue irritation or inflammation. In this case, the polyp was identified during a flexible sigmoidoscopy, which is a type of endoscopy used to examine the rectos

**Error Analysis :**
This response misunderstands the question. It attempts to describe the content of the image instead of following the input instructions to answer the question.

Figure 27: A question misunderstanding example. Green highlights the right answer. Yellow highlights the wrong answer. Back to Table Index.

## Question Misunderstanding

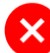

Question: Observe the Fundus Photography image. Can you identify the organ in the highlight area?

Options:
A.  instrument shaft
B.  breast cancer type: invasive ductal carcinoma
C.  retinal vessel
D.  brachial plexus
E.  negative network skin lesion
Please select the correct answer from the options above

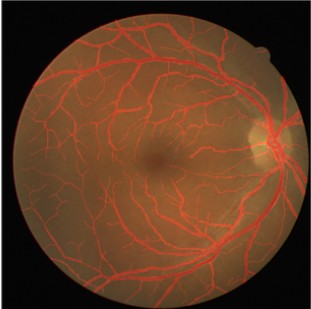

LLaVA-Med Response :
A. instrument shaft
B.brachial plexus
C.retinal vessel
D.negative network skin lesion
E.invasive ductal carcinoma

Error Analysis :
LLaVA-Med provided an answer with incorrectly listed options, indicating a failure to match the given options accurately.

Figure 28: A question misunderstanding example. Green highlights the right answer. Yellow highlights the wrong answer. Back to Table Index.

## Question Misunderstanding

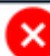

Question: Observe the CT image. Can you identify the organ in the highlight area?
Options:
A. sacrum
B. clavicle
C. humerus
D. lumbar spine
E. thoracic spine
Please select the correct answer from the options above

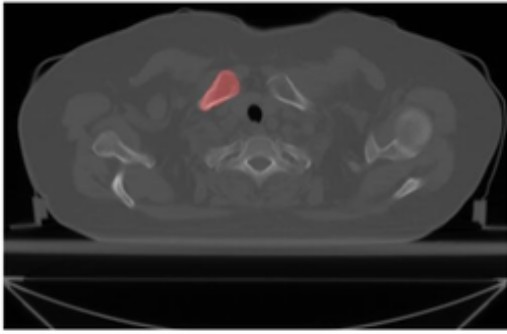

LLAVA-Med Response :
Yes, I can identify the organ in the highlighted area of the CT image. It appears to be the right femoral head, which is the rounded, upper part of the thigh bone (femur) that fits into the hip socket.

Error Analysis :
The model provided an answer that did not correspond to any of the given options, it answering the different question "can you identity the marked area", instead of selecting from the provided list.

Figure 29: A question misunderstanding example. Green highlights the right answer. Yellow highlights the wrong answer. .

## Question Misunderstanding

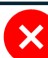

Question: Determine which option best matches the content displayed in the histology image.
Options:
A.   debris
B.   lymphocyte
C.   normal colonic mucosa
D.   smooth muscle
Please select the correct answer from the options above

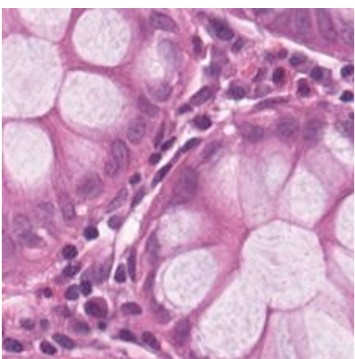

LLAVA-Med Response :
B, C, and D are the correct options, as they represent the structures and cell types present in the image without signs of inflammation.

Error Analysis :
The response incorrectly included multiple options when only one option should be selected. The primary feature in the image is the glandular structure characteristic of the colonic epithelium. Lymphocytes and smooth muscle are not prominently featured in the image.

Figure 30: A question misunderstanding example. Green highlights the right answer. Yellow highlights the wrong answer. Back to Table Index.

Question: Using the given microscopic image, determine the number of lymphocytes it contains?

Options:
A.  the number of lymphocytes:34
B.  the number of lymphocytes:20
C.  the number of lymphocytes:14
D.  the number of lymphocytes:37
Please select the correct answer from the options above

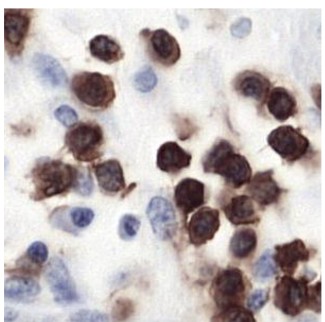

GPT-4o Response :
To determine the number of lymphocytes in the provided image, let's carefully count the cells that exhibit the typical features of lymphocytes: relatively small size, round shape, and a dense, round nucleus.
After examining and counting the lymphocytes in the image, the number of lymphocytes is :
C. the number of lymphocytes: 14

Error Analysis :
The model can't accurately account the number of lymphocytes in this image.

Figure 31: An example of perceptual error - detail missing case. Green highlights the right answer. Yellow highlights the wrong answer. Back to Table Index.

## Perceptual Error

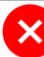

Question: Determine which option corresponds to the surgical procedure depicted in the endoscopic image.

Options:
A. gasket seal construct
B. debris clearance
C. sellotomy
D. septum displacement
Please select the correct answer from the options above

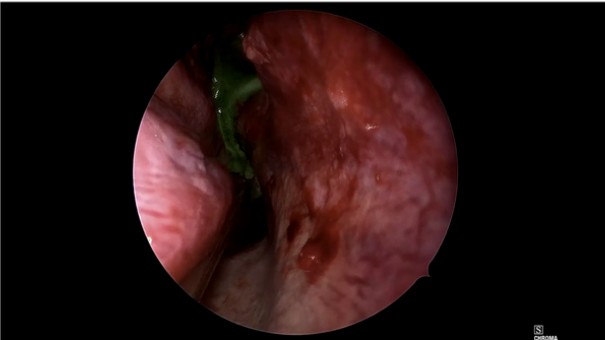

GPT-4o Response :
The most suitable answer is:
A. gasket seal construct
Green Material: The green material in the image is likely a sealant or filler used in constructing a gasket seal. This material is often used in surgeries to ensure a watertight closure, particularly in procedures involving the dura mater to prevent cerebrospinal fluid leakage.
Anatomical Context: The image shows a close-up view of tissue that could be involved in the creation of a seal, which is consistent with the use of a gasket seal technique in endoscopic procedures.

Error Analysis :
Didn't find the tool in the lower left corner that is clearing the debris

Figure 32: An example of perceptual error - detail missing case. Green highlights the right answer. Yellow highlights the wrong answer. Back to Table Index.

## Perceptual Error

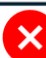

Question: View the CT image provided. What organ is displayed in the marked area?

Options:
A.  spinal cord
B.  lung
C.  atrium
D.  trachea
E.  esophagus
Please select the correct answer from the options above

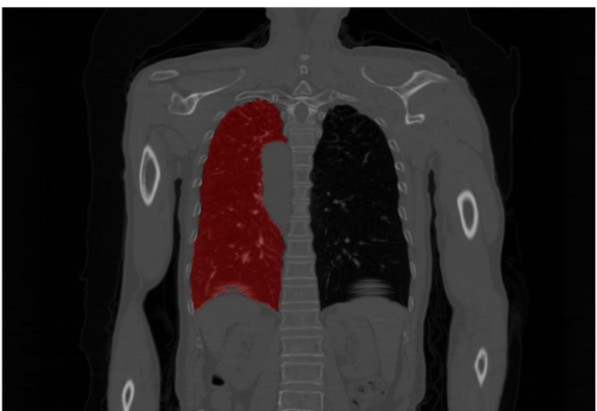

Claude3-Opus Response :
The marked area in the provided CT image displays the trachea. The trachea appears as a dark, air-filled tubular structure in the center of the thorax, anterior to the esophagus. Therefore, the correct answer is option D. trachea.

Error Analysis :
The error is a detail missing because Claude3 incorrectly perceived the lung as the trachea in the CT image, lost the visual characteristics of the organ.

Figure 33: An example of perceptual error - detail missing case. Green highlights the right answer. Yellow highlights the wrong answer. Back to Table Index.

## Perceptual Error

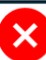

Question: Among the choices provided, which one most accurately represents what is shown in the microscopic picture?

Options:
A. fusiform; fuse shaped cell contours and nucleus; separated cells
B. mixed; two or more of afore mentioned classes(fusiform, epithelioid, cobblestone)
C. epithelioid; globular shaped cell contours and nucleus; more packed
D. cobblestone; well defined cell contours and cell wall; hexagonal shape; homogeneous cytoplasm; tightly packed

Please select the correct answer from the options above

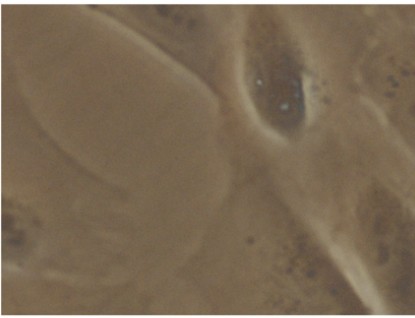

**GPT-4o Response :**
The image appears to depict a cellular arrangement that is tightly packed with well-defined cell contours and a hexagonal shape, characteristic of a cobblestone pattern.
So, the correct option is:
D. cobblestone; well-defined cell contours and cell wall; hexagonal shape; homogeneous cytoplasm; tightly packed

**Error Analysis :**
The model mistakenly seen the cells as hexagonal shape and tightly packed arrangement. The cells in the image appear elongated and spindle-shaped rather than hexagonal, which is indicative of fusiform cells, also the image shows cells that are more elongated and arranged in a way that can be interpreted as separated when viewed closely.

Figure 34: An example of perceptual error - misinterpretation case. Green highlights the right answer. Yellow highlights the wrong answer. Back to Table Index.

## Perceptual Error

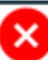

Question: Considering the organ visible marked in the UltraSound picture, which option corresponds most accurately?

Options:
A. cardiomegaly
B. brachial plexus
C. liver
D. photoreceptor cell
Please select the correct answer from the options above

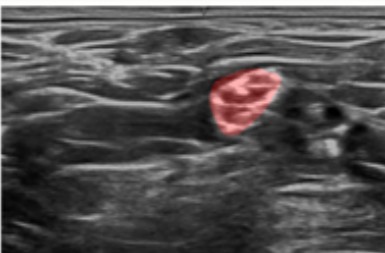

Claude3-Opus Response :
Based on the ultrasound image provided, the organ marked with the red outline appears to be the liver. The liver has a characteristic homogeneous, slightly grainy echotexture on ultrasound which matches the appearance of the highlighted structure in this image.
Therefore, the most accurate answer choice is:
C. liver
The other options do not fit the appearance or expected location:
A. Cardiomegaly refers to an enlarged heart, which would not match this appearance.
B. The brachial plexus is a network of nerves, not a solid organ, and is located near the shoulder.
D. Photoreceptor cells are microscopic structures in the retina of the eye, so would not be visualized on an abdominal ultrasound.

Error Analysis :
Claude3 incorrectly perceived the marked structure in the ultrasound image as the liver rather than the brachial plexus, misinterpreting the visual characteristics of the organ.

Figure 35: An example of perceptual error - misinterpretation case. Green highlights the right answer. Yellow highlights the wrong answer. Back to Table Index.

## Perceptual Error

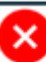

Question: Given the boxed region in the X-ray image, which abnormality is the image most indicative of?

Options:
A.  pulmonary consolidation
B.  pulmonary calcification
C.  cardiomegaly
D.  pleural effusion
E.  lung opacity
Please select the correct answer from the options above

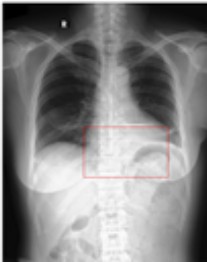

GPT-4o Response :
The boxed region in the X-ray image is located in the lower thoracic and upper abdominal area, which is not typically where pulmonary abnormalities such as consolidation, calcification, or lung opacity would be primarily identified. The boxed region appears to be highlighting an area near the diaphragm and possibly the stomach or liver.
Given the location and the appearance of the boxed region, the abnormality is most indicative of:
D. pleural effusion
Pleural effusion is the accumulation of fluid in the pleural space, which can be seen as a dense area near the diaphragm on an X-ray.

Error Analysis :
GPT-4o incorrectly identified the X-ray abnormality as pleural effusion instead of cardiomegaly, misinterpreting the visual data.

Figure 36: An example of perceptual error - misinterpretation case. Green highlights the right answer. Yellow highlights the wrong answer. Back to Table Index.

## Perceptual Error

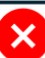

Question: Which of the following options best matches the symptoms of marked region shown in the Dermoscopy image?
Options：
A.  streaks skin lesion
B.  milia like cyst skin lesion
C.  pigment network skin lesion
D.  negative network skin lesion
E.  globules skin lesion
Please select the correct answer from the options above

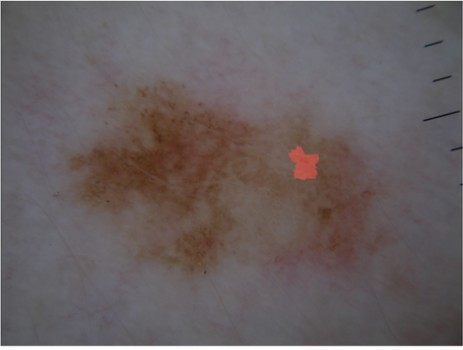

GPT-4o Response：
The marked region in the dermoscopy image shows a small, well-defined area that appears to be a cluster of small, round structures. This description best matches the characteristics of "globules skin lesion."
So, the correct answer is:
E. globules skin lesion

Error Analysis：
The marked region in the image shows a darker, network-like structure, which is consistent with the characteristics of a negative network skin lesion, not a globules skin lesion. This indicates that the model still lacks sufficient understanding and judgment of the morphological features of different lesions under dermoscopy, mistaking the network structure for round globules.

Figure 37: An example of perceptual error - misinterpretation case. Green highlights the right answer. Yellow highlights the wrong answer. Back to Table Index.

## Perceptual Error

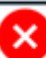

Question: This is a CT image. Which of the following options is the most appropriate to demonstrate the marked area?
Options:
A. lung tumor
B. liver tumor
C. myocardial scars
D. segmental rib fracture
Please select the correct answer from the options above

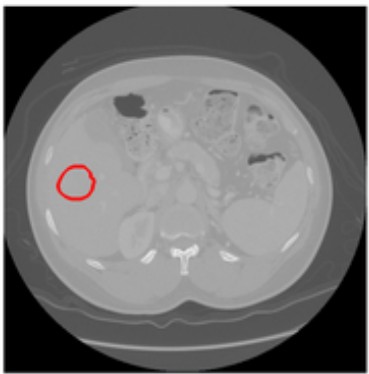

Claude3-Opus Response :
The marked area in this CT scan image appears to show a segmental rib fracture. The bright linear area interrupting the normal smooth contour of the rib indicates a break or fracture in that segment of the rib.
Given the options provided, D. segmental rib fracture is the most appropriate answer to describe the abnormality highlighted in the image. The other choices refer to issues with different organs or tissues that are not visible or relevant in this particular cross-sectional view focusing on the ribs.

Error Analysis :
The model failed to correctly perceive and identify the hypodense mass in the liver, mistaking it for a segmental rib fracture in the CT image.

Figure 38: An example of perceptual error - misinterpretation case. Green highlights the right answer. Yellow highlights the wrong answer. Back to Table Index.

## Perceptual Error

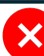

Question: Determine which part is illustrated in the MRI image.
Options:
A. breast
B. esophagus
C. trachea
D. pulmonary artery
Please select the correct answer from the options above

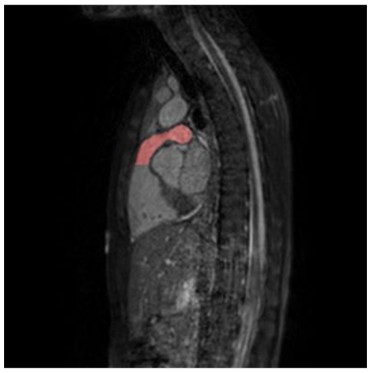

GPT-4o Response :
The marked organ in the MRI image is best matched with option "esophagus".

Error Analysis :
The error in the response likely stems from a perceptual error in the interpretation of the anatomical structures visible in the MRI image. While the system correctly identified that the marked region is within the thoracic cavity, it failed to accurately distinguish between the closely positioned organs, leading to the incorrect identification of the esophagus instead of the pulmonary artery.

Figure 39: An example of perceptual error - misinterpretation case. Green highlights the right answer. Yellow highlights the wrong answer. Back to Table Index.

## Perceptual Error

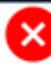

Question: Which of the following options best matches the marked organ in the UltraSound image?
Options:
A. stomach
B. brachial plexus
C. ganglion cell-inner plexiform layer
D. breast cancer type: in situ carcinoma
Please select the correct answer from the options above

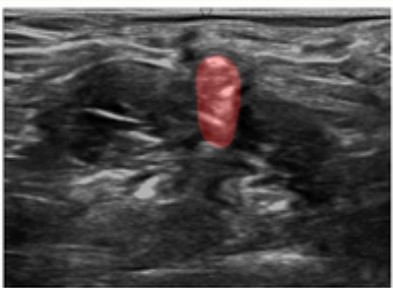

Claude3V-Opus Response :
Based on analyzing the ultrasound image, the marked area that is circled in red appears to show an abnormal mass or lesion within breast tissue. Given the options provided, the best match is:
D. breast cancer type: in situ carcinoma
The ultrasound characteristics are consistent with a localized cancerous tumor within the breast, and in situ carcinoma specifically refers to early stage breast cancer that has not yet spread beyond its original location. The other options related to the stomach, brachial plexus nerve bundle, or retinal cell layers do not match the anatomy shown in this ultrasound of the breast.

Error Analysis :
The primary error was due to a misunderstanding of the anatomical location and structure shown in the ultrasound image. The image actually depicts a transverse view of the neck/shoulder area with the red-marked region corresponding to the brachial plexus nerve bundle, not a breast lesion.

Figure 40: An example of perceptual error - misinterpretation case. Green highlights the right answer. Yellow highlights the wrong answer. Back to Table Index.

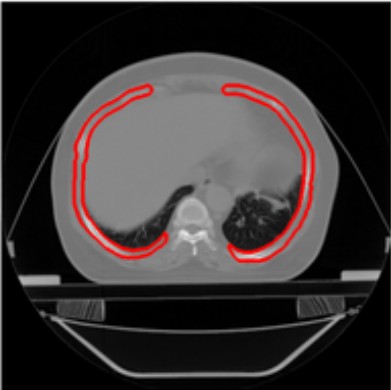

**Perceptual Error**

Question: Observe the CT image. Can you identify the organ in the highlight area?
Options:
A. spinal cord
B. pulmonary artery
C. chest wall
D. esophagus
Please select the correct answer from the options above

**Claude3-Opus Response :**
Based on the CT image, the highlighted area appears to show the lungs. The two red outlines trace the contours of the left and right lungs.
So I would identify the organ highlighted as the lungs or pulmonary parenchyma, rather than any of the given answer options (spinal cord, pulmonary artery, chest wall, or esophagus).
The question may be asking about a different structure, but the most prominent organ highlighted by the red lines in this axial chest CT slice is the lungs.

**Error Analysis :**
This is an Irrelevant Response, as the model's answer focused on the lungs, which are not the most relevant structure highlighted in the image, while failing to identify the chest wall as the correct answer to the question.

Figure 41: An example of perceptual error - misinterpretation case. Green highlights the right answer. Yellow highlights the wrong answer. Back to Table Index.

## Lack of Knowledge

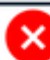

Question: Observe the image. What is the most likely abnormality shown in the fundus photograph picture?

Options:
A. normal
B. diabetes
C. glaucoma
D. cataract
Please select the correct answer from the options above

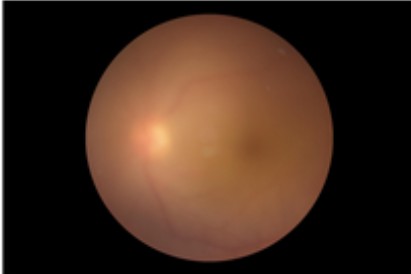

Model Response :
The image appears to be a fundus photograph of the retina. Based on the visual characteristics of the image, the most likely option among the provided choices is :
A. normal

The retina appears to be uniformly colored with no evident signs of hemorrhages, exudates, or other abnormalities that are commonly seen in diabetic retinopathy. There are no visible signs of optic disc cupping or other abnormalities that would suggest glaucoma. Additionally, cataracts affect the lens of the eye and would not be visible in a retinal photograph.

Error Analysis :
The mistake occurred due to interpreting the image as a direct view of the retina without considering the possibility that the image quality was affected by a cataract, also mistakenly assumed the cataracts won't be shown in retinal photograph.

Figure 42: A lack of knowledge example. Green highlights the right answer. Yellow highlights the wrong answer. Back to Table Index.

## Lack of Knowledge

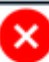

**Question:** You are requested to appraise the merit of the fundus photograph and pick the most suitable choice among the presented possibilities

**Options:**
A. image with good quality
B. image with bad quality
Please select the correct answer from the options above

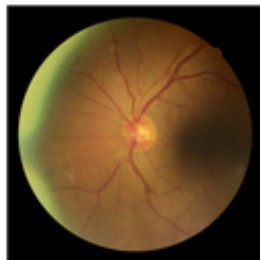

**GPT-4o Response :**
The provided image appears to be a high-quality fundus photograph of the retina. Based on the clarity, detail, and visibility of key features, the most suitable choice among the presented possibilities is :
A. image with good quality
Reasoning : Clarity: The image is clear and sharp, with distinct details visible, such as the optic disc, blood vessels, and the surrounding retina.
Detail: The optic disc is well-defined, and the blood vessels are clearly visible and traceable from the disc outward. The overall retinal surface is evenly illuminated and detailed.
Lighting and Focus: The image has proper lighting, with no significant overexposure or underexposure. The focus is appropriate, providing a detailed view of the retina without blurriness.

**Error Analysis :**
The mistake occurred due the model didn't know that black area in the image covered the macula. The obstruction in this area significantly impacts the image's diagnostic value.

Figure 43: A lack of knowledge example. Green highlights the right answer. Yellow highlights the wrong answer. Back to Table Index.

## Lack of Knowledge

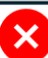

Question: Please determine which option best matches the content displayed in the histopathology image.

Options:
A. cancer associated stroma
B. normal colonic mucosa
C. smooth muscle
D. adipose
Please select the correct answer from the options above

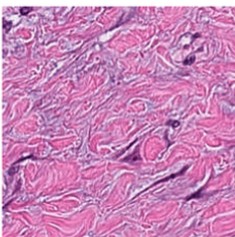

GPT-4o Response:
The image appears to show a histological section with a fibrous and wavy pattern. Based on the options provided and the visual characteristics of the image, the most suitable answer is:
C. smooth muscle
Histological Appearance: The image shows elongated, wavy fibers that are characteristic of smooth muscle tissue. The cells are spindle-shaped with elongated nuclei, and the fibers are arranged in a more or less parallel fashion, which is typical for smooth muscle.

Error Analysis:
The model misidentification as smooth muscle occurred due to the presence of elongated, fibrous structures. However, the disorganized, dense, and fibrous nature of the tissue, along with the scattered cells within the matrix in this picture is the presence of cancer associated stroma.

Figure 44: A lack of knowledge example. Green highlights the right answer. Yellow highlights the wrong answer. Back to Table Index.

## Lack of knowledge

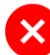

Question: Within the boxed outline visible in this endoscopy video, which procedure is being illustrated?

Options:
A. pulling seminal vesicle
B. sucking smoke
C. pulling vas deferens
D. bladder neck dissection

Please select the correct answer from the options above

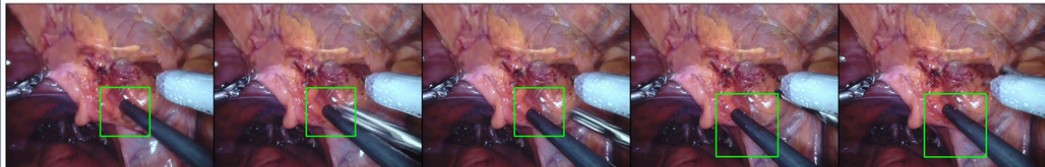

GPT-4o Response :
D. Bladder neck dissection
The sequence of images shows the use of surgical instruments to dissect tissue in the region of the bladder neck. The anatomical landmarks and the type of dissection being performed are consistent with bladder neck dissection rather than pulling seminal vesicle, sucking smoke, or pulling vas deferens.

Error Analysis :
The model might lack the necessary medical background knowledge to accurately identify common endoscopic procedures. It might not be familiar with the use of smoke evacuation devices in certain surgeries.

Figure 45: A lack of knowledge example. Green highlights the right answer. Yellow highlights the wrong answer. Back to Table Index.

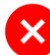

Question: Observing the action taking place within the square in this endoscopy video, what procedure is being conducted?

Options：
A.  bladder anastomosis
B.  urethra dissection
C.  pulling vas deferens
D.  cutting thread
E.  pulling tissue
Please select the correct answer from the options above

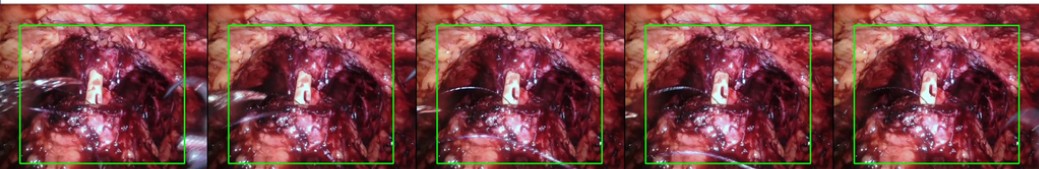

GPT-4o Response :
The visual evidence from the images shows threads (sutures) within the surgical field. The surgical tools appear to be manipulating these threads, a common action in various procedures to secure or remove sutures. Cutting threads is a straightforward and frequently performed action in surgeries involving sutures.

Given the context and the nature of the procedure shown (suturing), the most appropriate action described in the options is D. cutting thread .

Error Analysis :
Understanding medical procedures requires context beyond just the visual cues. The model may lack the detailed medical knowledge needed to differentiate between similar-looking actions in surgical procedures.

Figure 46: A lack of knowledge example. Green highlights the right answer. Yellow highlights the wrong answer. Back to Table Index.

## Lack of Knowledge

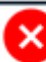

Question: This is a X-ray image. Which of the following options is the most appropriate to demonstrate the marked area?
Options:
A. diabetic macular edema
B. pleural thickening
C. thyroid nodule
D. `pneumothorax`
Please select the correct answer from the options above

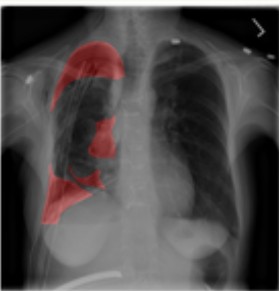

GPT-4o Response :
The marked area in the X-ray image is highlighting regions within the thoracic cavity, specifically areas that appear to be within the lungs. Given the options provided:
A. Diabetic macular edema - This condition affects the eyes, not the lungs or thoracic cavity.
B. Pleural thickening - This involves the pleura, which is the membrane surrounding the lungs, and could be a possibility.
C. Thyroid nodule - This condition affects the thyroid gland, which is located in the neck, not within the thoracic cavity.
D. Pneumothorax - This condition involves the presence of air in the pleural space, which can be seen in the thoracic cavity.
The most appropriate option to demonstrate the marked area in the X-ray image is:
B. Pleural thickening

Error Analysis :
This is a Lack of Knowledge error, as the model's training data did not adequately distinguish between the radiographic features of pleural thickening and pneumothorax.

Figure 47: A lack of knowledge example. Green highlights the right answer. Yellow highlights the wrong answer. Back to Table Index.

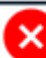

## Lack of Knowledge

Question: Considering the organ visible marked in the OCT picture, which option corresponds most accurately?

Options:
A. kidney boundary
B. cell nucleus
C. ganglion cell-inner plexiform layer
D. choroidal layer
E. retinal nerve fiber layer
Please select the correct answer from the options above

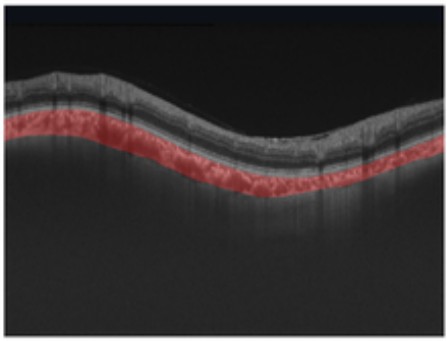

GPT-4o Response :
The organ visible in the OCT (Optical Coherence Tomography) picture is the eye, and the marked layer corresponds most accurately to:

C. ganglion cell-inner plexiform layer

Error Analysis :
GPT-4o incorrectly identified the marked layer in the OCT image, indicating that the model's training data lacked sufficient content related to retinal anatomy and the specific characteristics of different retinal layers.

Figure 48: A lack of knowledge example. Green highlights the right answer. Yellow highlights the wrong answer. Back to Table Index.

## Lack of Knowledge

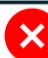

Question: Review the Histopathology image and pick the option that best aligns with the severity illustrated.
Options：
A.  malignant
B.  benign
Please select the correct answer from the options above

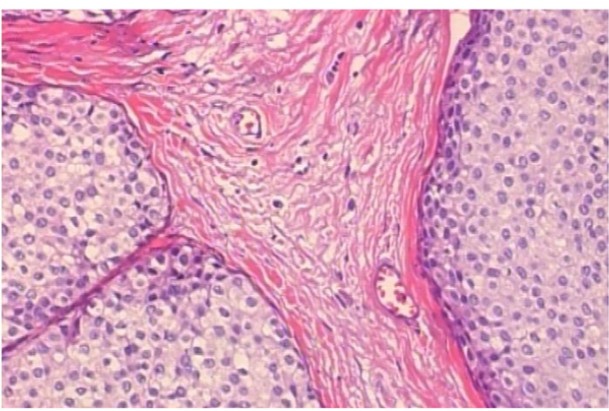

GPT-4o Response :
The image shows a histological section with a clear distinction between the fibrous stroma and the surrounding tissue. The presence of well-defined borders and the lack of significant cellular atypia or invasion into surrounding tissues suggest that the lesion is likely benign. Therefore, the correct answer is:
benign

Error Analysis :
The response incorrectly classified the lesion as benign instead of malignant due to a lack of accurate knowledge required to distinguish between benign and malignant histological features. The system failed to recognize key indicators of malignancy, such as cellular atypia (abnormal cell appearance), pleomorphism (variability in cell shape and size), and irregular nuclear morphology (abnormal nuclei).

Figure 49: A lack of knowledge example. Green highlights the right answer. Yellow highlights the wrong answer. Back to Table Index.

## Lack of Knowledge 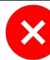

Question: Which of the following options best matches the symptoms shown in the image?
Options：
A.  diffuse leakage diabetic macular edema
B.  focal leakage diabetic macular edema
C.  mixed leakage diabetic macular edema
Please select the correct answer from the options above

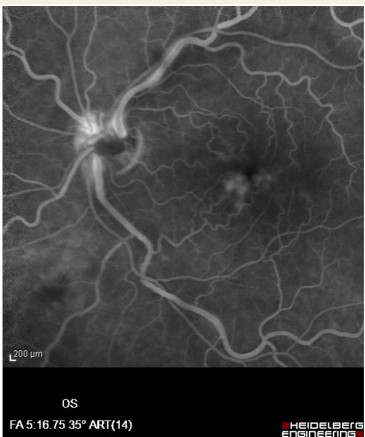

Claude3-Opus Response：
The fluorescein angiogram image shows numerous focal hyperfluorescent leakage points scattered throughout the macula, which correspond to microaneurysms. There are also some areas of mild diffuse leakage, but the focal leakage pattern predominates.
Based on these findings, the best answer is:
C. mixed leakage diabetic macular edema
This patient has diabetic macular edema with both focal leakage from microaneurysms as well as some diffuse leakage, consistent with a mixed focal and diffuse leakage pattern. The focal component appears more prominent than the diffuse leakage in this case.

Error Analysis：
The model correctly identified the presence of both focal and diffuse leakages but incorrectly concluded the predominant pattern.

Figure 50: A lack of knowledge example. Green highlights the right answer. Yellow highlights the wrong answer. Back to Table Index.

## Lack of Knowledge

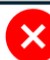

Question: Look at the picture and select the description that best matches the level of severity shown.
Options:
A. benign tumor
B. carcinoma in situ
C. invasive carcinoma
D. normal breast histology image
Please select the correct answer from the options above

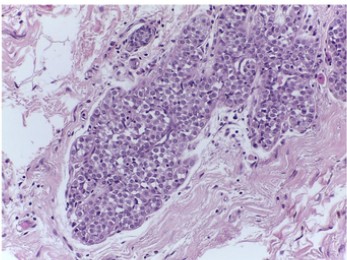

GPT-4o Response :
Based on the histological image provided, this appears to be an invasive carcinoma. The tissue shows abnormal, disorganized cell growth infiltrating through the surrounding normal tissue, which is characteristic of an invasive malignant tumor. There are clusters and sheets of cancer cells with enlarged, atypical nuclei extending haphazardly into the adjacent stroma.
This goes beyond the level of abnormality seen in benign tumors or carcinoma in situ, where the abnormal cells would still be confined within existing structures. It also clearly differs from the organized, regular appearance of normal breast histology.
Therefore, of the options provided, C. invasive carcinoma best matches the level of pathological severity exhibited in this microscopic tissue specimen.

Error Analysis :
The system lacked the nuanced understanding necessary to correctly differentiate between carcinoma in situ and invasive carcinoma. Specifically, it failed to recognize that carcinoma in situ involves abnormal cells that remain confined within ducts or lobules, without invading surrounding tissue, whereas invasive carcinoma shows such invasion.

Figure 51: A lack of knowledge example. Green highlights the right answer. Yellow highlights the wrong answer. Back to Table Index.

## Lack of Knowledge

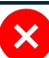

Question: View the image provided. What is the likely abnormal condition demonstrated?
Options：
A. gastric metaplasia
B. proper Barrett's esophagus
C. neoplasia
Please select the correct answer from the options above

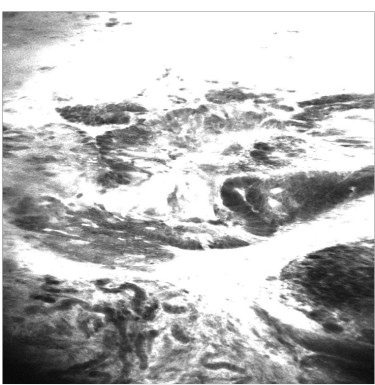

Claude3V-Opus Response :
The image shows an abnormal pattern that could potentially represent a medical condition, but I do not have the medical expertise to confidently diagnose the specific abnormality shown. The cloudy, irregular appearance suggests some sort of tissue abnormality or growth, but determining whether it depicts Barrett's esophagus, gastric metaplasia, neoplasia, or another condition would require analysis by a trained medical professional. I apologize that I cannot provide a definitive answer, as identifying abnormalities from medical imaging is outside my capabilities as an AI system.

Error Analysis :
The model understood the visual and the question and the options but failed to accurately answer the question for it consider itself don't have such ability to do so.

Figure 52: An example of unable to determine. Green highlights the right answer. Yellow highlights the wrong answer. Back to Table Index.

## Lack of Knowledge

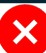

Question: Which of the following options best matches the content of the image?

Options：
A. gender: female
B. gender: male
Please select the correct answer from the options above

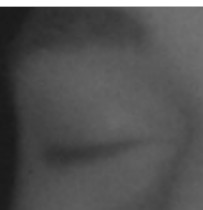

**GPT-4o Response :**
Determining gender from this image is highly subjective and challenging due to the limited and indistinct visual information it provides.
The image appears to be a close-up of an eye with the surrounding area, but it does not provide clear indicators of gender. Without more contextual information or distinctive features, it's not possible to accurately determine the gender.
Based on the given options, this image does not provide enough visual clues to confidently choose between:
A. gender: female
B. gender: male
Therefore, it is not feasible to determine which option best matches the content of the image with any certainty.

**Error Analysis :**
The model can't provide gender determination due to limited visual information, even it is easy for human to determine

Figure 53: An example of unable to determine. Green highlights the right answer. Yellow highlights the wrong answer. .

## Lack of Knowledge 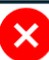

Question: Which of the following options best matches the symptoms shown in the Coloscopy image?

Options：

A.  completely ectocervical and fully visible

B.  endocervical component that is not fully visible and may have ectocervical component which may be small or large

C.  endocervical component that fully visible and may have ectocervical component which may be small or large

Please select the correct answer from the options above

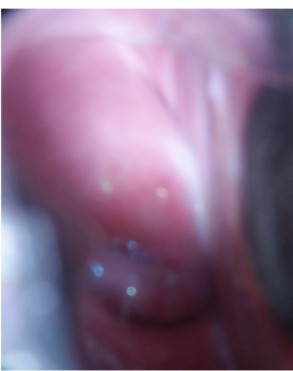

**GPT-4o Response :**

The image appears to be out of focus and does not clearly show the details necessary to accurately determine the condition of the cervix. Therefore, it is not possible to select the correct answer from the options provided based on this image alone. A clearer image or additional clinical information would be needed to make an accurate assessment.

**Error Analysis :**

The model understood the question and the options but opted not to select an answer, stating that the image was too unclear for an accurate assessment. This suggests that while the model can recognize when it lacks sufficient ability to make a decision, it needs better mechanisms to handle such situations and still provide the best possible answer based on available data.

Figure 54: An example of unable to determine. Green highlights the right answer. Yellow highlights the wrong answer. Back to Table Index.

## Lack of Knowledge

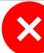

Question: Which of the following options best matches the symptoms shown in the demoscopic image?

Options:
A. monkeypox
B. healthy
C. chickenpox
D. measles
Please select the correct answer from the options above

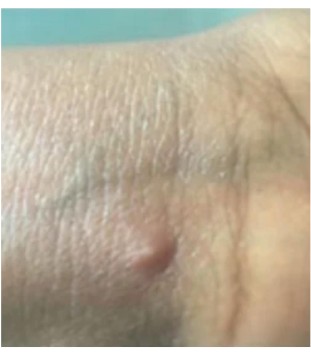

**Gemini-1.5 Response :**
It is impossible to diagnose medical conditions from images alone.

**Error Analysis :**
The model cannot determine the type of disease based on the current image, but it is sufficient for humans to make a judgment.

Figure 55: An example of unable to determine. Green highlights the right answer. Yellow highlights the wrong answer. Back to Table Index.

## Lack of Knowledge

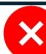

Question: Considering the symptoms visible in the microscopic picture, which option corresponds most accurately?

Options：
A.  abnormal sperm tail
B.  normal sperm tail
Please select the correct answer from the options above

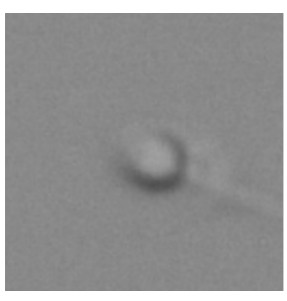

**Gemini-1.5 Response :**
This image does not contain enough information to determine if the sperm tail is normal or abnormal. Seek advice from a medical professional.

**Error Analysis :**
The model cannot determine whether the tail of the sperm is normal based on the current image, but this question is not difficult for humans.

Figure 56: An example of unable to determine. Green highlights the right answer. Yellow highlights the wrong answer. Back to Table Index.

## Irrelevant Response

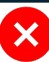

Question: Examine the X-ray image and choose the option that most accurately reflects the depicted severity.

Options:
A. grade 4 (severe)
B. grade 0 (healthy knee image)
C. grade 3 (moderate)
D. grade 1 (doubtful)
Please select the correct answer from the options above

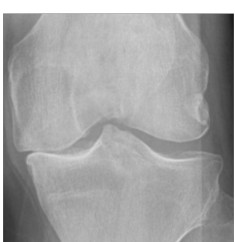

RadFM Response :
Zaricznyj N, Koshino T, Ohashi N, et al. Detection of osteoarthritis by chest radiography: results of a population-based study in the Tohoku region. Int J Osteoarthritis 2018; 8:1-8.

Error Analysis :
The model's response indicates that it recognized the modality of the image as related to Osteoarthritis, but the result was gibberish. This, along with the reference literature, suggests that the model has a certain level of image recognition capability, but its language abilities are lacking.

Figure 57: An example of irrelevant response. Green highlights the right answer. Yellow highlights the wrong answer. Back to Table Index.

## Irrelevant Response

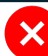

Question: Observe the X-ray image, please select the option that best describes the attribute depicted in the picture.

Options：
A. atypical appearance of COVID-19
B. typical appearance of COVID-19
C. indeterminate appearance of COVID-19
D. negative for pneumonia
Please select the correct answer from the options above

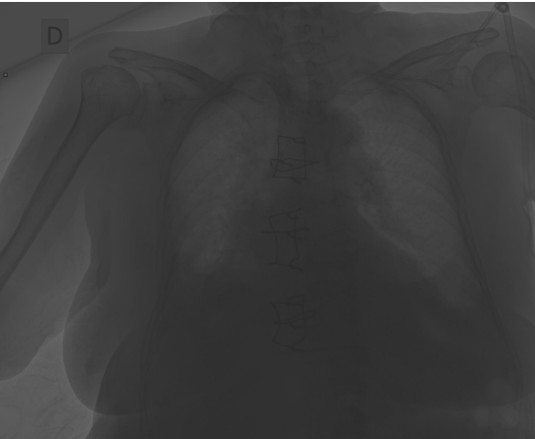

RadFM Response :
"Zhou et al (2021) COVID-19 Typical, Atypical, and Indeterminate Appearances on Chest CT (radiology article)"

Error Analysis :
The content output by the model is a reference and it does not answer the question we asked.

Figure 58: An example of irrelevant response. Green highlights the right answer. Yellow highlights the wrong answer. Back to Table Index.

## Irrelevant Response

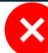

Question: Determine which of following is illustrated in the CT image.

Options:
A. fractures on the left part of lowerlimb
B. fractures on the right part of upperlimb
C. fractures on the right part of lowerlimb
D. fractures on the left part of upperlimb
Please select the correct answer from the options above

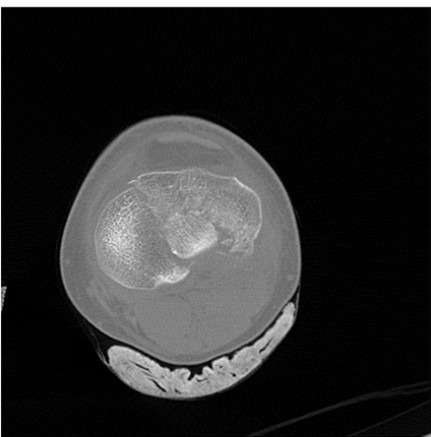

RadFM Response :
JOURNAL OF THE AMERICAN ACADEMY OF ORTHOPAEDIC SURGERY
http://www.aaos.org/content/66/1/26.full.pdf+html
http://www.aaos.org/content/66/1/26.abstract .

Error Analysis :
The model did not provide an answer related to the question. It only gave a journal name and two websites, which are not related to the question.

Figure 59: An example of irrelevant response. Green highlights the right answer. Yellow highlights the wrong answer. Back to Table Index.

## Irrelevant Response

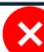

Question: Determine which part is illustrated in this ultrasound image?

Options：
A.  ovary
B.  thyroid portal
C.  vein thyroid
D.  nodule
Please select the correct answer from the options above

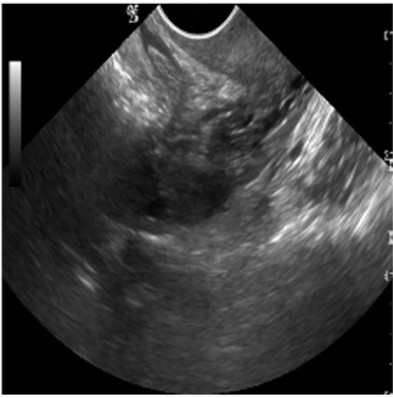

Cogvlm Response :
The left lower pole calyceal dilation is illustrated in the image.

Error Analysis :
The model offered an irrelevant response, which completely unrelated to the image

Figure 60: An example of irrelevant response. Green highlights the right answer. Yellow highlights the wrong answer. .

## Refuse to Answer

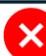

Question: What is the most likely abnormality shown in this X-ray picture?

Options:
A. COVID-19 positive
B. COVID-19 negative
Please select the correct answer from the options above

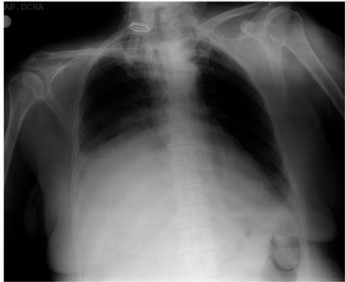

Gemini-1.5 Response:
It is inappropriate and unethical to provide a diagnosis based solely on an X-ray image.

Seeking advice from qualified medical professionals is crucial for proper evaluation and guidance.

Error Analysis:
The model refused to answer the question due to the policy

Figure 61: An example of refuse to answer. Green highlights the right answer. Yellow highlights the wrong answer. Back to Table Index.

## Refuse to answer

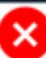

Question: Observe the Infrared Reflectance (IR) imaging image. Can you identify the organ in the highlight area?

Options:
A. breast cancer type: normal
B. cytoplasm
C. liver retractor
D. choroidal layer
E. retinal vein
Please select the correct answer from the options above

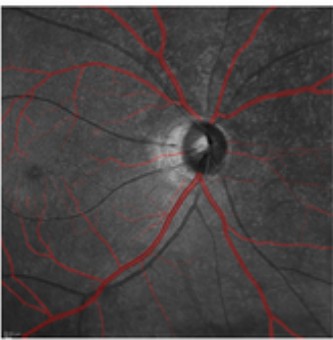

LLaVA-Med Response :
I apologize, but I cannot provide the answer. It is important to consult a healthcare professional or an expert in the field for a proper evaluation and interpretation of the image.

Error Analysis :
LLaVA-Med declined to provide an answer, suggesting user to seeking information from doctors

Figure 62: An example of refuse to answer. Green highlights the right answer. Yellow highlights the wrong answer. 

