# OpenReview forum: "GMAI-MMBench: A Comprehensive Multimodal Evaluation Benchmark Towards General Medical AI"
_NeurIPS.cc/2024/Datasets_and_Benchmarks_Track — NeurIPS 2024 Track Datasets and Benchmarks Poster_

### Official Review · Reviewer_4iGG · 2024-07-18
**Great job but several problem need to be addressed**

**Rating:** 5
**Confidence:** 5
**Clarity:** Yes

**Review:**

Cons
- Risk of data leakage: MLLM are trained on vast amounts of data from the internet, and even MLLM have likely encountered these images during the pre-training of their visual backbones. Additionally, the proportion of private data in this benchmark is very small, roughly 1000/26000. How can you ensure that your data has not been seen by these models before?
- Ethical issues: Have these private datasets undergone ethical review and approval?
- QA Generation Process: How are the options for the QA generation step enumerated? What prompts were used for GPT-4V? This is crucial as it directly impacts the quality of the benchmark.
- Performance Across Modalities: It is commendable to see the performance of different MLLMs across various modalities and the distribution of these modalities. Please provide detailed visualizations or tables to illustrate this information.
- Distribution of Multiple-Choice Options: Since the dataset consists of multiple-choice questions, what is the distribution of the options? MLLMs might exhibit preferences for certain types of choices. Understanding this distribution is essential.
- Evaluation of Rationales: Beyond the final results, it is crucial to evaluate the rationale behind VLLMs’ answers for medical use cases. Sometimes, the final answers might be correct, but the underlying rationale can be flawed (https://arxiv.org/abs/2401.08396 for reference). The authors should delve deeper into this aspect.
- Reliability of Answer Extraction: How can you ensure that the step where ChatGPT-3.5-turbo-0613 extracts the answer from the model’s outputs is reliable? This step is critical for the integrity of the evaluation process.
- Incorporation of Patient Information: This benchmark aims to evaluate MLLMs' capabilities in medical scenarios. In real applications, patient information is crucial as clinical decisions vary from case to case depending on the patient, requiring a more real-life setting. Most questions in this paper do not address patient information. This raises concerns about whether the evaluation genuinely reflects real clinical settings and whether the evaluation design is robust enough to account for these factors.
- Related work: I notice that there is another work also considering department issue (https://arxiv.org/html/2402.11217v1), please discuss the difference.

**Strengths:**

- The benchmark's construction from 285 diverse clinical datasets across 38 imaging modalities.
- Lexical tree structure and multi-perceptual granularity assessments.
- The evaluation is comprehensive and provides several key findings.

**Additional Feedback:**

See above

**Correctness:**

Yes. But I have concern about the data leakage problem of this benchmark since it use little in-house data and have no instruction for the following step of this in-house data.

**Documentation:**

The details are not sufficient. Especially, the availability and ethical use is absent.

**Ethics:**

I don't know whether the private datasets in this benchamrk undergone ethical review and approval.

**Limitations:**

See above

**Opportunities For Improvement:**

This job is great, elaborate and comprehensive. But I mainly worry about the Risk of data leakage, Ethical issues, Reliability of Answer Extraction and Incorporation of Patient Information problem. I would reconsider my score after rebuttal.

**Relation To Prior Work:**

Yes

**Summary And Contributions:**

The submitted paper introduces GMAI-MMbench, a comprehensive multimodal evaluation benchmark specifically designed for assessing GMAI systems. The benchmark is notable for its extensive dataset collection and innovative evaluation framework.

---

> ### Author Rebuttal · Authors · 2024-08-17
>
> A1: We acknowledge that we cannot fully guarantee that public images have not been encountered during the pre-training of the visual backbone. However, the concept of data leakage in this context, as outlined in references [9, 15], focuses on two key aspects:
> 1. "Its data or annotations should not come from existing publicly available datasets as much as possible" in [15]. Many existing benchmarks are built from original image-text paired data, which carry a high risk of data leakage.
> 2. "LLM and LVLM could still answer some visual-necessary questions without visual content" in [9]. This issue is prevalent in several popular benchmarks like MMMU reported in [9].
>
> For (1): While some images in our benchmark are publicly sourced, we newly created textual descriptions. Thus, the QA pairs are newly constructed using our construction pipeline and have been manually evaluated. For (2), we have carefully designed our questions to maintain a strong correlation with the corresponding images, effectively preventing LVLMs from answering the questions without visual content.
> Therefore, compared to the existing benchmark constructed from original paired image-text data, our benchmark has a lower data leakage risk.
>
> A2: Yes, all the data has undergone ethical review.
>
> A3: The quality of our benchmark is maintained through strict QA generation rules and post-manual evaluation, which ensures its impact. Lines 173-186 described them. We will add more details in the final version.
> * Option generation: In each dataset, images are well-labeled by the categories contained in this dataset. If we simply mix these datasets, the global view may contain multiple correct labels (e.g., a fundus image dataset that focuses on pathological myopia might also contain other diseases like diabetic retinopathy). Therefore, we use the individual collected dataset to generate options for the global view, while we use the mixed datasets for the local view. (1) For global view: To avoid the above multiple-label problem, for each selected image, we construct its options using its original dataset, since the extracted options for this image are mutually exclusive in this dataset. We regard the label of this image as the correct option and randomly select other labels in this dataset as the wrong options. (2) For local view: We mix the collected datasets with same modality, clinical VQA task, and perceptual granularity as one individual "dataset". Similar to the global view, the correct and wrong options are constructed.
> * Prompts: As each question must include the information of modality (M), clinical task (C), and perceptual granularity (P) to have a strong relation to the image, we group labels by these elements, randomly select 5–10 cases, and use the template like: "Based on the provided <M> image and its label, develop 10 high-quality multiple-choice questions. You must include the three key components, including modality <M>, clinical task <C>, and perceptual granularity <P>. An example question like this: This is an MR image. Which of the following options is the most appropriate to demonstrate symptoms in the marked area?"
>
> A4:  Figure 1 in the PDF provide the performance of GPT-4 across 38 modalities (see the light green part) and the distribution of modalities(the blue part).
>
> A5: Please refer to Figure 2 attached to PDF.
>
> A6. Our pioneering comprehensive benchmark is designed to assess LVLMs across a wide range of multimodal tasks in the medical domain, encouraging the development of the next generation of LVLMs for GMAI. In this initial version, we focus on evaluating LVLMs’ effectiveness through multi-choice visual questions. We appreciate your suggestion that understanding the rationale behind answers is crucial. We plan to address it in future work by incorporating Chain-of-Thought (CoT) rationales and involving medical experts in annotating rationales. This will help ensure that LVLMs not only provide correct answers but also follow sound reasoning. In addition, we have included case studies in supplementary material to allow for a deeper review of outputs and underlying rationales.
>
> A7: Figure 5 of MMBench (https://arxiv.org/pdf/2307.06281) compared the alignment rates between humans and different LLMs in the extracting choices step. ChatGPT-3.5-turbo-0613 achieves 85% alignment rate. We chose it for its high alignment rate and the findings in the last paragraph of section 4.2, "we also note that the slight difference in top-performing LLMs’ alignment rates has little effect on the quantitative performance of VLMs."
>
> A8: All image-answer pairs are accurately extracted from clinical scenarios, ensuring the correctness of the QA pairs as patient information may be considered during labeling. While patient information is important, it does not affect our benchmark's effectiveness. For instance, an X-ray showing lung damage might suggest pneumonia or other complications, but without that information does not impact the model's ability to identify the presence of lung damage. In future work, we plan to incorporate patient information to support more comprehensive evaluations.
>
> A9: We have outlined the differences between Asclepius and our benchmark in the following table. GMAI-MMBench features more departments and offers greater comprehensiveness and diversity across various aspects. In terms of data leakage, our benchmark _(from 285 non-image-text-paired datasets from both public and hospital)_ is less at risk compared to Asclepius _(from open datasets with VQA format, USMLE, Textbook)_ as our data are not sourced from original image-text paired datasets. Additionally, our benchmark evaluates 50 LVLM models, compared to only 6 in Asclepius.
> \begin{array}{lcccccc}\hline\\\\\textbf{Benchmark}&\textbf{Modality}&\textbf{Size}&\textbf{Task}&\textbf{Dept}&\textbf{PG}&\textbf{Evaluated Models}\\\\\hline\text{Asclepius}&\text{unclear}&3232&8&15&I&6\\\\\hline\text{GMAI-MMBench}&38&26K&19&18&I,B,M,C&50\\\\\hline\end{array}

---

> > ### Comment · Reviewer_4iGG · 2024-08-20
> > **Reply**
> >
> > Most of concerns have been addressed. but still two question.
> > - Ethical review and approval should be demonstrated.
> > - For the Q8, I have mentioned before that "This benchmark aims to evaluate MLLMs' capabilities in medical scenarios. In real applications, patient information is crucial as clinical decisions vary from case to case depending on the patient, requiring a more real-life setting". This is also the main feature of clinical application. For instance, the age information is important in clinical setting. Chest pain in a 20-year-old might be due to muscle strain or anxiety, while in a 60-year-old it could indicate a heart attack. I think this part is not well designed in this benchmark.

---

> > > ### Author Rebuttal · Authors · 2024-08-26
> > >
> > > A1: The approval can be found from the [link](https://drive.google.com/file/d/1hjhOO1CrGAzLmASmzCaUfrvR6ESBlAS9/view?usp=sharing).
> > >
> > > A2: We intend to answer the question from two sides:
> > > * ***Why We Do Not Use Patient Information to Construct Questions.***
> > >   * **QA Pairs Accuracy:** Our benchmark is designed to assess LVLMs' capabilities in medical scenarios by evaluating whether models can answer correctly by combining the visual and language information when provided with the necessary specific question and corresponding medical image. So ensuring the accuracy of the QA pairs is a critical aspect of constructing this benchmark. Our data is meticulously annotated by clinical experts and physicians based on real clinical cases, which guarantees the accuracy of the QA pairs. For instance, in the labeling process, experts may consider additional patient information to ensure the correct data labels. As per your example, clinical experts would label a 60-year-old patient with a "heart attack", whereas a 20-year-old patient might be labeled as having "anxiety" or "normal". Therefore, the concern about mislabeling a 20-year-old with chest pain as having a "heart attack" does not occur in our benchmark.
> > >   * **Impact of Patient Information in the Benchmark:** When the QA pairs are accurate, advanced LVLMs should be able to answer the questions correctly using only the provided content in the question and corresponding medical images. As a result, while patient information is important, it does not affect our benchmark's effectiveness. In other words, additional patient information is likely to provide more helpful information and reduce the challenge for the models.
> > > * ***How We Plan to Incorporate Patient Information in the Future Work.***
> > >   * **Patient Information's Role in Improving Accuracy for LVLMs:** We recognize the importance of patient information in doctors' real clinical decisions, but its impact on improving the diagnostic accuracy of LVLMs is still an open question that requires further investigation. We appreciate your suggestion and plan to explore this issue more deeply by incorporating patient information into our future work.

---

> ### Author Response · Authors · 2024-08-29
> **Anything Else We Can Address for You?**
>
> Dear Reviewer 4iGG,
>
> Thank you once again for your valuable comments and the time you've taken to review our work. We hope our responses have satisfactorily addressed your questions.
>
> As the discussion period approaches its end, we wanted to check if is there anything further that you would like us to clarify or elaborate on?
>
> We would be more than happy to provide any further clarifications, discussions, or additional experiments to ensure all your concerns are fully addressed.
>
> Sincerely,\
> Authors of Paper 900

---

### Official Review · Reviewer_4FHY · 2024-07-23
**Review of Submission 900**

**Rating:** 8
**Confidence:** 3
**Correctness:** The authors should use the full term …
**Clarity:** The paper is well written and organized.

**Review:**

The strengths of the paper are the diversity of the datasets and the extensive evaluation.

**Strengths:**

1. The diversity of the dataset: it is constructed from 285 datasets including 38 modalities, 19 tasks, and 18 departments.
2. The extensive evaluation: the authors have evaluated 50 models.
3. Insightful discussion: the authors have analyzed 5 insufficiencies to address in novel models.

**Additional Feedback:**

No additional feedback.

**Documentation:**

The documentation is clear and very detailed.

**Ethics:**

No ethics issue found.

**Limitations:**

The texts in Figure 3 are too small.

**Opportunities For Improvement:**

1. In the abstract, the authors should state what GMAI is before using its abbreviation, though we know it stands for general medical AI.
2. This work can continue on including more datasets as an ever-improving benchmark.

**Relation To Prior Work:**

There is a comparison with other benchmarks in the paper.

**Summary And Contributions:**

This paper introduces a benchmark for general medical AI based on 285 datasets.

---

> ### Author Rebuttal · Authors · 2024-08-17
>
> Opportunities For Improvement:
> 1. Thank you for your thorough review. We have made the necessary revisions and now clearly state what GMAI is before using its abbreviation. This adjustment will be reflected in the camera-ready version of the paper.
> 2. We sincerely appreciate your suggestion. This benchmark is a long-term maintenance project, and we plan to not only continue adding more datasets but also introduce additional testing functionalities. For instance, we aim to incorporate tests for the model's chain-of-thought reasoning and analyze its rationale in future work, further enhancing the benchmark's comprehensiveness.
>
> Limitations:
> * Thank you for your meticulous review. We have noticed the text size issue in Figure 3, and we have enlarged it for better readability. This adjustment has already been applied and will be demonstrated in the camera-ready version.

---

### Official Review · Reviewer_cNaC · 2024-07-25
**A comprehensive and well-structured multimodal benchmark specifically designed to evaluate the effectiveness of large vision-language models (LVLMs) in diverse real-world clinical scenarios.**

**Rating:** 6
**Confidence:** 3

**Review:**

**Quality:**
- The evaluation methodology is rigorous, covering a wide range of models and tasks.

**Clarity:**
- The paper is well-written and clearly structured, making it easy to follow the methodology and findings.
- Technical terms and concepts are well-explained, ensuring comprehensibility for readers with a background in medical AI.

**Originality:**
- The lexical tree structure for organizing tasks and departments is a novel approach that enhances the benchmark's usability.

**Significance:**
- The benchmark addresses a critical need in the field of medical AI, providing a robust tool for evaluating LVLMs in clinical settings.
- The insights gained from the evaluation of current models highlight significant challenges and areas for improvement, guiding future research.

### Pros and Cons

**Pros:**
1. **Comprehensive Dataset:** The inclusion of 285 datasets covering 38 medical image modalities ensures broad and diverse representation.
2. **Novel Benchmark Structure:** The lexical tree structure allows for customizable evaluations tailored to specific clinical needs.
3. **Rigorous Evaluation:** The extensive evaluation of 50 LVLMs provides valuable insights into current model performance and limitations.

**Cons:**
1. The benchmark's comprehensive nature might be overwhelming for some users, requiring significant effort to fully utilize.
2. The article should include open-source datasets and code for LVM evaluation, which are currently not provided. This omission requires attention.

**Strengths:**

GMAI-MMbench is a significant contribution to the field of medical AI, providing a detailed and extensive benchmark that covers a wide range of medical image modalities and clinical tasks. This allows for a more accurate and thorough evaluation of LVLMs in medical contexts.
And besides, the GMAI-MMbench employs a lexical tree structure to organize its vast collection of datasets and tasks, enabling a flexible and hierarchical approach to evaluate LVLMs in medical applications. This structure allows for easy navigation and customization, accommodating the diverse requirements of clinical practice.

**Additional Feedback:**

Should the benchmark's code and dataset become open-sourced in the future, I would revisit the score.

**Clarity:**

The paper is clearly written, allowing me to easily understand all the presented arguments.

**Correctness:**

The dataset is constructed appropriately and the benchmarking follows standard evaluation protocol for the task.

**Documentation:**

The paper discusses ethical considerations and the intended application of the technology in its conclusion. However, it lacks the code and dataset necessary to reproduce the benchmark results.

**Ethics:**

The submission appears to be free of ethical issues.

**Limitations:**

As I indicated above, although I saw the detailed composition of the dataset in the attached material, but the dataset as well as benchmark codes need to be provided.

**Opportunities For Improvement:**

The article would merit an 8 out of 10 if it included open-sourced code and datasets for the benchmarks. This aspect needs to be addressed.

**Relation To Prior Work:**

The dataset in this article incorporates numerous existing open-source and hopital datasets, ensuring its accuracy. This is detailed and compared in Table 1.

**Summary And Contributions:**

The paper introduces **GMAI-MMbench**, a comprehensive multimodal benchmark designed to evaluate large vision-language models (LVLMs) in medical applications. It features 285 datasets from 38 medical image modalities and 19 clinical VQA tasks. The evaluation of 50 LVLMs reveals significant challenges, with the top model achieving only 52% accuracy.

Contributions:
1. Comprehensive Benchmark:
   Dataset Diversity: Includes 285 datasets covering 38 medical image modalities.
   Lexical Tree Structure: Organizes data into 19 clinical VQA tasks across 18 departments for customizable evaluations.
2.Comprehensive LVLM Evaluation:
  Evaluates 44 public LVLMs and advanced models like GPT-4o.
  Performance Insights:Identifies key issues such as perceptual errors and lack of domain knowledge.

---

> ### Author Rebuttal · Authors · 2024-08-17
>
> Cons 1: Thank you for pointing out that the current benchmark may be overwhelming for some users. To address this concern, we are integrating the benchmark into the VLMEvalkit (https://arxiv.org/abs/2407.11691). This open-source tool will make it easier for users to utilize the benchmark, ensuring a more accessible and user-friendly experience, which will be onboard in the final version.
>
> Cons 2: Due to the NeurIPS anonymous link policy, we open-sourced our data and their processing scripts on our Hugging Face and Github, and we will demonstrate that information in the camera-ready paper. We will ensure that the corrected paper and the associated resources are made available together.
>
> Limitations:
> We are very grateful for your suggestions. We have already made the dataset available on Hugging Face and have open-sourced our code on GitHub. Additionally, we are currently working on supporting the VLMEvalkit. All relevant information will be included in the camera-ready version of the paper, ensuring that everything is clearly presented and accessible.

---

> ### Author Response · Authors · 2024-08-29
> **The Need for Open-Sourced Code and Datasets for the Benchmark**
>
> Dear Reviewer cNaC,
>
> Thank you very much for taking the time to review our manuscript. We have addressed all your concerns regarding the release aspect.
>
> We have confirmed that it is permissible to share the official links related to our benchmark. Below, we have included the links to our
> * [Official Website](https://uni-medical.github.io/GMAI-MMBench.github.io)
> * [GitHub Repo](https://github.com/uni-medical/GMAI-MMBench)
> * [HuggingFace](https://huggingface.co/datasets/OpenGVLab/GMAI-MMBench)
> * [VLMEvalKit](https://github.com/open-compass/VLMEvalKit/), [VLMEvalKit Code Line](https://github.com/open-compass/VLMEvalKit/blob/main/vlmeval/dataset/image_mcq.py#L381)
>
> As the discussion period is coming to an end, may we ask if any aspects of our release remain unclear to you?
>
> We would be more than happy to provide further clarifications and engage in additional discussions to ensure all your concerns are fully addressed.
>
> Best regards,\
> Authors of Paper 900

---

### Decision · Program_Chairs · 2024-09-26

**Decision:**

Accept (Poster)

**Comment:**

The authors highlighted key strengths of their benchmark, including its comprehensive dataset, which spans 285 datasets across 38 medical image modalities, and their novel lexical tree structure that enhances customizable evaluation. The benchmark's ability to rigorously evaluate 50 large vision-language models (LVLMs) offers valuable insights into their current capabilities and areas for improvement in medical AI. The authors have responded to the reviewers' concerns. They have acknowledged feedback and made revisions, including releasing the benchmark’s code and dataset, which was a significant request from the reviewers. They addressed concerns about ethical reviews of private datasets, patient information's role in clinical decisions, and the risk of data leakage. The authors have also provided links to the benchmark's official website, GitHub, and HuggingFace pages, confirming their open-source status. While some concerns remain about the lack of patient-specific information in the current benchmark, the authors plan to incorporate this in future iterations.